# Beyond Least Squares: Uniform Approximation and the Hidden Cost of Misspecification

**Davide Maran**
Politecnico di Milano
davide.maran@polimi.it

**Csaba Szepesvári**
Google DeepMind and University of Alberta
szepi@google.com

## Abstract

We study the problem of controlling worst-case errors in misspecified linear regression under the random design setting, where the regression function is estimated via (penalized) least-squares. This setting arises naturally in value function approximation for bandit algorithms and reinforcement learning (RL). Our first main contribution is the observation that the amplification of the misspecification error when using least-squares is governed by the *Lebesgue constant*, a classical quantity from approximation theory that depends on the choice of the feature subspace and the covariate distribution. We also show that this dependence on the misspecification error is tight for least-squares regression: in general, no method minimizing the empirical squared loss, including regularized least-squares, can improve it substantially. We argue this explains the empirical observation that some feature-maps (e.g., those derived from the Fourier bases) "work better in RL" than others (e.g., polynomials): given some covariate distribution, the Lebesgue constant is known to be highly sensitive to choice of the feature-map. As a second contribution, we propose a method that augments the original feature set with auxiliary features designed to reduce the error amplification. We then prove that the method successfully competes with an "oracle" that knows the best way of using the auxiliary features to reduce this amplification. For example, when the domain is a real interval and the features are monomials, our method reduces the amplification factor to $\mathcal{O}(1)$ as $d \to \infty$, while without our method, least-squares with the monomials (and in fact polynomials) will suffer a worst-case error amplification of order $\Omega(d)$. It follows that there are functions and feature maps for which our method is consistent, while least-squares is inconsistent.

## 1 Introduction

Value function approximation plays a central role in modern reinforcement learning (RL) and contextual bandit algorithms [Sutton and Barto, 2018, Lattimore and Szepesvári, 2020]. In many such settings, policies are evaluated or selected based on value estimates obtained by regressing observed returns. To this end, (penalized) linear regression—based on empirical squared loss—serves as a core subroutine due to its simplicity and favorable computational properties [Ernst et al., 2005, Antos et al., 2008]. A fundamental challenge arises, however, when the true value function or reward model lies outside the span of the chosen features—a situation referred to as *model misspecification*. Recent work by Du et al. [2020] highlighted that in this setting for any $d$, there are feature maps so that the worst-case prediction error incurred by least-squares regression can be $\sqrt{d}$ larger than the misspecification error, even if the learner can control the covariate distribution.[1] This amplification is concerning because it implies that adding more features may not improve the learner's performance if

---

[1]Du et al. [2020] prove the stronger result that any learner that has access to a polynomially sized sample is ought to suffer a "large" worst-case prediction error no matter the method they use.

the misspecification error decreases at a rate of $\Omega(1/\sqrt{d})$. As such, in the RL and bandits communities much ink was spilled on the implications and the control of error amplification [Lattimore et al., 2020, Dong and Yang, 2023, Amortila et al., 2023, Maran et al., 2024, Amortila et al., 2024].

In this paper, we further investigate this error amplification and suggest a method designed to drastically reduce it. We study this problem in the context of *misspecified linear regression* under the *random design setting*, where inputs are drawn from an unknown distribution. Our first contribution is to identify how the amplification of the misspecification error depends directly on the interaction between the sampling distribution and the feature subspace. Specifically, we show that this amplification is governed by the *Lebesgue constant*—a classical quantity in approximation theory that captures how well the 2-norm projection underlying least-squares regression projects arbitrary functions onto the span of the features. This result provides a significant refinement of previous results in this direction. While prior work established a worst-case, feature-agnostic amplification factor of $\sqrt{d}$ (which is known to be tight for some feature maps), our approach identifies the governing principle for this amplification, explaining why the true factor can range from as low as 1 for favorable features. This distinction is critical, as it allows for significantly tighter finite-sample guarantees than those derived from the universal $\sqrt{d}$ scaling. Moreover, we prove that this dependence on the Lebesgue constant is tight: no estimator based on least-squares can substantially improve upon this bound. This sensitivity of least-squares regression to misspecification is in fact a modern re-emergence of a phenomenon that is well known in classical approximation theory. When polynomial bases are used on equispaced grids, the Runge phenomenon causes the uniform error to explode near the boundary despite good $L^2$ behaviour, precisely because the associated Lebesgue constants grow in an uncontrolled way. Likewise, in Fourier approximation, the well-known Gibbs phenomenon, although far milder than the Runge blow-up, causes localized oscillatory overshoots near discontinuities [Gottlieb and Shu, 1997].

The disparity in error severity is not accidental. It corresponds directly to their differing Lebesgue constant growth rates: logarithmic for Fourier bases, exponential (in the case of equispaced grids) for polynomials. Yet, despite its central role in approximation theory since the early works of Gregory et al. [1848], De La Vallée Poussin et al. [1919] and Szegő [1939], the link between the Lebesgue constant and misspecification error has, to the best of our knowledge, been overlooked in the modern statistical and reinforcement learning literature. In fact, while the superiority of Fourier bases over polynomials for value function approximation has been empirically observed (see Konidaris et al. [2011]), this phenomenon has lacked a precise theoretical explanation.

Motivated by this connection to the Lebesgue constant, our second contribution is a method for *reducing misspecification error amplification*. The approach works by augmenting the original feature set with auxiliary features and then using a weighted ridge regression approach to explicitly regularize the corresponding projection operator to give small error amplification. To illustrate, we show that when the domain is an interval and the base and auxiliary features are monomials, our method reduces the amplification factor to $\mathcal{O}(1)$ as $d \to \infty$. In contrast, standard least squares suffers from arbitrarily large worst-case errors in the same setting.

## 2 Problem Formulation

We consider the problem of estimating a function with *uniform* accuracy using a misspecified linear model. We first detail the statistical setting, introduce the standing assumptions and define the performance criterion that will be used in the rest of the paper.

Let $\mathbb{R}$ denote the set of reals, $\mathcal{X}$ be a measurable input space, and let $f : \mathcal{X} \to \mathbb{R}$ denote an unknown measurable *target* (or regression) function that we wish to estimate from a sample $D_n = ((x_1, y_1), \ldots, (x_n, y_n))$ of $n$ independent, identically distributed pairs that belong to $\mathcal{X} \times \mathbb{R}$ and which satisfy that almost surely for all $t \in [n] := \{1, \ldots, n\}$,

$$f(x_t) = \mathbb{E}[y_t | x_t].$$

We denote the marginal distribution of the inputs $x_t$ by $\mu$, while the distribution underlying $(x_t, y_t)$ is denoted by $P$.

**Assumption 1** (Sub–Gaussian Noise). *For some $\sigma > 0$, $y_1 - f(x_1)$ is $\sigma$-subgaussian conditionally on $x_1$. That is, almost surely,*

$$\lambda \in \mathbb{R} \qquad \mathbb{E}[\exp(\lambda(y_1 - f(x_1)))|x_1] \leq \exp(\lambda^2 \sigma^2 / 2)$$

.

We are interested in the problem of *linear function approximation*. That is, the goal is to approximate $f$ as a linear combination of $d$ *basis functions*, $\varphi_1, \ldots, \varphi_d : \mathcal{X} \to \mathbb{R}$. When these are clear from the context, for $\theta \in \mathbb{R}^d$, we let $f_\theta : \mathcal{X} \to \mathbb{R}$ be defined via

$$f_\theta(x) = \sum_{i=1}^{d} \theta_i \varphi_i(x), \quad x \in \mathcal{X}.$$

We shall also collect them into a *feature-map* $\boldsymbol{\varphi} : \mathcal{X} \to \mathbb{R}^d$ and write $f_\theta(x) = \boldsymbol{\varphi}(x)^\top \theta$, where $\boldsymbol{\varphi}(\cdot) = (\varphi_1(\cdot), \ldots, \varphi_d(\cdot))^\top$. Motivated by the applications mentioned earlier, we depart from the bulk of the literature in this setting and evaluate performance via the uniform (or maximum) norm. For a function $g : \mathcal{X} \to \mathbb{R}$, this is defined as $\|g\|_\infty = \sup_{x \in \mathcal{X}} |g(x)|$. We let $L^\infty(\mathcal{X})$ denote the set of functions with finite maximum norm. **In what follows, we assume that both $f$ and our basis functions $\varphi_i$ belong to this set.** For $f \in L^\infty(\mathcal{X})$ and $\theta \in \mathbb{R}^d$ we let

$$\mathcal{E}_\infty(\theta, f) := \|f_\theta - f\|_\infty, \qquad \mathcal{E}_\infty(f) := \inf_{\theta \in \mathbb{R}^d} \mathcal{E}_\infty(\theta, f).$$

Thus, $\mathcal{E}_\infty(\theta, f)$ is the maximum error suffered when approximating $f$ with $f_\theta$. The quantity $\mathcal{E}_\infty(f)$ represents the best possible uniform approximation error achievable by our basis functions, and its value is *unknown* to the learner. When $\mathcal{E}_\infty(f) > 0$, we refer to this as the *misspecified setting*, and refer to $\mathcal{E}_\infty(f)$ as the *misspecification error*. In the next section, we investigate the behavior of the error $\mathcal{E}_\infty(\hat{\theta}_n, f)$ when $\hat{\theta}_n$ is the *ordinary least-squares* (OLS) parameter estimate[2]:

$$\hat{\theta}_n = \arg\min_{\theta \in \mathbb{R}^d} \sum_{t=1}^{n} \left(y_t - \boldsymbol{\varphi}(x_t)^\top \theta\right)^2.$$

As we will see, while the OLS estimator is simple, a rigorous analysis of its uniform error under misspecification is involved.

## 3 Characterizing the behavior of OLS

Let $\mathcal{F} = \{f_\theta : \theta \in \mathbb{R}^d\}$ denote the subspace of $L^\infty(\mathcal{X})$ spanned by the basis functions underlying the feature-map $\boldsymbol{\varphi}$. **From now on we assume that the population Gram matrix $V_\mu = \int \boldsymbol{\varphi}(x) \boldsymbol{\varphi}(x)^\top \mu(dx)$ is non-singular.** As $n \to \infty$, the Strong Law of Large Numbers ensures that $f_{\hat{\theta}_n}$ converges almost surely to

$$\Pi_{d,\mu} f := \arg\min_{g \in \mathcal{F}} \|g - f\|_\mu^2, \tag{1}$$

where $\| \cdot \|_\mu$ denotes the $L^2(\mu)$-norm, defined for any measurable $g : \mathcal{X} \to \mathbb{R}$ by $\|g\|_\mu^2 = \int_{\mathcal{X}} g^2(x) \mu(dx)$. Since $\mu$ is a probability measure, we have $\|g\|_\mu^2 \leq \|g\|_\infty^2$. The map $\Pi_{d,\mu}$ defined in Eq. (1) is the *orthogonal projection* onto $\mathcal{F}$ with respect to the $L^2(\mu)$ inner product and is well-defined thanks to our assumption on the Gram matrix $V_\mu$. The map $\Pi_{d,\mu}$ is linear, idempotent and satisfies $\Pi_{d,\mu} f = f$ for all $f \in \mathcal{F}$. Moreover, it is non-expansive in the $L^2(\mu)$-norm.

By continuity, the almost sure convergence of the OLS estimate implies that, almost surely, $\lim_{n \to \infty} \mathcal{E}_\infty(\hat{\theta}_n, f) = \|\Pi_{d,\mu} f - f\|_\infty$. The first question we must address is how large $\|\Pi_{d,\mu} f - f\|_\infty$ can be relative to the best possible error, $\mathcal{E}_\infty(f)$. In other words, by how much is the misspecification error $\mathcal{E}_\infty(f)$ amplified when we use the $L^2(\mu)$ projection of $f$ onto $\mathcal{F}$? The following classical result provides an answer:

**Lemma 1** (Lebesgue's lemma, e.g., Proposition 4.1 from Chapter 2 of DeVore and Lorentz [1993] ). *Let $(\mathcal{S}, \| \cdot \|)$ be a normed vector space, $\mathcal{F}$ a subspace of $\mathcal{S}$ and $\Pi : \mathcal{S} \to \mathcal{S}$ be a linear map such that $\Pi(\mathcal{S}) \subseteq \mathcal{F}$ and $\Pi$ is an identity on $\mathcal{F}$.[3] Then, for any $f \in \mathcal{S}$,*

$$\|f - \Pi f\| \leq (1 + \|\Pi\|) \inf_{g \in \mathcal{F}} \|f - g\|,$$

*where $\|\Pi\| := \sup_{f \in \mathcal{S}: \|f\| \leq 1} \|\Pi f\|$ is the* operator norm *of $\Pi$.*

---

[2]In principle, this minimizer may not be unique; however, it is unique under the assumptions required for the results we are about to show.

[3]A map $\Pi$ with this property is called a projection. Note that orthogonal projections are projections, but there are projections of course which are not orthogonal with respect to any inner product.

When $\Pi$ maps between two normed spaces, $(X, \|\cdot\|_X)$ and $(Y, \|\cdot\|_Y)$, we denote its operator norm by $\|\Pi\|_{X \to Y}$. Applying this result to our setting with $(\mathcal{S}, \|\cdot\|) = (L^\infty(\mathcal{X}), \|\cdot\|_\infty)$ we obtain

$$\|f - \Pi f\|_\infty \leq (1 + \|\Pi\|_\infty) \inf_{g \in \mathcal{F}} \|f - g\|_\infty,$$

where $\|\Pi\|_\infty$ denotes the operator norm $\|\Pi\|_{L^\infty(\mathcal{X}) \to L^\infty(\mathcal{X})}$. In honor of its discoverer, this quantity is formally named as follows:

**Definition 1** (Lebesgue constant). *For a linear operator $\Pi : L^\infty(\mathcal{X}) \to L^\infty(\mathcal{X})$, its induced norm is called the* Lebesgue constant *associated with $\Pi$, and is denoted by*

$$\Lambda(\Pi) := \|\Pi\|_\infty = \sup_{f \in L^\infty(\mathcal{X}):\|f\|_\infty \leq 1} \|\Pi f\|_\infty.$$

Since the Lebesgue constant of our projection operators will be frequently needed, to minimize clutter we introduce the shorthand

$$\Lambda_{d,\mu} := \Lambda(\Pi_{d,\mu}).$$

Notably, it is the subspace $\mathcal{F}$—not the specific feature map used to define it—that is the fundamental object: $\mathcal{F}$ alone determines the intrinsic misspecification error $\mathcal{E}_\infty(f)$, while its interplay with $\mu$ governs the projection $\Pi_{d,\mu}$ and the amplification factor $\Lambda_{d,\mu}$. With the notation just introduced, Lemma 1 yields the upper bound

$$\|f - \Pi_{d,\mu}f\|_\infty \leq (1 + \Lambda_{d,\mu})\mathcal{E}_\infty(f). \tag{2}$$

It is easy to see that $\Lambda_{d,\mu} \geq 1$ (consider any nonzero $f \in \mathcal{F}$). Unfortunately, there is no general upper limit on how large this constant can be. Moreoover, the bound in Lemma 1 is essentially tight:

**Theorem 1.** *For any $\varepsilon > 0$ and any subspace $\mathcal{F}$ there exist $f \in L^\infty(\mathcal{X})$ such that*

$$\|f - \Pi_{d,\mu}f\|_\infty \geq (\Lambda_{d,\mu} - 1 - \varepsilon)\mathcal{E}_\infty(f).$$

We relegate all proofs to the Appendix. The proof of this specific result can be found in Appendix C.1.

Given the tightness of the lower bound established above, we expect any guarantee on the uniform error $\mathcal{E}_\infty(\hat{\theta}_n, f)$ to inherently involve $\Lambda_{d,\mu}\mathcal{E}_\infty(f)$. Our main result of this section confirms this.

To state the result, we need one additional quantity characterizing the feature space. Let $(\overline{\varphi}_i)_{1 \leq i \leq d}$ be an orthonormal basis of $\mathcal{F}$ with respect to $L^2(\mu)$ (obtained, for instance, via the Gram-Schmidt procedure on the original basis functions). We let $\overline{\varphi} : \mathcal{X} \to \mathbb{R}^d$ denote the orthogonalized featuremap defined via $\overline{\varphi}(x) = (\overline{\varphi}_1(x), \ldots, \overline{\varphi}_d(x))^\top$. We further define $\vartheta_{d,2} = \sup_{x \in \mathcal{X}} \|\overline{\varphi}(x)\|_2$.

While the orthonormal basis $(\overline{\varphi}_i)_{1 \leq i \leq d}$ is not unique, the quantity $\vartheta_{d,2}$ is *uniquely defined*. In particular (see Proposition 16), it is the $L^2(\mu) \to L^\infty(\mathcal{X})$ operator norm of the projection $\Pi_{d,\mu}$:

$$\vartheta_{d,2} = \|\Pi_{d,\mu}\|_{L^2(\mu) \to L^\infty(\mathcal{X})}. \tag{3}$$

Being $\mu$ a probability measure, $\Lambda_{d,\mu} = \|\Pi_{d,\mu}\|_{L^\infty(\mathcal{X}) \to L^\infty(\mathcal{X})} \leq \|\Pi_{d,\mu}\|_{L^2(\mu) \to L^\infty(\mathcal{X})} = \vartheta_{d,2}$.

**Theorem 2.** *Let $\mathcal{X}$ be finite and Assumption 1 hold. Let $\mathcal{X}$ be finite and Assumption 1 hold. For any $\delta \in (0, 1/3]$ and any $n \geq 20\vartheta_{d,2}^2 \log(d/\delta)$, the OLS estimate $\hat{\theta}_n$ satisfies, with probability at least $1 - 3\delta$,*

$$\mathcal{E}_\infty(\hat{\theta}_n, f) \leq (1 + \Lambda_{d,\mu})\mathcal{E}_\infty(f) + 3(\sigma + \Lambda_{d,\mu}\mathcal{E}_\infty(f))\vartheta_{d,2}\sqrt{\frac{\log(|\mathcal{X}|/\delta)}{n}}$$

$$+ \frac{poly(d, \vartheta_{d,2}, \Lambda_{d,\mu}\mathcal{E}_\infty(f))}{n}.$$

The first term in the bound matches the deterministic amplification derived in Eq. (2), accounting for the irreducible approximation gap between $\Pi_{d,\mu}f$ and $f$. The remaining terms bound the additional finite-sample stochastic error. While stated for finite $\mathcal{X}$ for simplicity, the result extends to continuous domains via standard covering arguments For example, if $\mathcal{X} = [-1, 1]$ and the basis functions are $L_\varphi$-Lipschitz, a uniform bound over $\mathcal{X}$ can be obtained by considering an $\varepsilon/L_\varphi-$cover of $\mathcal{X}$. This

extension incurs only a mild logarithmic factor proportional to $\log(L_\varphi/\varepsilon)$. Alternatively, a sometimes tighter bound can be achieved by covering the feature set $\{\boldsymbol{\varphi}(x) \,:\, x \in \mathcal{X}\} \subset \mathbb{R}^d$ directly.

Beyond $\Lambda_{d,\mu}$, the bound also depends on the constant $\vartheta_{d,2}$, which scales with the dimension $d$. In particular, we show that $\vartheta_{d,2} \geq \sqrt{d}$ holds regardless of the chosen feature map (see Proposition 18 in the appendix). The scaling of the terms in Theorem 2 aligns with standard expectations. The first term represents the unavoidable *approximation error* (bias) discussed previously. The remaining terms quantify the *estimation error* (variance) due to finite sampling. Specifically, the term that involves $\sigma$ captures the effect of additive noise, while the remaining term accounts for the additional variance induced by the random design, which itself is amplified by the Lebesgue constant and the intrinsic misspecification level. Below we show that when an *a priori* upper bound $\varepsilon$ on $\mathcal{E}_\infty(f)$ is available (as can be the case in certain numerical applications when the target function belongs to some known class of functions, such as a smoothness class), we can obtain a *semi-empirical bound*. This bound, which relies on data-dependent quantities, has the potential to significantly improve upon the worst-case bound given in Theorem 2.

**A uniform, semi-empirical bound**  Our purpose here is to bound the uniform error of the OLS estimate using empirical quantities. Let $\mu_n = \frac{1}{n}\sum_{t=1}^n \delta_{x_t}$ denote the empirical measure associated with the inputs $(x_1, \ldots, x_n)$. A key advantage of this analysis is that it relies on the empirical Lebesgue constant $\Lambda_{d,\mu_n}$ associated with $\Pi_{d,\mu_n}$, allowing us to drop all assumptions regarding how the inputs are generated (i.e., it applies to fixed design). The empirical operator $\Pi_{d,\mu_n}$ takes the form

$$\Pi_{d,\mu_n} f(\cdot) := \boldsymbol{\varphi}_d(\cdot)^\top (\Phi^\top \Phi)^{-1} \Phi^\top \mathbf{f}_n \qquad \text{where} \qquad \mathbf{f}_n := [f(x_1), \ldots, f(x_n)]^\top,$$

and $\Phi \in \mathbb{R}^{n \times d}$ is the design matrix with rows $\boldsymbol{\varphi}^\top(x_t)$. Analogous to the population case, assuming that $V_{\mu_n}$ **is non-singular**, we let $(\widehat{\varphi}_i)_{1 \leq i \leq d}$ be an orthonormal basis of $\mathcal{F}$ in $L^2(\mu_n)$, define the feature map $\widehat{\boldsymbol{\varphi}} : \mathcal{X} \to \mathbb{R}^d$ via $\widehat{\boldsymbol{\varphi}}(x) = (\widehat{\varphi}_1(x), \ldots, \widehat{\varphi}_d(x))^\top$, and let $\vartheta_{d,2}^{(n)} = \sup_{x \in \mathcal{X}} \|\widehat{\boldsymbol{\varphi}}(x)\|_2$.

**Theorem 3.** *Let $\mathcal{X}$ be finite, Assumption 1 hold, $\hat{\theta}_n$ be the OLS estimate. Then, for any fixed $\delta > 0$, with probability at least $1 - \delta$,*

$$\mathcal{E}_\infty(\hat{\theta}_n) \leq (1 + \Lambda_{d,\mu_n})\mathcal{E}_\infty(f) + \frac{\sigma \vartheta_{d,2}^{(n)} \sqrt{2\log(2|\mathcal{X}|/\delta)}}{\sqrt{n}}.$$

Compared to Theorem 2, this bound offers several improvements: it eliminates the lower-order $\mathcal{O}(1/n)$ term entirely and it removes the dependence on the misspecification error $\mathcal{E}_\infty(f)$ from the leading stochastic term (the $1/\sqrt{n}$ term). Furthermore, the population Lebesgue constant $\Lambda_{d,\mu}$ is replaced by its empirical counterpart $\Lambda_{d,\mu_n}$, which may be smaller than $\Lambda_{d,\mu}$. When $\mathcal{X}$ is finite, $\Lambda_{d,\mu_n}$, which is a matrix maximum norm, can be calculated in $\mathcal{O}(n|\mathcal{X}|)$ time.

If the input points $(x_1, \ldots, x_n)$ can be chosen, one may attempt to optimize this bound directly. Both $\Lambda_{d,\mu_n}$ and $\vartheta_{d,2}^{(n)}$ depend on $\mu_n$. In experimental optimal design, a *G-optimal design* minimizes $\vartheta_{d,2}^{(n)}$ by carefully selecting $\mu_n$. A fundamental result by Kiefer and Wolfowitz [1960] establishes that for $n = \Omega(d)$, there exists a design $\mu_n$ such that $\vartheta_{d,2}^{(n)} = \mathcal{O}(\sqrt{d})$, which is the best possible scaling.

Assuming that $\mu_n$ is a $G$-optimal design, we can compare our result with Proposition 5.1 from Lattimore et al. [2020] (see their equation (2) and the corresponding bound in high probability). Rephrased in our notation, their bound (adapted to high probability with $\sigma = 1$) states that if $\mu$ is an optimal design, then

$$\mathcal{E}_\infty(\hat{\theta}_n, f) \leq \mathcal{O}\left(\sqrt{d}\mathcal{E}_\infty(f) + \sqrt{\frac{d\log(|\mathcal{X}|/\delta)}{n}}\right).$$

This result is recoverable as a special case of our Theorem 2. Indeed, for a $G$-optimal design, $\vartheta_{d,2}^{(n)} = \sqrt{d}$, and it is straightforward to show that $\Lambda_{d,\mu_n} \leq \vartheta_{d,2}^{(n)}$ always holds. In the large sample limit ($n \to \infty$), the term $\sqrt{d}\mathcal{E}_\infty(f)$ dominates their bound. Therefore, our finer bound involving the Lebesgue constant yields a strictly better guarantee whenever $\Lambda_{d,\mu_n}$ is smaller than $\sqrt{d}$. For example, consider a *partition-based feature map* where each $\varphi_i$ is an indicator function of a distinct region $\mathcal{X}_i$ in a partition of $\mathcal{X}$. In this case, $\Lambda_{d,\mu} = 1$ regardless of $\mu$, offering a massive improvement over the worst-case $\sqrt{d}$-factor. While Proposition 5.1 can be refined to replace $\sqrt{d}$ by $\vartheta_{d,2}^{(n)}$ in the variance term, this improvement still falls short of recovering the $\Lambda_{d,\mu_n}$ factor in the bias term.

| Basis functions | $\mu$ | $\Lambda_{d,\mu}$ | Source | Note |
|---|---|---|---|---|
| Polynomial | uniform on regular $d$-grid | $\Omega(2^d)$ | [Quarteroni et al., 2010] | |
| Polynomials | uniform | $\Theta(d)$ | DeVore and Lorentz [1993] | $\vartheta_{d,2} \approx d$ |
| Fourier | uniform | $\mathcal{O}(\log(d))$ | [Katznelson, 2004, p.59, Ex. 1] | |
| Continuous B-splines | uniform | $\mathcal{O}(1)$ | Huang [2003] | |
| Wavelets | uniform | $\mathcal{O}(1)$ | Chen and Christensen [2013] | |

Table 1: Examples of Lebesgue constants. Domain is $\mathcal{X} = [-1, 1]$.

### 3.1 The Lebesgue constant: properties and particular cases

While we established with Eq. (3) that $\Lambda_{d,\mu} \leq \vartheta_{d,2}$ always hold, $\vartheta_{d,2}$ itself is lower-bounded by $\sqrt{d}$ (see Proposition 18). To find cases where $\Lambda_{d,\mu}$ is significantly smaller than $\sqrt{d}$, we must look at specific feature maps. Table 1 summarizes known results for several classical bases on $[-1, 1]$.

As shown in the table, the Lebesgue constant varies dramatically depending on the basis and measure. Polynomials on a regular $d$-grid exhibit the worst behavior, with exponential growth $\Omega(2^d)$. Even with a uniform measure, polynomials still suffer from a linear growth $\Theta(d)$. In stark contrast, Fourier series enjoy a much slower logarithmic growth $\mathcal{O}(\log d)$. It follows that if a target function's $L^2$ approximation error decreases as $\mathcal{O}(1/d^s)$ with some $s > 0$, the additional uniform error incurred by least-squares is minimal for $d$ large. Interestingly, as was noted earlier, empirical work in reinforcement learning has identified Fourier bases as a strong general-purpose choice [Konidaris et al., 2011]. Our analysis provides a theoretical justification for this: their slowly growing Lebesgue constant ensures reasonable error control even under misspecification.

Finally, *localized basis functions* like wavelets and B-splines achieve the ideal constant scaling $\mathcal{O}(1)$, independent of $d$. This makes them excellent candidates when uniform accuracy is paramount. We speculate that *tile coding*, a popular localized representation in RL, likely shares these favorable extrapolation properties.

A practical limitation of relying on tabulated Lebesgue constants is their dependence on the specific sampling distribution $\mu$. Calculating these constants is non-trivial, and standard results typically only exist for simple, idealized distributions (e.g., uniform). The following proposition provides a way to transfer these known bounds to other distributions, provided they are not too dissimilar:

**Proposition 4.** *Let $\mu, \nu$ be two discrete probability measures supported on a countable set $\mathcal{X}$ such that for all $x \in \mathcal{X}$, $0 \leq c \leq \frac{\mu(x)}{\nu(x)} \leq C$. Then, $\Lambda_{d,\mu} \leq \frac{C}{c}\Lambda_{d,\nu}$.*

## 4  Regularized estimators

Crefthm:lowerboundone establishes a fundamental limitation of the OLS estimator: its worst-case error is inescapably amplified by the Lebesgue constant. Importantly, this bound holds even in the infinite data limit, meaning the issue is not standard overfitting to finite-sample noise. Rather, the problem stems from the geometry of the $L^2(\mu)$-projection itself: due to the rigidity of the feature subspace, minimizing the average error can force the projection $\Pi_{d,\mu}f$ to exhibit large oscillations entirely absent from the target $f$, particularly in low-density regions.

A natural strategy to dampen such oscillations is regularizing the loss. In the next theorem, however, we show that the standard Ridge Regression approach is ineffective for this purpose, even when the ideal orthonormal basis $\bar{\varphi}_d$ is known and used.

**Theorem 5.** *Let $\hat{\theta}_{n,\text{RIDGE}}$ be the $\lambda-$ridge regression estimate. For any feature map $\varphi_d(\cdot) : \mathcal{X} \to \mathbb{R}^d$, there exists a target function $f \in L^{\infty}(\mathcal{X})$ such that, in the infinite data limit,*

$$\mathcal{E}_{\infty}(\hat{\theta}_{\infty,\text{RIDGE}}) = \Omega\left(\max\left\{(\Lambda_{d,\mu} - 2\lambda)\mathcal{E}_{\infty}(f), \frac{\lambda}{\lambda+1}\right\}\right).$$

This result highlights a "damned if you do, damned if you don't" dilemma for ridge regression. If we choose a large penalty $\lambda \approx \Lambda_{d,\mu}/2$ to counteract the amplification, the second term in the lower bound

approaches 1, preventing convergence even as the misspecification error $\mathcal{E}_\infty(f) \to 0$. Conversely, if $\lambda$ is small, we essentially recover the poor $\Omega(\Lambda_{d,\mu})$ worst-case bound of OLS. Crucially, this phenomenon persists even in the infinite data regime, indicating that it is not merely a sample size issue, but a geometric defect of the projection operator itself. Consequently, standard techniques designed to achieve small test mean-squared error —such as cross-validation or early stopping [Ghojogh and Crowley, 2019]—cannot overcome this fundamental geometric limitation, as they will asymptotically converge to the OLS solution, which is what minimizes the test mean-squared error.

Let us examine why ridge regression fails. The proof of Theorem 5 relies on the explicit form of the corresponding ridge operator $\Pi_{d,\mu}^{\text{Ridge}}$ in the infinite data limit:

$$\Pi_{d,\mu}^{\text{Ridge}} f(x) = \alpha \sum_{i=1}^{d} \overline{\varphi}_i(x) \int_{\mathcal{X}} \overline{\varphi}_i(z) f(z) \, d\mu(z) \qquad \text{where } \alpha = \frac{1}{1+\lambda}. \tag{4}$$

Importantly, this is *not* a projection operator because it does not preserve functions in $\mathcal{F}$. For example, applying this operator to the basis function $f = \overline{\varphi}_1$ yields an error of $\overline{\varphi}_1 - \Pi_{d,\mu}^{\text{Ridge}}\overline{\varphi}_1 = \frac{\lambda}{1+\lambda}\overline{\varphi}_1$. To obtain an error bound that scales with the misspecification $\mathcal{E}_\infty(f)$ (i.e., a bound that is zero when $f \in \mathcal{F}$), every function in $\mathcal{F}$ must be a fixedpoint of the operator. In Eq. (4), this requires $\alpha = 1$, which forces $\lambda = 0$, bringing us back to OLS and its associated amplification problems.

**Stabilization via Feature Augmentation**   Instead of ridge regression, we propose a different approach: we augment the feature map and use the additional degrees of freedom purely to "stabilize" the operator, rather than to improve approximation: After all, our bounds depend on both $\mathcal{E}_\infty(f)$ and $\Lambda_{d,\mu}$. Selecting sufficiently many features $\mathcal{E}_\infty(f)$ may be under control; hence, the idea is to use additional features to control $\Lambda_{d,\mu}$. Let us denote our original feature map by $\boldsymbol{\varphi}_d$ and its corresponding subspace by $\mathcal{F}_d$. We now augment this map with $D - d$ additional features, yielding the extended map $\boldsymbol{\varphi}_D$. While these extra features could be arbitrary, many standard bases (Fourier, polynomials, splines) have a natural nested structure that provides a canonical sequence of extensions.

Let $\overline{\boldsymbol{\varphi}}_D$ denote the orthonormal basis for this extended space obtained via Gram-Schmidt on $\boldsymbol{\varphi}_D$ with respect to $L^2(\mu)$. We now define a weighted ridge regression operator on this extended basis. For any sequence of weights $\boldsymbol{\lambda} = (\lambda_1, \ldots, \lambda_D) \in [0, \infty)^D$, let

$$\Pi_{\boldsymbol{\alpha},\mu}^{\text{Ridge}} f(x) := \sum_{i=1}^{D} \alpha_i \overline{\varphi}_i(x) \int_{\mathcal{X}} \overline{\varphi}_i(z) f(z) \, d\mu(z) \qquad \text{where} \qquad \alpha_i = \frac{1}{1+\lambda_i}. \tag{5}$$

Crucially, we want this new operator to serve as a superior replacement for $\Pi_{d,\mu}$ while maintaining the approximation power of our original space $\mathcal{F}_d$. We do not aim to target the potentially better (but likely much less stable) approximation of the full space $\mathcal{F}_D$. This requirement forces us to use *zero* regularization on the first $d$ components ($\lambda_i = 0 \implies \alpha_i = 1$ for $i \leq d$), ensuring that every function in $\mathcal{F}_d$ is fixed by the operator. This leads to the set of valid *attenuation parameters*:

$$\mathcal{A}_d^D := \{\boldsymbol{\alpha} \in [0,1]^D : \forall i \leq d, \ \alpha_i = 1\}. \tag{6}$$

For any $\boldsymbol{\alpha} \in \mathcal{A}_d^D$, the operator $\Pi_{\boldsymbol{\alpha},\mu}^{\text{Ridge}}$ fixes every function in $\mathcal{F}_d$, thereby avoiding the pitfall of standard ridge regression.

**Remark 1** (Connection to Averaging Projections). *This formulation generalizes the classical technique of averaging projections to increase stability. For instance, the averaged operator $\overline{\Pi} = \frac{1}{D-d+1} \sum_{k=d}^{D} \Pi_{k,\mu}$ is exactly equivalent to Eq. (5) with a specific choice of linearly decaying weights $\boldsymbol{\alpha} \in \mathcal{A}_d^D$. The intuition is that while individual high-degree projections $\Pi_{k,\mu}$ may oscillate wildly, these oscillations often cancel out when averaged, leaving a stable estimate. Our framework allows for optimizing these weights directly.*

## 4.1   Weighted ridge estimator and the Oracle Operator

Every operator $\Pi_{\boldsymbol{\alpha},\mu}^{\text{Ridge}}$ with $\boldsymbol{\alpha} \in \mathcal{A}_d^D$ maintains the elements of the original subspace $\mathcal{F}_d$ as its fixed points. Hence, the operators satisfy the conditions of Lebesgue's Lemma (Lemma 1) with $\mathcal{F} = \mathcal{F}_d$. Denoting by $\Lambda_{\boldsymbol{\alpha},\mu}$ the Lebesgue constant of $\Pi_{\boldsymbol{\alpha},\mu}^{\text{Ridge}}$, we immediately obtain the following result:

**Proposition 6.** *Let* $\boldsymbol{\alpha} \in \mathcal{A}_d^D$ *(see* (6)*) and* $\Pi_{\boldsymbol{\alpha},\mu}^{Ridge}$ *be defined as in* (5)*. Then, for any* $f \in L^\infty(\mathcal{X})$,

$$\|f - \Pi_{\boldsymbol{\alpha},\mu}^{Ridge} f\|_\infty \leq (1 + \Lambda_{\boldsymbol{\alpha},\mu})\mathcal{E}_\infty(f).$$

Established in Proposition 6 that the error amplification is governed by $\Lambda_{\boldsymbol{\alpha},\mu}$, our goal is to select the parameter $\boldsymbol{\alpha} \in \mathcal{A}_d^D$ that minimizes this constant. We refer to this ideal choice as the ORACLE choice:

$$\boldsymbol{\alpha}_\mu^{\text{Oracle}} := \underset{\boldsymbol{\alpha} \in \mathcal{A}_d^D}{\arg\min}\, \Lambda_{\boldsymbol{\alpha},\mu}\,, \qquad \Lambda_\mu^{\text{Oracle}} := \min_{\boldsymbol{\alpha} \in \mathcal{A}_d^D} \Lambda_{\boldsymbol{\alpha},\mu}.$$

Unfortunately, $\boldsymbol{\alpha}_\mu^{\text{Oracle}}$ is unknown to the learner, as it depends on the unknown distribution $\mu$. The remainder of this section addresses two key questions:

- **Q1** Can we design a finite sample estimator whose error, for fixed $\boldsymbol{\alpha} \in \mathcal{A}_d^D$, asymptotically scales with $\Lambda_{\boldsymbol{\alpha},\mu}$?
- **Q2** Can we design a finite sample estimator whose error asymptotically scales with the optimal $\Lambda_\mu^{\text{Oracle}}$?

**Q1** To answer these questions, we first generalize Theorem 3 to incorporate regularization with a chosen parameter $\boldsymbol{\alpha}$. Although $\mu$ is unknown, we can define an empirical counterpart of the operator in Eq. (5) using the empirical measure $\mu_n$. Recalling that $\widehat{\boldsymbol{\varphi}}_D$ is the feature map obtained by orthogonalizing $\boldsymbol{\varphi}_D$ w.r.t. $\mu_n$, we have

$$\Pi_{\boldsymbol{\alpha},\mu_n}^{\text{Ridge}} f(\cdot) := \sum_{i=1}^D \alpha_i \widehat{\varphi}_i(\cdot) \frac{1}{n} \sum_{t=1}^n \widehat{\varphi}_i(x_t) f(x_t). \tag{7}$$

This empirical operator has two key properties: (1) it depends on the evaluations $f(x_t)$, and (2) its output is a linear combination of the basis functions $\widehat{\varphi}_i$. Property (1) allows us to estimate $\Pi_{\boldsymbol{\alpha},\mu_n}^{\text{Ridge}} f(\cdot)$ from noisy data by simply replacing the unknown values $f(x_t)$ with the observed targets $y_t$. Property (2) ensures that the resulting estimate can be parameterized as $\boldsymbol{\varphi}_D(\cdot)^\top \hat{\theta}$ for some coefficient vector $\hat{\theta}$. Specifically, let $R_n$ be the upper triangular matrix from the Gram-Schmidt procedure, such that $\boldsymbol{\varphi}_D(\cdot)^\top = \widehat{\boldsymbol{\varphi}}_D(\cdot)^\top R_n$. Letting $I_\alpha = \text{diag}(\boldsymbol{\alpha})$ be the the regularization weights matrix, we define our estimator as follows

$$\hat{\theta}_{n,\boldsymbol{\alpha}} := R_n^{-1} I_\alpha \frac{1}{n} \sum_{t=1}^n \widehat{\boldsymbol{\varphi}}_d(x_t) y_t. \tag{8}$$

**Theorem 7.** *Let Assumption 1 hold. Then, for any* $\delta > 0$, *with probability* $1 - \delta$,

$$\mathcal{E}_\infty(\hat{\theta}_{n,\boldsymbol{\alpha}}) \leq (1 + \Lambda_{\boldsymbol{\alpha},\mu_n})\mathcal{E}_\infty(f) + \frac{\sigma \widehat{\varphi}_{2,D} \sqrt{2 \log(2|\mathcal{X}|/\delta)}}{\sqrt{n}}.$$

This result confirms that the amplification error of our estimator scales with $\Lambda_{\boldsymbol{\alpha},\mu_n}$. To fully answer question **Q1**, we must show that for large $n$, this empirical constant is a good proxy for the population constant $\Lambda_{\boldsymbol{\alpha},\mu}$. The following proposition establishes this convergence:

**Proposition 8.** *Fix* $\delta > 0$. *With probability* $1 - \delta$, *the following bounds holds simultaneously for every* $\boldsymbol{\alpha} \in \mathcal{A}_d^D$: $|\vartheta_{D,2} - \vartheta_{D,2}^{(n)}| = \widetilde{\mathcal{O}}(\vartheta_{D,2}^2 \sqrt{\log(1/\delta)/n})$, *and*

$$|\Lambda_{\boldsymbol{\alpha},\mu_n} - \Lambda_{\boldsymbol{\alpha},\mu}| = \widetilde{\mathcal{O}}\left(\frac{\sqrt{d}\vartheta_{D,2}^2 \sqrt{\log(1/\delta)}}{\sqrt{n}} + \frac{\sqrt{d}\vartheta_{D,2}^3 \log(1/\delta)}{n}\right).$$

**Q2** For this more challenging goal, we must optimize $\boldsymbol{\alpha}$ to converge to the oracle value, despite not knowing the true distribution $\mu$. Our strategy is to rely on computable quantities: the empirical operator $\Pi_{\boldsymbol{\alpha},\mu_n}^{\text{Ridge}}$ (Eq. (7)) and its associated empirical Lebesgue constant $\Lambda_{\boldsymbol{\alpha},\mu_n}$. A key observation enables this approach: the Lebesgue constant is convex with respect to $\boldsymbol{\alpha}$:

**Proposition 9.** *The function* $J : \mathcal{A}_d^D \to (0, +\infty)$ *defined by* $J(\boldsymbol{\alpha}) := \Lambda_{\boldsymbol{\alpha},\mu_n}$ *is convex in* $\boldsymbol{\alpha}$.

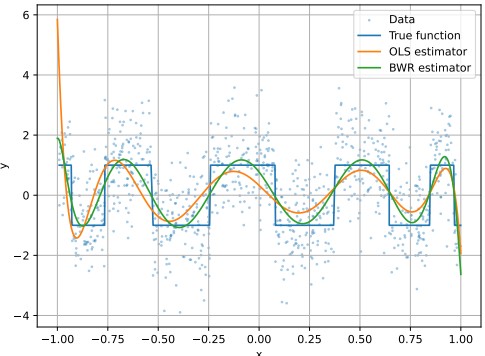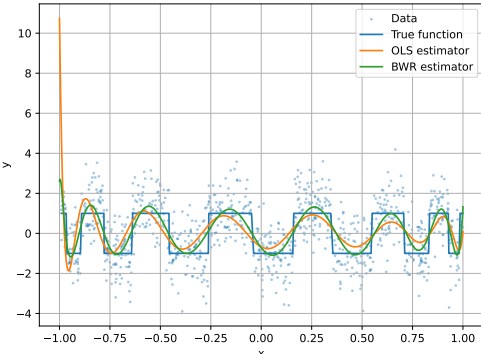

Figure 1: Comparison between the OLS estimator and the BWR estimator using polynomial features on $[-1, 1]$, with $d = 10$ features (left) and $d = 15$ features (right). The inputs are chosen uniformly at random from $[-1, 1]$. Even if the true function is bounded, OLS suffers from large oscillations near the boundaries due to the high Lebesgue constant. In contrast, BWR achieves a much more uniform approximation error across the domain by effectively controlling the amplification effect.

This convexity allows us to provably find an approximate minimizer in a finite number of iterations. We employ a standard subgradient method [Boyd et al., 2003]: starting from an arbitrary $\boldsymbol{\alpha} \in \mathcal{A}_d^D$, we iteratively update it until convergence. The details are provided in Algorithm 1 (Appendix D.4).

**Theorem 10.** *Fix $\epsilon > 0$. After $I = \widetilde{\mathcal{O}}(\epsilon^{-2}\vartheta_{D,2}^{(n)^2}(D - d))$ iterations, Algorithm 1 outputs $\boldsymbol{\alpha}^{(I)} \in \mathcal{A}_d^D$ such that $J(\boldsymbol{\alpha}^{(I)}) \leq \inf_{\boldsymbol{\alpha} \in \mathcal{A}_d^D} J(\boldsymbol{\alpha}) + \epsilon$.*

By definition of $J$, this result guarantees that $\boldsymbol{\alpha}^{(I)}$ is an approximate minimizer of the empirical Lebesgue constant $\Lambda_{\boldsymbol{\alpha}, \mu_n}$. To finally answer **Q2**, we define the BWR (*Best Weighted Regularizer*) estimator by plugging $\boldsymbol{\alpha}^{(I)}$ into Eq. (8):

$$\hat{\theta}_{n,\text{BWR}} := R_n^{-1} I_{\boldsymbol{\alpha}^{(I)}} \frac{1}{n} \sum_{t=1}^{n} \widehat{\boldsymbol{\varphi}}_D(x_t) y_t. \tag{9}$$

**Theorem 11.** *Let Assumption 1 hold and fix $\delta > 0$. Then, with probability $1 - \delta$,*

$$\mathcal{E}_\infty(\hat{\theta}_{n,BWR}) \leq (1 + \Lambda_\mu^{Oracle})\mathcal{E}_\infty(f) + \widetilde{\mathcal{O}}\left( \frac{\vartheta_{D,2}\sqrt{D \log(|\mathcal{X}|/\delta)}}{\sqrt{n}} + \frac{\vartheta_{D,2}^2 \log(|\mathcal{X}|/\delta)}{n} \right).$$

This oracle inequality affirmatively answers **Q2**: our estimator is asymptotically able to compete with the Oracle Lebesgue constant.

## 5    Case study: polynomial basis

The method introduced in Section 4 aims to reduce error amplification by explicitly controlling the Lebesgue constant. While broadly applicable, its impact is best illustrated in settings where standard estimators suffer from poor uniform behavior. A canonical example is polynomial regression on a compact interval, where the feature map $\boldsymbol{\varphi}_d$ consists of the first $d$ monomials, $\{1, x, x^2, \ldots, x^{d-1}\}$.

Consider the standard setting where $\mathcal{X} = [-1, 1]$ and the data-generating distribution $\mu$ is uniform on this interval. Even in this favorable scenario, the Lebesgue constant for the polynomial basis grows linearly with the degree, $\Lambda_{d,\mu} \approx d$. Consequently, the worst-case uniform error for OLS scales as $\mathcal{O}(d \cdot \mathcal{E}_\infty(f))$, meaning small misspecification errors can be amplified into large prediction errors.

In contrast, the BWR estimator augments the feature space—for instance, by doubling the degree to $D = 2d$—and optimizes the attenuation vector $\boldsymbol{\alpha}$ to minimize the empirical Lebesgue constant. This yields a projection operator that preserves the original degree-$d$ polynomials exactly while using the

extra degrees of freedom to stabilize the approximation. Theoretically, this reduces the amplification factor from $\mathcal{O}(d)$ to $\mathcal{O}(1)$, as the following theorem shows:

**Theorem 12.** *Let $\mu$ be the uniform distribution on $[-1, 1]$. There exists a constant $C > 0$ independent of $d$ such that, if we choose $D = 2d$ and $\boldsymbol{\varphi}_D(x) = [1, \ldots x^{2d-1}]$ as the augmented feature map for the target space spanned by $\boldsymbol{\varphi}_d(x) = [1, \ldots, x^{d-1}]^\top$, we have $\Lambda_\mu^{Oracle} \leq C$.*

This theoretical improvement translates directly to empirical performance, as shown in Fig. 1. While OLS exhibits characteristic large oscillations near the boundaries (a manifestation of the classical Runge phenomenon), BWR remains stable across the entire domain. By effectively controlling the Lebesgue constant, BWR achieves a significantly smaller uniform error despite using the same base features for the final representation.

The above simulations visually demonstrate how the amplification factor is exacerbated by increasing $d$. We complement them with an asymptotic result showing just how severe this factor can be, even for target functions where the approximation error $\mathcal{E}_\infty(f) \to 0$ as $d \to \infty$. In fact, there exist a bounded function that can be uniformly approximated by polynomials, yet for which the OLS estimator diverges with a uniform error roughly of order $\Omega(d)$:

**Proposition 13.** *Fix $\gamma > 0$. Let $\hat{\theta}_n$ be the OLS estimator, and $\hat{\theta}_{n,BWR}$ be our estimator defined in equation (9). There exists a function $f : [-1, 1] \to \mathbb{R}$ such that, $\mathcal{E}_\infty(f) \to 0$ as $d \to \infty$, and under Assumption 1 with $\mu = \mathcal{U}([-1, 1])$, the following hold with probability one:*

$$\lim_{d \to \infty} \lim_{n \to \infty} \|f(\cdot) - \boldsymbol{\varphi}_d(\cdot)^\top \hat{\theta}_{n,BWR}\|_\infty = 0 \qquad \text{while} \qquad \lim_{n \to \infty} \|f(\cdot) - \boldsymbol{\varphi}_d(\cdot)^\top \hat{\theta}_n\|_\infty \gtrsim d^{1-\gamma}.$$

## 6 Related works

The problem we address, while motivated by the goal of designing principled algorithms for bandits and reinforcement learning, has roots in several fields, including mathematical analysis, econometrics, and approximation theory. We provide a brief overview here, with an extended discussion in Appendix A.

In *mathematical analysis* the problem of projecting onto a linear subspace of $L^\infty(\mathcal{X})$ in a way that minimizes the uniform error have long been a central topic. Classical results on orthogonal polynomials Szegő [1939] and Fourier series Katznelson [2004] share this goal. More recently, Kobos and Lewicki [2024] proposed an approach for general feature maps. In *econometrics*, a related line of research studies pointwise estimators based on least-squares from noisy samples [Newey, 1997, Belloni et al., 2015, Li and Liao, 2020], which can be naturally adapted to yield uniform convergence guarantees. Most recently, this problem has resurfaced in *bandits and reinforcement learning* under the name misspecified linear function approximation [Du et al., 2020, Lattimore et al., 2020, Maran et al., 2024, Dong and Yang, 2023, Amortila et al., 2024].

The specific regularization technique we propose in Section 4 is inspired by classical methods for regularizing Fourier series [de la Vallée Poussin, 1918, De La Vallée Poussin et al., 1919]. Variants of this technique remain an active topic of study in numerical mathematics today [Németh, 2016, Themistoclakis and Van Barel, 2017, Occorsio and Themistoclakis, 2025].

## 7 Conclusion

We investigated the problem of uniform error control in misspecified linear regression under the random design setting. Our key insight is that the amplification of $\mathcal{E}_\infty(f)$ by least-squares methods is governed by the Lebesgue constant, a fundamental concept from approximation theory. We showed that this amplification is tight and intrinsic to the geometry of $L^2$-projection, thereby exposing a fundamental limitation of ordinary and ridge least-squares methods, even in the infinite data regime.

To overcome this limitation, we introduced a novel regularization framework based on weighted ridge regression over extended feature sets, which preserves the approximation power of the base features while using the auxiliary features to stabilize the projection operator. We proved that this approach allows us to, asymptotically for $n \to \infty$, compete with the best possible (oracle) projection in terms of uniform error, and we proposed an efficient algorithm for learning such weights from data. In the canonical case of polynomial features, we demonstrated a dramatic improvement: from $\Omega(d)$ amplification with OLS to the optimal $\mathcal{O}(1)$ with our method.

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

# A    Related Works

**Classical approximation theory** The idea of approximating a class of functions with a family of vector spaces in a uniform sense has always been an important topic in mathematical analysis. On the more general level, this theory takes the name of *Kolmogorov's n-width* (Kolmogoroff [1936]; see Lorentz [1966] and Pinkus [2012] for a more modern formalization). The idea, central to this paper, of finding a *linear* operator that well approximates the *non-linear* $L^\infty$ projection operator has also been the main topic of multiple line of research. In particular, many result about orthogonal polynomials Szegő [1939] or Fourier Series Katznelson [2004] approximation have this goal. More recently, Kobos and Lewicki [2024] studied the problem for general feature map, investigating the class of linear operators that achieve the lower bound.

**Asymptotic pointwise and uniform convergence of LS series in the econometric literature** In the econometric literature, the series least squares (LS) estimators have been analyzed primarily through an asymptotic lens: with the sample size $n \to +\infty$ and the basis dimension $d \to +\infty$, one studies asymptotic Gaussianity of the estimator of the function in each single point. Newey [1997] provided seminal results for this literature, which were then improved by Belloni et al. [2015], the first to use the Lebesgue constant in this field, and by Li and Liao [2020], who generalize the result to time series data. All these contributions, however, remain *asymptotic*: they provide limiting distributions or rates without explicit high–probability bounds, and—crucially—they do not propose algorithmic modifications capable of *reducing* the amplification factor induced by the Lebesgue constant.

**Uniform bounds for linear regression in the context of Online Learning** As anticipated in the introduction, the problem of getting $L^\infty$ bounds for regression over a domain naturally arises in the context of Online Learning with linear function approximation; bandits and RL in particular. Du et al. [2020] established the first $\sqrt{d}$ amplification lower bound in some specific cases, which was then refined by Lattimore et al. [2020], who also derives the corresponding an upper bound of $\sqrt{d}$, using an optimal design argument. In fact, it can be proved that the factor $\sqrt{d}$ is precisely the maximal Lebesgue constant of any feature map for $\mu$ that is the optimal design. These lower bound hold for a worst-case feature map, but *allowing the learner to choose the data distribution*. Following these works, many papers tried to understand how this amplification factor could be reduced. Maran et al. [2024] shows how to remove it in case of a locally linear feature map; Dong and Yang [2023] improves the $\sqrt{d}$ amplification in case of sparsity. Perhaps, the most similar paper to our one is Amortila et al. [2024], which proposes a method to mitigate the effect of misspecification w.r.t. the least-squares fitting. Still, the latter focuses on a different objective, i.e. the error under covariate shift (measuring the MSE under a distribution $\nu \neq \mu$), and scales with the density ratio $\nu(\cdot)/\mu(\cdot)$. Generalizing to the uniform error would mean to take $\nu(\cdot)$ as a Dirac's delta, which would make this bound vacuous.

**De la Valleè Poussin approach** The to reduce the Lebesgue constant by adding auxiliary features is rooted in a concept that dates back in the history of mathematics to Baron de la Vallée Poussin [de la Vallée Poussin, 1918, De La Vallée Poussin et al., 1919]. The technique he invented is still studied today in numerical mathematics [Németh, 2016, Themistoclakis and Van Barel, 2017, Occorsio and Themistoclakis, 2025].

**Finite-sample bounds for ridge regression** Hsu et al. [2014] gives finite-sample bounds for ridge regression under random design. The results, when translated into our setting, bound the error between $f_{\hat{\theta}_n}$ and $\bar{f}$ where $\bar{f} := g \circ \varphi$ and the bound is expressed in terms of $\bar{f} - \Pi_{\mu,d}f$. Here for $u \in \mathbb{R}^d$, $g(u) = \int f(x)\mu(dx|u)$ where $\mu(dx|u)$ is the disintegration of $\mu$ with respect to the push-forward of $\mu$ under $\varphi$. In particular, for $S \subset \mathcal{X}$, $u \in \mathbb{R}^d$, $\mu(S|u) = \int \mathbb{I}(x \in S, \varphi(x) = u)\mu(dx)$. In the special case when $\varphi$ is injective, $\bar{f} = f$. Just like in the result that can be extracted from the work of Lattimore et al. [2020], the bounds in this work depend on $\vartheta_{d,2}$ (or $\vartheta_{d,2}^{(n)}$) and scale similarly.

In fact, papers like Lattimore et al. [2020] adopt the following way to bound the uniform error of least squares. Let $V_n = \sum_{t=1}^n \varphi(x_t)\varphi(x_t)^\top$ be the Gramian matrix and $\theta_\star$ be the vector realizing the $L^\infty$ projection, so that $\varphi(x)^\top \theta_\star = \Pi_\infty f(x)$. Then each $y_t$ takes the form $\varphi(x_t)^\top \theta_\star + \varepsilon(x_t) + \eta_t$,

meaning

$$\boldsymbol{\varphi}(x)^\top(\theta_\star - \widehat{\theta}_n) = \boldsymbol{\varphi}(x)^\top \left( \theta_\star - V_n^{-1} \sum_{t=1}^n \boldsymbol{\varphi}(x_t)(\boldsymbol{\varphi}(x_t)^\top \theta_\star + \varepsilon(x_t)) \right) + \text{ stoc. part}$$

$$= \boldsymbol{\varphi}(x)^\top \left( -V_n^{-1} \sum_{t=1}^n \boldsymbol{\varphi}(x_t)\varepsilon(x_t) \right) + \text{ stoc. part}$$

$$= -\sum_{t=1}^n \varphi(x)^\top V_n^{-1} \boldsymbol{\varphi}(x_t)\varepsilon(x_t) + \text{ stoc. part}.$$

While the stochastic part is bounded as in our Theorems 2 and 3, the one containing the misspecification is treated as follows

$$|\boldsymbol{\varphi}(x)^\top(\theta_\star - \widehat{\theta}_n)| \leq \left| \sum_{t=1}^n \boldsymbol{\varphi}(x)^\top V_n^{-1} \boldsymbol{\varphi}(x_t)\varepsilon(x_t) \right|$$

$$\leq \sum_{t=1}^n \left| \boldsymbol{\varphi}(x)^\top V_n^{-1} \boldsymbol{\varphi}(x_t)\varepsilon(x_t) \right|$$

$$\leq \sum_{t=1}^n \left| \boldsymbol{\varphi}(x)^\top V_n^{-1} \boldsymbol{\varphi}(x_t) \right| \mathcal{E}_\infty(f)$$

$$\leq \sum_{t=1}^n \|\boldsymbol{\varphi}(x)\|_{V_n^{-1}} \|\boldsymbol{\varphi}(x_t)\|_{V_n^{-1}} \mathcal{E}_\infty(f)$$

$$= \|\boldsymbol{\varphi}(x)\|_{V_n^{-1}} \sum_{t=1}^n \|\boldsymbol{\varphi}(x_t)\|_{V_n^{-1}} \mathcal{E}_\infty(f)$$

$$\leq \|\boldsymbol{\varphi}(x)\|_{V_n^{-1}} \sqrt{n \sum_{t=1}^n \boldsymbol{\varphi}(x_t)^\top V_n^{-1} \boldsymbol{\varphi}(x_t)} \mathcal{E}_\infty(f)$$

$$\leq \|\boldsymbol{\varphi}(x)\|_{V_n^{-1}} \sqrt{n} \mathcal{E}_\infty(f)$$

$$= \|\boldsymbol{\varphi}(x)\|_{(V_n/n)^{-1}} \mathcal{E}_\infty(f)$$

$$= \|(V_n/n)^{-1/2} \boldsymbol{\varphi}(x)\|_2 \mathcal{E}_\infty(f).$$

By definition, $(V_n/n)^{-1/2}\boldsymbol{\varphi}(x) = \widehat{\boldsymbol{\varphi}}(x)$. Therefore, when making the supremum over $x \in \mathcal{X}$ we end up with $\vartheta_{d,2}^{(n)} \mathcal{E}_\infty(f)$. As we pointed out in the main paper and also noted by Lattimore et al. [2020], whatever the choice of $(x_t)_t$ and the feature map, $\vartheta_{d,2}^{(n)} \geq \sqrt{d}$. Therefore, this strategy is doomed to achieve sub-optimal guarantees, whenever $\Lambda_{d,\mu_n} < \mathcal{O}(\sqrt{d})$.

# B  General-interest results

We start from the usual Bernstein's inequality Boucheron et al. [2003], here written for variables that are bounded in $[-B, B]$ and in the "high probability" form.

**Theorem 14.** *Let $(x_t)_{t=1}^n$ be a sequence of zero-mean random variable bounded in $[-B, B]$. Let $\sigma^2 := \sum_{t=1}^n Var(X_t)$. Then, with probability at least $1 - \delta$*

$$\left| \sum_{t=1}^n X_t \right| \leq \sqrt{2\sigma^2 \log(2/\delta)} + \frac{2B}{3} \log(2/\delta).$$

**Lemma 2.** *Let $\overline{\boldsymbol{\varphi}}_d$ be an orthonormal feature map w.r.t. $\rho$.*

$$\mathbb{E}_{x\sim\rho} \left[ \overline{\boldsymbol{\varphi}}_d(x)\overline{\boldsymbol{\varphi}}_d(x)^\top \right] = I_d,$$

*where $I_d$ is the $d-$dimensional identity matrix.*

*Proof.* In this proof, let us denote with $\boldsymbol{e}_i$, for $i = 1, \ldots d$, the standard basis of $\mathbb{R}^d$. By definition of outer product between two vectors we get what follows.

$$\mathbb{E}_{x \sim \rho} \left[ \overline{\boldsymbol{\varphi}}_d(x) \overline{\boldsymbol{\varphi}}_d(x)^\top \right] = \mathbb{E}_{x \sim \rho} \left[ \sum_{i=1}^d \sum_{j=1}^d \overline{\varphi}_i(x) \overline{\varphi}_j(x) \boldsymbol{e}_i \boldsymbol{e}_j^\top \right]$$

$$= \sum_{i=1}^d \sum_{j=1}^d \mathbb{E}_{x \sim \rho} \left[ \overline{\varphi}_i(x) \overline{\varphi}_j(x) \right] \boldsymbol{e}_i \boldsymbol{e}_j^\top$$

$$= \sum_{i=1}^d \sum_{j=1}^d \delta_{ij} \boldsymbol{e}_i \boldsymbol{e}_j^\top = I_d.$$

This completes the proof. $\qquad\square$

**Lemma 3.** *Let $\{v_t\}_{t=1}^k$ be a sequence of independent $d-$dimensional random vectors such that*

$$\mathbb{E}[v_t v_t^\top] = \sigma I_d \qquad \|v_t\|_2^2 \leq B.$$

*Let $V := \sum_{t=1}^k v_t v_t^\top$. Then,*

1. *W.p. at least $1 - \delta$*

$$\lambda_{\min}(V) \geq \left( 1 - \sqrt{\frac{5B \log(d/\delta)}{k\sigma^2}} \right) k\sigma^2,$$

*if $\left( 1 - \sqrt{\frac{5B \log(d/\delta)}{k\sigma^2}} \right) \leq 1/2.$*

2. *W.p. at least $1 - \delta$*

$$\lambda_{\max}(V) \leq \left( 1 + \sqrt{\frac{2B \log(d/\delta)}{k\sigma^2}} \right) k\sigma^2,$$

*if $\left( 1 + \sqrt{\frac{2B \log(d/\delta)}{k\sigma^2}} \right) \leq 1.$*

*Proof.* Note that, as $\lambda_{\max}(v_t v_t^\top) = \|v_t\|_s^2 \leq B$, we can then apply Theorem 5.1.1 from Tropp et al. [2015] taking

$$\mu_{\min} = \mu_{\max} = k\sigma^2 \qquad L = B,$$

which ensures that

$$\forall \varepsilon \in (0, 1), \ \mathbb{P} \left( \lambda_{\min}(V) \leq (1 - \varepsilon) k\sigma^2 \right) \leq d \left( \frac{e^{-\varepsilon}}{(1 - \varepsilon)^{1-\varepsilon}} \right)^{k\sigma^2/B},$$

while

$$\forall \varepsilon > 0, \ \mathbb{P} \left( \lambda_{\max}(V) \geq (1 + \varepsilon) k\sigma^2 \right) \leq d \left( \frac{e^{\varepsilon}}{(1 + \varepsilon)^{1+\varepsilon}} \right)^{k\sigma^2/B}.$$

The thesis is going to follow by just simplifying the previous expressions. We recall from elementary Taylor expansions that

$$\varepsilon < 0.5 \implies -\varepsilon - 4\varepsilon^2/5 \leq \log(1 - \varepsilon) \leq -\varepsilon - \frac{\varepsilon^2}{2}.$$

and

$$\varepsilon < 1 \implies \varepsilon - \frac{\varepsilon^2}{2} \leq \log(1 + \varepsilon) \leq \varepsilon - \frac{\varepsilon^2}{4}.$$

Therefore, we have, for $\varepsilon < 0.5$

$$
\begin{aligned}
\frac{e^{-\varepsilon}}{(1 - \varepsilon)^{1-\varepsilon}} &= \exp(-\varepsilon - (1 - \varepsilon)\log(1 - \varepsilon)) \\
&\leq \exp\left(-\varepsilon - (1 - \varepsilon)(-\varepsilon - 4\varepsilon^2/5)\right) \\
&= \exp\left(-\varepsilon + \varepsilon - \varepsilon^2/5 + \mathcal{O}(\varepsilon^3)\right) \approx e^{-\varepsilon^2/5}.
\end{aligned}
$$

On the other side, for $\varepsilon \leq 1$,

$$
\begin{aligned}
\frac{e^{\varepsilon}}{(1 + \varepsilon)^{1+\varepsilon}} &= \exp(\varepsilon - (1 + \varepsilon)\log(1 + \varepsilon)) \\
&\leq \exp(\varepsilon - (1 + \varepsilon)(\varepsilon - \varepsilon^2/2)) \\
&= \exp(-\varepsilon^2/2 + \mathcal{O}(\varepsilon^3)) \approx e^{-\varepsilon^2/2}.
\end{aligned}
$$

This tells us that

$$\forall \varepsilon \in (0, 1/2), \ \mathbb{P}\left(\lambda_{\min}(V) \leq (1 - \varepsilon)k\sigma^2\right) \leq d e^{-k\sigma^2\varepsilon^2/(5B)},$$

and

$$\forall \varepsilon \in (0, 1), \ \mathbb{P}\left(\lambda_{\max}(V) \geq (1 + \varepsilon)k\sigma^2\right) \leq d e^{-k\sigma^2\varepsilon^2/(2B)}.$$

We can reformulate the previous results in the high-probability notation. Indeed, taking $\delta = d e^{-k\sigma^2\varepsilon^2/(5B)}$, we get

$$\varepsilon = \sqrt{\frac{5B \log(d/\delta)}{k\sigma^2}},$$

which entails that

$$\sqrt{\frac{5B \log(d/\delta)}{k\sigma^2}} \leq 1/2 \implies \mathbb{P}\left(\lambda_{\min}(V) \leq \left(1 - \sqrt{\frac{5B \log(d/\delta)}{k\sigma^2}}\right)k\sigma^2\right) \leq \delta.$$

Doing the same for the other result, we get

$$\sqrt{\frac{2B \log(d/\delta)}{k\sigma^2}} \leq 1 \implies \mathbb{P}\left(\lambda_{\max}(V) \geq \left(1 + \sqrt{\frac{2B \log(d/\delta)}{k\sigma^2}}\right)k\sigma^2\right) \leq \delta,$$

which completes the proof.

$\square$

**Proposition 15.** *The Lebesgue constant satisfies* $\Lambda_{d,\mu} = \sup_{x \in \mathcal{X}} \int_{\mathcal{X}} \left|\sum_{i=1}^{d} \overline{\varphi}_i(z)\overline{\varphi}_i(x)\right| \ d\mu(z).$

*Proof.* See Cheney [1966], chapter 4. $\square$

# C   Proofs from Section 3

## C.1   Lower bound for LS

Let $\Pi_\infty f = \arg\min_{g \in \mathcal{F}} \|f - g\|_\infty$ with ties broken arbitrarily. Theorem 1.1 of Chapter 3 in the book of DeVore and Lorentz [1993] guarantees that at least one minimizer exists. (As discussed there, uniqueness may or may not hold.)

**Lemma 4.** *We have*

$$\sup_{f \in \mathcal{F}} \frac{\|\Pi_{d,\mu} f - f\|_\infty}{\mathcal{E}_\infty(f)} \geq \Lambda_{d,\mu} - 1\,.$$

*Proof.* By definition of Lebesgue constant, for every $\varepsilon > 0$ there is a function $g$ such that

$$\|\Pi_{d,\mu} g\|_\infty \geq (\Lambda_{d,\mu} - \varepsilon)\|g\|_\infty.$$

Take $f = \Pi_\infty g - g$. We will use twice that for any $h \in \mathcal{F}$, $\|h\|_\infty = \|0 - h\|_\infty \geq \inf_{u \in \mathcal{F}} \|u - h\|_\infty = \|\Pi_\infty h - h\|_\infty$. Now,

$$
\begin{aligned}
\|\Pi_{d,\mu} f - f\|_\infty &= \|\Pi_{d,\mu}(\Pi_\infty g - g) - \Pi_\infty g + g\|_\infty \\
&= \|\Pi_\infty g - \Pi_{d,\mu} g - \Pi_\infty g + g\|_\infty \\
&= \| - \Pi_{d,\mu} g + g\|_\infty \\
&\geq \|\Pi_{d,\mu} g\|_\infty - \|g\|_\infty \\
&\geq (\Lambda_{d,\mu} - 1 - \varepsilon)\|g\|_\infty \\
&\geq (\Lambda_{d,\mu} - 1 - \varepsilon)\|\Pi_\infty g - g\|_\infty \\
&= (\Lambda_{d,\mu} - 1 - \varepsilon)\|f\|_\infty \\
&\geq (\Lambda_{d,\mu} - 1 - \varepsilon)\|\Pi_\infty f - f\|_\infty.
\end{aligned}
$$

The result follows by letting $\varepsilon \to 0$.  $\square$

**Theorem 1.** *For any $\varepsilon > 0$ and any subspace $\mathcal{F}$ there exist $f \in L^\infty(\mathcal{X})$ such that*

$$\|f - \Pi_{d,\mu} f\|_\infty \geq (\Lambda_{d,\mu} - 1 - \varepsilon)\mathcal{E}_\infty(f)\,.$$

*Proof.* The result is immediate from Lemma 4.  $\square$

For the next result we abbreviate $\|\cdot\|_{L^2(\mu) \to L^\infty(\mathcal{X})}$ by $\|\cdot\|_{2 \to \infty}$.

**Proposition 16.**

$$\vartheta_{d,2} = \sup_{x \in \mathcal{X}} \|\overline{\varphi}(x)\|_2 = \|\Pi_{d,\mu}\|_{2 \to \infty}$$

*Proof.* The following equalities hold:

$$
\begin{aligned}
\|\Pi_{d,\mu}\|_{2 \to \infty} &= \sup_{\|f\|_2 = 1} \|\Pi_{d,\mu} f\|_\infty \\
&= \sup_{\|f\|_2 = 1} \left\| \sum_{i=1}^d \overline{\varphi}_i(\cdot)\langle f, \varphi_i\rangle \right\|_\infty \\
&= \sup_{\|f\|_2 = 1} \sup_{x \in \mathcal{X}} \left| \sum_{i=1}^d \overline{\varphi}_i(x)\langle f, \varphi_i\rangle \right| \\
&= \sup_{x \in \mathcal{X}} \sup_{\|f\|_2 = 1} \left| \sum_{i=1}^d \overline{\varphi}_i(x)\langle f, \varphi_i\rangle \right| \\
&= \sup_{x \in \mathcal{X}} \sup_{v \in \mathbb{R}^d, \|v\|_2 = 1} \left| \sum_{i=1}^d \overline{\varphi}_i(x)v_i \right| \\
&= \sup_{x \in \mathcal{X}} \|\overline{\varphi}(x)\|_2 = \vartheta_{d,2}.
\end{aligned}
$$

In particular, the first is by definition of induced $2 \to \infty$-norm, the second one by definition of projection operator, and the third by definition of infinity norm. The fourth passage follows exchanging the two supremum, while the fifth from Parseval's theorem and the sixth one by duality of the two-norm (i.e. for any $\boldsymbol{w}$, $\boldsymbol{w} = \sup_{\|\boldsymbol{v}\|_2=1} \langle \boldsymbol{v}, \boldsymbol{w} \rangle$). $\qquad \square$

## C.2 Towards the proof of Theorem 2

**Lemma 5.** *Fix $\delta > 0$, and $n \geq 20\vartheta_{d,2}^2 \log(d/\delta)$. Let*

$$V_n = \sum_{t=1}^n \overline{\boldsymbol{\varphi}}(x_t)\overline{\boldsymbol{\varphi}}(x_t)^\top.$$

*Then, $\lambda_{\min}(V_n) \geq n/2$.*

*Proof.* The matrices we are summing correspond to $\overline{\boldsymbol{\varphi}}(x_t)\overline{\boldsymbol{\varphi}}(x_t)^\top$ each one being semi-positive definite with the biggest eigenvalue bounded by $\vartheta_{d,2}^2$ almost surely (indeed, $v^\top \overline{\boldsymbol{\varphi}}(x_t)\overline{\boldsymbol{\varphi}}(x_t)^\top v$ is maximized for $v$ parallel to $\overline{\boldsymbol{\varphi}}(x_t)$ and produces $\|\overline{\boldsymbol{\varphi}}(x_t)\|_2^2$). Moreover, as we have seen in Lemma 2,

$$\mathbb{E}\left[ \sum_{t=1}^n \overline{\boldsymbol{\varphi}}(x_t)\overline{\boldsymbol{\varphi}}(x_t)^\top \right] = \sum_{t=1}^n \mathbb{E}\left[ \overline{\boldsymbol{\varphi}}(x_t)\overline{\boldsymbol{\varphi}}(x_t)^\top \right] = nI_d.$$

These two ingredients allow us to apply Lemma 3 part one, which ensures that with probability at least $1 - \delta$

$$\lambda_{\min}(V_n) \geq \left(1 - \sqrt{\frac{5\vartheta_{d,2}^2 \log(d/\delta)}{n}}\right) n,$$

if $\left(1 - \sqrt{\frac{5\vartheta_{d,2}^2 \log(d/\delta)}{n}}\right) \leq 1/2$. Therefore, taking $n \geq 20\vartheta_{d,2}^2 \log(d/\delta)$, we get $\lambda_{\min}(V_n) \geq n/2$, which completes the proof.

$\qquad \square$

**Lemma 6.** *Let $\zeta(\cdot) := f(\cdot) - \Pi_{d,\mu}f(\cdot)$. With probability at least $1 - \delta$,*

$$\left| \overline{\boldsymbol{\varphi}}(z)^\top V_n^{-1} \sum_{t=1}^n \overline{\boldsymbol{\varphi}}(x_t)\zeta(x_t) \right| \leq \frac{2\Lambda_{d,\mu}\mathcal{E}_\infty(f)\vartheta_{d,2}}{\sqrt{n}} \sqrt{\log(1/\delta)},$$

*plus a lower-order term depending on $n^{-1}$ which takes the form of $\widetilde{\mathcal{O}}\left(n^{-1}d^{1/2}\vartheta_{d,2}^2 \Lambda_{d,\mu}\mathcal{E}_\infty(f) + n^{-3/2}d\vartheta_{d,2}^3 \Lambda_{d,\mu}\mathcal{E}_\infty(f)\right)$.*

*Proof.* We start rearranging the equation as follows

$$\left| \overline{\boldsymbol{\varphi}}(z)^\top V_n^{-1} \sum_{t=1}^n \overline{\boldsymbol{\varphi}}(x_t)\zeta(x_t) \right| = \left| \overline{\boldsymbol{\varphi}}(z)^\top \left(\frac{1}{n}V_n\right)^{-1} \frac{1}{n}\sum_{t=1}^n \overline{\boldsymbol{\varphi}}(x_t)\zeta(x_t) \right|$$

$$= \left| \overline{\boldsymbol{\varphi}}(z)^\top (I_d + \Delta_n) \frac{1}{n}\sum_{t=1}^n \overline{\boldsymbol{\varphi}}(x_t)\zeta(x_t) \right|$$

$$\leq \left| \overline{\boldsymbol{\varphi}}(z)^\top \frac{1}{n}\sum_{t=1}^n \overline{\boldsymbol{\varphi}}(x_t)\zeta(x_t) \right|$$

$$+ \left| \overline{\boldsymbol{\varphi}}(z)^\top \Delta_n \frac{1}{n}\sum_{t=1}^n \boldsymbol{\varphi}_d(x_t)\zeta(x_t) \right|.$$

For $\Delta_n := (V_n/n)^{-1} - I_d$. To bound both parts, we start by giving a result for $\frac{1}{n}\sum_{t=1}^n v^\top \overline{\boldsymbol{\varphi}}(x_t)\zeta(x_t)$ that holds for one fixed $v \in \mathbb{R}^d$. Indeed,

1. Every random variable $v^\top \overline{\varphi}(x_t)\zeta(x_t)$ is bounded by $\|v\|_2 \vartheta_{d,2} \Lambda_{d,\mu} \mathcal{E}_\infty(f)$ a.s.

2. The variance of the same random variable is
$$
\begin{aligned}
\mathbb{E}_{x\sim\rho}[(v^\top \overline{\varphi}(x)\zeta(x))^2] &= \mathbb{E}_{x\sim\rho}[\zeta(x)^2 v^\top \overline{\varphi}(x)^\top \overline{\varphi}(x)v] \\
&\leq (\Lambda_{d,\mu}\mathcal{E}_\infty(f))^2 v^\top \mathbb{E}_{x\sim\rho}[\overline{\varphi}(x)^\top \overline{\varphi}(x)]v \\
&= (\Lambda_{d,\mu}\mathcal{E}_\infty(f))^2 v^\top I_d v \\
&= (\Lambda_{d,\mu}\mathcal{E}_\infty(f))^2 \|v\|_2^2,
\end{aligned}
$$
the main step following from Lemma 2.

So by Bernstein's inequality (Theorem 14),
$$
\frac{1}{n}\sum_{t=1}^n v^\top \overline{\varphi}(x_t)\zeta(x_t) \leq \frac{2\Lambda_{d,\mu}\mathcal{E}_\infty(f)\|v\|_2}{\sqrt{n}}\sqrt{\log(1/\delta)} + \frac{2\|v\|_2 \vartheta_{d,2}\Lambda_{d,\mu}\mathcal{E}_\infty(f)}{3n}\log(1/\delta). \quad (10)
$$

We can use the previous equation to bound both parts. For the first, we just take $v = \overline{\varphi}(z)$, which respects $\|v\|_2 \leq \vartheta_{d,2}$, in Eq. (10) and get
$$
\left| \overline{\varphi}(z)^\top \frac{1}{n}\sum_{t=1}^n \overline{\varphi}(x_t)\zeta(x_t) \right| \leq \frac{2\Lambda_{d,\mu}\mathcal{E}_\infty(f)\vartheta_{d,2}}{\sqrt{n}}\sqrt{\log(1/\delta)} + \frac{2\vartheta_{d,2}\Lambda_{d,\mu}\mathcal{E}_\infty(f)}{3n}\log(1/\delta).
$$

Let us now focus on the second part. Indeed,
$$
\left| \overline{\varphi}(z)^\top \Delta_n \frac{1}{n}\sum_{t=1}^n \overline{\varphi}(x_t)\zeta(x_t) \right| \leq \vartheta_{d,2}\|\Delta_n\|_2 \left\| \frac{1}{n}\sum_{t=1}^n \overline{\varphi}(x_t)\zeta(x_t) \right\|_2
$$

Now, using Lemma 3 as done in the proof of Lemma 5, we have
$$
\|\Delta_n\|_2 \leq \vartheta_{d,2}\sqrt{\frac{5\log(d/\delta)}{n}},
$$
while for the last part we can write
$$
\begin{aligned}
\left\| \frac{1}{n}\sum_{t=1}^n \overline{\varphi}(x_t)\zeta(x_t) \right\|_2 &= \sup_{\|v\|_2=1} \frac{1}{n}\sum_{t=1}^n v^\top \overline{\varphi}(x_t)\zeta(x_t) \\
&\leq \sup_{\|v\|_2 \in B_d^{1/n}} \frac{1}{n}\sum_{t=1}^n v^\top \overline{\varphi}(x_t)\zeta(x_t) + \frac{\vartheta_{d,2}\Lambda_{d,\mu}\mathcal{E}_\infty(f)}{n},
\end{aligned}
$$

where $B_d^{1/n}$ is a $1/n$ covering of the set of vectors such that $\|v\|_2 = 1$. It is well-known that we can choose $B_d^{1/n}$ so that $|B_d^{1/n}| \approx n^{-d}$, so that, making a union bound together with Eq. (10), we get

$$
\left\| \frac{1}{n}\sum_{t=1}^n \overline{\varphi}(x_t)\zeta(x_t) \right\|_2 \leq \frac{2\Lambda_{d,\mu}\mathcal{E}_\infty(f)}{\sqrt{n}}\sqrt{d\log(1/\delta)} + \frac{2\vartheta_{d,2}d\Lambda_{d,\mu}\mathcal{E}_\infty(f)}{3n}\log(1/\delta) + \frac{\Lambda_{d,\mu}\mathcal{E}_\infty(f)}{n}.
$$

As a consequence,

$$
\begin{aligned}
&\left| \overline{\varphi}(z)^\top \Delta_n \frac{1}{n}\sum_{t=1}^n \overline{\varphi}(x_t)\zeta(x_t) \right| \\
&\leq \vartheta_{d,2}^2 \sqrt{\frac{5\log(d/\delta)}{n}}\left( \frac{2\Lambda_{d,\mu}\mathcal{E}_\infty(f)}{\sqrt{n}}\sqrt{d\log(1/\delta)} + \frac{2\vartheta_{d,2}d\Lambda_{d,\mu}\mathcal{E}_\infty(f)}{3n}\log(1/\delta) + \frac{\Lambda_{d,\mu}\mathcal{E}_\infty(f)}{n} \right) \\
&= \widetilde{\mathcal{O}}\left( n^{-1}d^{1/2}\vartheta_{d,2}^2 \Lambda_{d,\mu}\mathcal{E}_\infty(f) + n^{-3/2}d\vartheta_{d,2}^3 \Lambda_{d,\mu}\mathcal{E}_\infty(f) \right).
\end{aligned}
$$

This completes the proof. $\qquad\square$

## C.3 Proof of Theorem 2

**Theorem 2.** *Let $\mathcal{X}$ be finite and Assumption 1 hold. Let $\mathcal{X}$ be finite and Assumption 1 hold. For any $\delta \in (0, 1/3]$ and any $n \geq 20\vartheta_{d,2}^2 \log(d/\delta)$, the OLS estimate $\hat{\theta}_n$ satisfies, with probability at least $1 - 3\delta$,*

$$\mathcal{E}_\infty(\hat{\theta}_n, f) \leq (1 + \Lambda_{d,\mu})\mathcal{E}_\infty(f) + 3(\sigma + \Lambda_{d,\mu}\mathcal{E}_\infty(f))\vartheta_{d,2}\sqrt{\frac{\log(|\mathcal{X}|/\delta)}{n}}$$
$$+ \frac{poly(d, \vartheta_{d,2}, \Lambda_{d,\mu}\mathcal{E}_\infty(f))}{n}.$$

*Proof.* In this proof, let $\zeta_{d,\mu}(\cdot) := f(\cdot) - \Pi_{d,\mu}f(\cdot)$ and $\eta_t = y_t - f(x_t)$. Moreover, we will call $\hat{\theta}_n$ the OLS estimator parametrized w.r.t. $\overline{\varphi}$, rather than $\varphi$. We will also call $\hat{f}_n(\cdot) = \overline{\varphi}(\cdot)^\top\hat{\theta}_n$ the corresponding estimated function (which does not change with the parameterization of the basis, as it only depends on $\mathcal{F}$).

We start making the following decomposition:

$$|\overline{\varphi}(x)^\top\hat{\theta}_n - f(x)| \leq |\overline{\varphi}(x)^\top\hat{\theta}_n - \Pi_{d,\mu}f(x)| + \|\Pi_{d,\mu}f - f\|_\infty$$
$$\leq |\overline{\varphi}(x)^\top\hat{\theta}_n - \Pi_{d,\mu}f(x)| + (1 + \Lambda_{d,\mu})\mathcal{E}_\infty(f).$$

To bound the first part, we let $\theta_\star$ be such that $\Pi_{d,\mu}f(\cdot) = \overline{\varphi}(\cdot)^\top\theta_\star$. By Assumption 1, the samples take the form $y_t = \overline{\varphi}(x_t)^\top\theta_\star + \zeta_{d,\mu}(x_t) + \eta_t$, where $(\eta_t)_{t=1}^n$ is a family of independent $\sigma-$subgaussian random variables. By definition, letting $V_n = \sum_{t=1}^n \overline{\varphi}(x_t)\overline{\varphi}(x_t)^\top$, the LS solution takes the form $\overline{\varphi}(x_t)^\top\hat{\theta}_n$, where

$$\hat{\theta}_n = V_n^{-1}\sum_{t=1}^n \overline{\varphi}(x_t)y_t$$
$$= V_n^{-1}\sum_{t=1}^n \overline{\varphi}(x_t)(\overline{\varphi}(x_t)^\top\theta_\star + \eta_t + \zeta_{d,\mu}(x_t))$$
$$= \theta_\star + V_n^{-1}\sum_{t=1}^n \overline{\varphi}(x_t)(\eta_t + \zeta_{d,\mu}(x_t)).$$

Therefore, we have

$$|\overline{\varphi}_d(x)^\top\hat{\theta}_n - \Pi_{d,\mu}f(x)| \leq \underbrace{\left|\overline{\varphi}(x)^\top V_n^{-1}\sum_{t=1}^n \overline{\varphi}(x_t)\eta_t\right|}_{(I)} + \underbrace{\left|\overline{\varphi}(x)^\top V_n^{-1}\sum_{t=1}^n \overline{\varphi}(x_t)\zeta_{d,\mu}(x_t)\right|}_{(II)}.$$

We are going to bound the two terms separately. First, let $E := \{\lambda_{\min}(V_n) \geq n/2\}$. From Lemma 5, under the assumptions of this theorem, we have $\mathbb{P}(E) \geq 1 - \delta$.

(I) Since $\eta_t$ are independent and $\sigma-$subgaussian conditionally to $(x_t)_{t=1}^n$ (Assumption 1), Lemma 5.4 and Theorem 5.3 from Lattimore and Szepesvári [2020] ensure that, with probability at least $1 - 2\delta$

$$\left|\overline{\varphi}(x)^\top V_n^{-1}\sum_{t=1}^n \overline{\varphi}(x_t)\eta_t\right| \leq \sqrt{2\sigma^2\sum_{t=1}^n \left(\overline{\varphi}(x)^\top V_n^{-1}\overline{\varphi}(x_t)\right)^2\log(1/\delta)}$$
$$= \sqrt{2\sigma^2\|\overline{\varphi}(x)\|_{V_n^{-1}}^2\log(1/\delta)}$$
$$= \sqrt{2\log(1/\delta)}\sigma\|\overline{\varphi}(x)\|_{V_n^{-1}}.$$

Moreover, if event $E$ holds,

$$\|\overline{\varphi}(x)\|_{V_n^{-1}} \leq \frac{2\|\overline{\varphi}(x)\|_2}{\sqrt{n}} \leq \frac{2\vartheta_{d,2}}{\sqrt{n}},$$

so that the full term is bounded by $\sqrt{8\log(1/\delta)}\sigma\vartheta_{d,2}n^{-1/2}$.

(II) This term is bounded by Lemma 6 which, with probability at least $1 - \delta$ gives

$$\left| \overline{\boldsymbol{\varphi}}(z)^\top V_n^{-1} \sum_{t=1}^n \overline{\boldsymbol{\varphi}}(x_t) \zeta_{d,\mu}(x_t) \right| \leq \frac{2\Lambda_{d,\mu} \mathcal{E}_\infty(f) \vartheta_{d,2}}{\sqrt{n}} \sqrt{\log(1/\delta)},$$

plus lower-order terms of the form $\frac{\text{poly}(d, \vartheta_{d,2}, \Lambda_{d,\mu} \mathcal{E}_\infty(f))}{n}$.

Note that, thanks to Lemma 5, event $E$ holds with probability $1 - \delta$ under the assumptions of this theorem. Moreover, imposing that both events in $(I)$ and $(II)$ verify, we get, with probability at least $1 - 3\delta$,

$$|\overline{\boldsymbol{\varphi}}(x)^\top \hat{\theta}_n - f(x)|_\infty \leq (1 + \Lambda_{d,\mu}) \mathcal{E}_\infty(f) + |\overline{\boldsymbol{\varphi}}(x)^\top \hat{\theta}_n - \Pi_{d,\mu} f(x)|$$
$$\leq (1 + \Lambda_{d,\mu}) \mathcal{E}_\infty(f) + \frac{3(\sigma + \Lambda_{d,\mu} \mathcal{E}_\infty(f)) \vartheta_{d,2}}{\sqrt{n}} \sqrt{\log(1/\delta)}$$

plus lower-order terms of the form $\frac{\text{poly}(d, \vartheta_{d,2}, \Lambda_{d,\mu} \mathcal{E}_\infty(f))}{n}$. This completes the proof. $\qquad \square$

## C.4 Bound scaling with the empirical Lebesgue constant

**Theorem 3.** *Let $\mathcal{X}$ be finite, Assumption 1 hold, $\hat{\theta}_n$ be the OLS estimate. Then, for any fixed $\delta > 0$, with probability at least $1 - \delta$,*

$$\mathcal{E}_\infty(\hat{\theta}_n) \leq (1 + \Lambda_{d,\mu_n}) \mathcal{E}_\infty(f) + \frac{\sigma \vartheta_{d,2}^{(n)} \sqrt{2 \log(2|\mathcal{X}|/\delta)}}{\sqrt{n}}.$$

*Proof.* In this proof, let $\zeta(\cdot) := f(\cdot) - \Pi_{d,\mu_n} f(\cdot)$ and $\eta_t = y_t - f(x_t)$. Moreover, we will call $\hat{\theta}_n$ the OLS estimator parametrized w.r.t. $\widehat{\boldsymbol{\varphi}}$, rather than $\boldsymbol{\varphi}$. We will also $\widehat{f}_n(\cdot) = \widehat{\boldsymbol{\varphi}}(\cdot)^\top \hat{\theta}_n$ the corresponding estimated function (which does not change with the parameterization of the basis, as it only depends on $\mathcal{F}$).

The following decomposition holds:

$$\|f(\cdot) - \widehat{f}_n(\cdot)\|_\infty \leq \|f(\cdot) - \Pi_{d,\mu_n} f(\cdot)\|_\infty + \|\Pi_{d,\mu_n} f(\cdot) - \widehat{f}_n(\cdot)\|_\infty$$
$$\leq (1 + \widehat{\Lambda}_{d,\mu}) \mathcal{E}_\infty(f) + \|\Pi_{d,\mu_n} f(\cdot) - \widehat{f}_n(\cdot)\|_\infty.$$

Now, we focus on the second term. As done in the previous proof of Theorem 2, we let $\theta_\star$ be such that $\Pi_{d,\mu_n} f(\cdot) = \widehat{\boldsymbol{\varphi}}(\cdot)^\top \theta_\star$ and $\zeta(\cdot) := f(\cdot) - \widehat{\boldsymbol{\varphi}}(\cdot)^\top \theta_\star$. In this way, our samples take the form $y_t = \widehat{\boldsymbol{\varphi}}(x_t)^\top \theta_\star + \zeta(x_t) + \eta_t$.

For any fixed $x \in \mathcal{X}$ we have

$$\widehat{f}_n(x) = \widehat{\boldsymbol{\varphi}}(x)^\top \hat{\theta}_n$$
$$= \widehat{\boldsymbol{\varphi}}(x)^\top \frac{1}{n} \sum_{t=1}^n \widehat{\boldsymbol{\varphi}}(x_t) y_t$$
$$= \widehat{\boldsymbol{\varphi}}(x)^\top \frac{1}{n} \sum_{t=1}^n \widehat{\boldsymbol{\varphi}}(x_t) (\widehat{\boldsymbol{\varphi}}(x_t)^\top \theta_\star + \zeta(x_t) + \eta_t)$$
$$= \widehat{\boldsymbol{\varphi}}(x)^\top \theta_\star + \underbrace{\widehat{\boldsymbol{\varphi}}(x)^\top \frac{1}{n} \sum_{t=1}^n \widehat{\boldsymbol{\varphi}}(x_t) \zeta(x_t)}_{(I)} + \underbrace{\widehat{\boldsymbol{\varphi}}(x)^\top \frac{1}{n} \sum_{t=1}^n \widehat{\boldsymbol{\varphi}}(x_t) \eta_t}_{(II)}.$$

Here, the last passage is due to the fact that, being $\widehat{\boldsymbol{\varphi}}(\cdot)$ orthogonal w.r.t. $\mu_n(\cdot)$, it follows $\frac{1}{n} \sum_{t=1}^n \widehat{\boldsymbol{\varphi}}(x_t) \widehat{\boldsymbol{\varphi}}(x_t)^\top = I_d$. Now, we analyze the two terms $(I)$ and $(II)$ separately.

$$(I) = \widehat{\boldsymbol{\varphi}}(x)^\top \frac{1}{n} \sum_{t=1}^n \widehat{\boldsymbol{\varphi}}(x_t)\zeta(x_t)$$

$$= \widehat{\boldsymbol{\varphi}}(x)^\top \int_{\mathcal{X}} \widehat{\boldsymbol{\varphi}}(z)\zeta(z)\, d\mu_n(z) = \widehat{\boldsymbol{\varphi}}(x)^\top \mathbf{0} = 0.$$

In fact, by definition of orthogonal projection, $\zeta(\cdot)$ is orthogonal in $L^2(\mu_n)$ to the span of $\widehat{\boldsymbol{\varphi}}(\cdot)$, so to each of its components in particular.

Let us look at the second term. Since $\eta_t$ are independent and $\sigma-$subgaussian conditionally on $(x_t)_{t=1}^n$, Lemma 5.4 and Theorem 5.3 from Lattimore and Szepesvári [2020] ensure that, with probability at least $1 - 2\delta$

$$\left| \widehat{\boldsymbol{\varphi}}(x)^\top n^{-1} \sum_{t=1}^n \widehat{\boldsymbol{\varphi}}(x_t)\eta_t \right| \leq \sqrt{2\sigma^2 n^{-1} \sum_{t=1}^n \left( \widehat{\boldsymbol{\varphi}}(x)^\top \widehat{\boldsymbol{\varphi}}(x_t) \right)^2 \log(1/\delta)}$$

$$= \sqrt{2\sigma^2 n^{-1} \|\widehat{\boldsymbol{\varphi}}(x)\|_2^2 \log(1/\delta)}$$

$$= \sqrt{2\log(1/\delta)}\sigma n^{-1/2} \|\widehat{\boldsymbol{\varphi}}(x)\|_2.$$

Where the second passage comes once again from the fact that $\frac{1}{n}\sum_{t=1}^n \widehat{\boldsymbol{\varphi}}(x_t)\widehat{\boldsymbol{\varphi}}(x_t)^\top = I_d$. This proves that $(II)$ is bounded by $\sqrt{2\log(1/\delta)}\sigma n^{-1/2}\widehat{\varphi}_{2,d}$. Making a union bound over $x \in \mathcal{X}$, this entails w.p. $1 - \delta$,

$$\sup_{x \in \mathcal{X}} \left| \widehat{\boldsymbol{\varphi}}(x)^\top n^{-1} \sum_{t=1}^n \widehat{\boldsymbol{\varphi}}(x_t)\eta_t \right| \leq \sqrt{2\log(|\mathcal{X}|/\delta)}\sigma n^{-1/2}\widehat{\varphi}_{2,d}.$$

We have proved that

$$\mathcal{E}_\infty(\hat{\theta}_n) = \|f(\cdot) - \widehat{f}_n(\cdot)\|_\infty$$
$$\leq (1 + \widehat{\Lambda}_{d,\mu})\mathcal{E}_\infty(f) + \|\Pi_{d,\mu_n}f(\cdot) - \widehat{f}_n(\cdot)\|_\infty$$
$$\leq (1 + \widehat{\Lambda}_{d,\mu})\mathcal{E}_\infty(f) + \sqrt{2\log(|\mathcal{X}|/\delta)}\sigma n^{-1/2}\widehat{\varphi}_{2,d}.$$

$\square$

### C.5 Proofs from Section 3.1

**Proposition 17.** *The Lebesgue constant is bounded by* $\Lambda_{d,\mu} \leq \vartheta_{d,2}$.

*Proof.* Let $f \in L^\infty(\mathcal{X})$ with $\|f\|_\infty = 1$. We have, for any $x \in \mathcal{X}$,

$$|\Pi_{d,\mu}f(x)| = \left| \sum_{i=1}^d \langle f, \overline{\varphi}_i \rangle \overline{\varphi}_i(x) \right|$$

$$\leq \sqrt{\sum_{i=1}^d \langle f, \overline{\varphi}_i \rangle^2 \sum_{i=1}^d \overline{\varphi}_i(x)^2}$$

$$\leq \sqrt{\|f\|_\mu^2 \|\overline{\varphi}_i(x)\|_2^2}$$

$$\leq \|f\|_\infty \sqrt{\|\overline{\varphi}_i(x)\|_2^2} \leq \vartheta_{d,2},$$

the last passage coming from the fact that as $\rho$ is a probability measure, $\|f\|_\mu \leq \|f\|_\infty$. The thesis follows taking the supremum on $f, x$. $\square$

**Proposition 18.** *Let $\varphi_d : \mathcal{X} \to \mathbb{R}^d$ be any feature map, and $\rho$ a probability measure. Then,*

$$\overline{\varphi}_2 \geq \sqrt{d}.$$

*Proof.* The key for this result is to note that, being $\rho$ a probability measure, $\vartheta_{d,2}^2 \geq \mathbb{E}_{x\sim\rho}\left[\|\overline{\varphi}_d(x)\|_2^2\right]$ (the supremum of a function upper bounds its integral on any probability measure). Then,

$$\begin{aligned}
\vartheta_{d,2} &\geq \sqrt{\mathbb{E}_{x\sim\rho}\left[\|\overline{\varphi}_d(x)\|_2^2\right]} \\
&= \sqrt{\mathbb{E}_{x\sim\rho}\left[\overline{\varphi}_d(x)^\top\overline{\varphi}_d(x)\right]} \\
&= \sqrt{\mathbb{E}_{x\sim\rho}\left[\mathrm{Tr}(\overline{\varphi}_d(x)^\top\overline{\varphi}_d(x))\right]} \\
&= \sqrt{\mathbb{E}_{x\sim\rho}\left[\mathrm{Tr}(\overline{\varphi}_d(x)\overline{\varphi}_d(x)^\top)\right]} \\
&= \sqrt{\mathrm{Tr}(\mathbb{E}_{x\sim\rho}\left[\overline{\varphi}_d(x)\overline{\varphi}_d(x)^\top\right])} \\
&\overset{*}{=} \sqrt{\mathrm{Tr}(I_d)} = \sqrt{d}.
\end{aligned}$$

Where the passage $(*)$ comes from Lemma 2. $\qquad\square$

**Proposition 19.** *Let $\mathcal{X} = [k]$ and $\varphi_i(j) = X_{ij}$, with all the $X_{ij}$ being independent bounded zero-mean unit variance random variables. Then, if $d = \mathcal{O}(\sqrt{k})$, the feature map $\varphi_d$, satisfies*

$$\Lambda_{d,\mu} = \mathcal{O}(\sqrt{d\log(k/\delta)})$$

*with probability at least $1 - \delta$. Moreover, $\mathbb{E}[\Lambda_{d,\mu}] \geq \Omega(\sqrt{d})$.*

*Proof.* By convenience, we call $\Phi \in \mathbb{R}^{k\times d}$ the matrix having, as columns, the features of $\varphi_d$. Precisely, the $i-$th column of $\Phi$ corresponds to $\varphi_i$. It is well-known that, in a finite dimensional space the orthogonal projection operator writes as

$$\Pi_{d,\mu} := \Phi(\Phi^\top\Phi)^{-1}\Phi^\top.$$

We call $\Phi_{m\cdot}$ the $m-$th row of $\Phi$ which, by assumption, is a random vector of independent entries bounded in $[-B, B]$ and with variance one. We have

$$\Phi^\top\Phi = \sum_{m=1}^k \Phi_{m\cdot}\Phi_{m\cdot}^\top, \qquad \mathbb{E}[\Phi_{m\cdot}\Phi_{m\cdot}^\top] = \sigma^2 I_d, \qquad \lambda_d(\Phi_{m\cdot}\Phi_{m\cdot}^\top) \leq dB^2.$$

At this point, we can apply Lemma 3, that ensures with probability $1 - 2\delta$, for $k$ sufficiently large,

$$\left(1 - \sqrt{\frac{5dB^2\log(d/\delta)}{k\sigma^2}}\right)k\sigma^2 \leq \lambda_{\min}(\Phi^\top\Phi) \leq \lambda_{\max}(\Phi^\top\Phi) \leq \left(1 + \sqrt{\frac{2dB^2\log(d/\delta)}{k\sigma^2}}\right)k\sigma^2.$$

Now, we can fix $\sigma = 1$ as in the assumption and rewrite the projection operator in the following form

$$\Pi_{d,\mu} := k^{-1}\Phi(k^{-1}\Phi^\top\Phi)^{-1}\Phi^\top = k^{-1}\Phi\Phi^\top + k^{-1}\Phi\Delta\Phi^\top,$$

where $\Delta$ has all the eigenvalues of magnitude less than $\sqrt{\frac{5dB^2\log(d/\delta)}{k\sigma^2}}$, by the previous result.

We now bound the infinity norm of the two terms separately. First,

$$\|k^{-1}\Phi\Phi^\top\|_\infty \overset{*}{=} \frac{1}{k}\max_{m=1,\dots k}\|(\Phi\Phi^\top)_{m\cdot}\|_1$$

$$= \max_{m=1,\dots k}\frac{1}{k}\sum_{n=1}^{k}\left|\sum_{i=1}^{d}\Phi_{mi}\Phi_{ni}\right|,$$

where $*$ holds since the infinity norm of a matrix corresponds to the maximum $1-$norm between its rows. Now, note that, as the rows are independent, each variable $\sum_{i=1}^{d}\Phi_{mi}\Phi_{ni}$, for $m\neq n$ is a sum of i.i.d. random variables such that

- $\Phi_{mi}\Phi_{ni}$ is bounded in $[-B^2, B^2]$ almost surely.
- The variance is

$$\mathbb{E}[(\Phi_{mi}\Phi_{ni})^2] = \mathbb{E}[\Phi_{mi}^2\Phi_{ni}^2] = \mathbb{E}[\Phi_{mi}^2]\mathbb{E}[\Phi_{ni}^2] = 1.$$

Therefore, Bernstein's inequality (14) ensures that, w.p. $1-\delta$

$$\left|\sum_{i=1}^{d}\Phi_{mi}\Phi_{ni}\right| \leq \sqrt{2d\log(2/\delta)} + \frac{2B^2}{3}\log(2/\delta).$$

Making a union bound over the $k^2 - k$ pairs $m\neq n$, we get, still with probability at least $1-\delta$,

$$\forall n\neq m \qquad \left|\sum_{i=1}^{d}\Phi_{mi}\Phi_{ni}\right| \leq \sqrt{4d\log(2k/\delta)} + \frac{4B^2}{3}\log(2k/\delta). \qquad (11)$$

At this point, we simply have, with probability $1-\delta$,

$$\|k^{-1}\Phi\Phi^\top\|_\infty = \max_{m=1,\dots k}\frac{1}{k}\sum_{n=1}^{k}\left|\sum_{i=1}^{d}\Phi_{mi}\Phi_{ni}\right|$$

$$\leq \frac{dB^2}{k} + \max_{m=1,\dots k}\frac{1}{k}\sum_{n=1,n\neq m}^{k}\left|\sum_{i=1}^{d}\Phi_{mi}\Phi_{ni}\right|$$

$$\overset{(11)}{\leq} \frac{dB^2}{k} + \max_{m=1,\dots k}\frac{1}{k}\sum_{n=1,n\neq m}^{k}\left(\sqrt{4d\log(2k/\delta)} + \frac{4B^2}{3}\log(2k/\delta)\right)$$

$$= \sqrt{4d\log(2k/\delta)} + \frac{4B^2}{3}\log(2k/\delta) + \frac{dB^2}{k}.$$

For the second term, we have

$$\|k^{-1}\Phi\Delta\Phi^\top\|_\infty \leq k^{-1}\max_{m=1,\dots k}\sum_{n=1}^{k}\left|\langle\Phi_{m\cdot}, (\Delta\Phi^\top)_{\cdot n}\rangle\right|$$

$$\leq k^{-1}\max_{m=1,\dots k}\sum_{n=1}^{k}\|\Phi_{m\cdot}\|_2\|(\Delta\Phi^\top)_{\cdot n}\|_2$$

$$\overset{*}{\leq} k^{-1}\max_{m=1,\dots k}\sum_{n=1}^{k}\frac{dB^2}{\sqrt{k}}$$

$$\leq \frac{dB^2}{\sqrt{k}},$$

where $*$ comes from the bound on the eigenvalues of $\Delta$. Putting everything together, we have proved that

$$\|\Pi_{d,\mu}\|_\infty \leq \sqrt{4d\log(2k/\delta)} + \frac{4B^2}{3}\log(2k/\delta) + \frac{dB^2}{k} + \frac{dB^2}{\sqrt{k}} = \sqrt{4d\log(2k/\delta)} + \mathcal{O}(d/\sqrt{k}).$$

To show that we cannot go much lower than this quantity, note that, even ignoring the contribution of $\Delta$ we have

$$\|\Pi_{d,\mu}\|_\infty \approx \|k^{-1}\Phi\Phi^\top\|_\infty = \max_{m=1,\dots k} \frac{1}{k} \sum_{n=1}^{k} \left| \sum_{i=1}^{d} \Phi_{mi}\Phi_{ni} \right|.$$

Therefore,

$$\begin{aligned}
\mathbb{E}[\|\Pi_{d,\mu}\|_\infty] &\approx \mathbb{E}\left[ \max_{m=1,\dots k} \frac{1}{k} \sum_{n=1}^{k} \left| \sum_{i=1}^{d} \Phi_{mi}\Phi_{ni} \right| \right] \\
&\geq \max_{m=1,\dots k} \frac{1}{k} \sum_{n=1}^{k} \mathbb{E}\left[ \left| \sum_{i=1}^{d} \Phi_{mi}\Phi_{ni} \right| \right] \\
&\geq \max_{m} \frac{1}{k} \sum_{n=1,n\neq m}^{k} \Omega(\sqrt{d}) = \Omega(\sqrt{d}).
\end{aligned}$$

The last passage comes from the fact that, for $n \neq m$, we have the expected value of the modulus a sum of $d$ independent random variables, which grows as $\sqrt{d}$. $\qquad\square$

**Proposition 20.** *Let $\mu, \nu$ be two discrete probability measures supported on a countable set $\mathcal{X}$ such that for all $x \in \mathcal{X}$, $0 \leq c \leq \frac{\mu(x)}{\nu(x)} \leq C$. Then, $\Lambda_{d,\mu} \leq \frac{C}{c}\Lambda_{d,\nu}$.*

*Proof.* The following identity holds for the Lebesgue constant

$$\begin{aligned}
\Lambda_{d,\mu} &= \sup_{x\in\mathcal{X}} \int_{\mathcal{X}} \overline{\varphi}(x)^\top \overline{\varphi}(z) \, d\mu(z) \\
&= \sup_{x\in\mathcal{X}} \int_{\mathcal{X}} \varphi(x)^\top R(\mu)^{-1} R(\mu)^{-\top} \, d\mu(z) \\
&= \sup_{x\in\mathcal{X}} \int_{\mathcal{X}} |\varphi(x)^\top G(\mu)^{-1}\varphi(z)| \, d\mu(z),
\end{aligned}$$

where $G(\mu) = \int_{\mathcal{X}} \varphi(x)\varphi(x)^\top \, d\mu(x)$ and $R(\mu)$ is its Cholesky factor, such that $R(\mu)^\top R(\mu) = G(\mu)$; here, the second passage comes from the fact that the Cholesky factor of a matrix corresponds to the $R$ factor in the $QR$ factorization, which is the one giving Graham-Schmidt orthogonalization Quarteroni et al. [2010]. In fact, letting $\overline{\varphi}(x)$ be the basis orthonomalized w.r.t. $\mu$, we have

$$\overline{\varphi}(x)^\top \overline{\varphi}(z)^\top = \varphi(x)^\top G(\mu)^{-1}\varphi(z).$$

Note that, by absolute continuity, we have, for any $x \in \mathcal{X}$

$$\int_{\mathcal{X}} |\boldsymbol{\varphi}(x)^\top G(\mu)^{-1} \boldsymbol{\varphi}(z)| \, d\mu(z) \le C \int_{\mathcal{X}} |\boldsymbol{\varphi}(x)^\top G(\mu)^{-1} \boldsymbol{\varphi}(z)| \, d\nu(z)$$

$$\le C \int_{\mathcal{X}} \left| \boldsymbol{\varphi}(x)^\top \left( \int_{\mathcal{X}} \boldsymbol{\varphi}(z')\boldsymbol{\varphi}(z')^\top \, d\mu(z') \right)^{-1} \boldsymbol{\varphi}(z) \right| \, d\nu(z)$$

$$\le C \int_{\mathcal{X}} \left| \boldsymbol{\varphi}(x)^\top c^{-1} \left( \int_{\mathcal{X}} \boldsymbol{\varphi}(z')\boldsymbol{\varphi}(z')^\top \, d\nu(z') \right)^{-1} \boldsymbol{\varphi}(z) \right| \, d\nu(z)$$

$$= \frac{C}{c} \int_{\mathcal{X}} \left| \boldsymbol{\varphi}(x)^\top \left( \int_{\mathcal{X}} \boldsymbol{\varphi}(z')\boldsymbol{\varphi}(z')^\top \, d\nu(z') \right)^{-1} \boldsymbol{\varphi}(z) \right| \, d\nu(z)$$

$$= \frac{C}{c} \int_{\mathcal{X}} |\boldsymbol{\varphi}(x)^\top G(\nu)^{-1} \boldsymbol{\varphi}(z)| \, d\nu(z).$$

Passing to the supremum, we get the thesis. $\qquad\square$

## D    Proofs from Section 4

### D.1    Lower bound for standard ridge regression

**Lemma 7.** *Let* $\Pi_{d,\mu}^\lambda$ *be the operator defined in this way:*

$$\Pi_{d,\mu}^\lambda f := \overline{\boldsymbol{\varphi}}(\cdot)^\top \theta_\lambda \qquad \theta_\lambda = \arg\min_\theta \|f(\cdot) - \overline{\boldsymbol{\varphi}}(\cdot)^\top \theta\|_{L^2}^2 + \lambda\|\theta\|_2^2. \tag{12}$$

*Then, we have*

$$\Pi_{d,\mu}^\lambda f = \frac{\Pi_{d,\mu} f}{1 + \lambda}.$$

*Proof.* We start from the definition of $\theta_\lambda$:

$$\theta_\lambda = \arg\min_\theta \|f(\cdot) - \overline{\boldsymbol{\varphi}}(\cdot)^\top \theta\|_{L^2}^2 + \lambda\|\theta\|_2^2$$

$$= \arg\min_\theta \|\Pi_{d,\mu} f(\cdot) + \zeta_{d,\mu}(\cdot) - \overline{\boldsymbol{\varphi}}(\cdot)^\top \theta\|_{L^2}^2 + \lambda\|\theta\|_2^2$$

$$= \arg\min_\theta \|\zeta_{d,\mu}\|_{L^2}^2 + \|\Pi_{d,\mu} f(\cdot) - \overline{\boldsymbol{\varphi}}(\cdot)^\top \theta\|_{L^2}^2 + \lambda\|\theta\|_2^2,$$

where the last passage comes from Parseval's theorem, as $\zeta_{d,\mu}$ is orthogonal in $L^2$ to the span of $\boldsymbol{\varphi}$, while $\Pi_{d,\mu} f(\cdot), \overline{\boldsymbol{\varphi}}(\cdot)^\top \theta$ belongs to this vector space. We then write the operator $\Pi_{d,\mu} f$ explicitly:

$$\theta_\lambda = \arg\min_\theta \|\Pi_{d,\mu} f(\cdot) - \overline{\boldsymbol{\varphi}}(\cdot)^\top \theta\|_{L^2}^2 + \lambda\|\theta\|_2^2$$

$$= \arg\min_\theta \left\| \sum_{i=1}^d \langle f, \overline{\varphi}_i \rangle_{L^2} \overline{\varphi}_i(\cdot) - \overline{\boldsymbol{\varphi}}(\cdot)^\top \theta \right\|_{L^2}^2 + \lambda\|\theta\|_2^2$$

$$= \arg\min_\theta \sum_{i=1}^d (\langle f, \overline{\varphi}_i \rangle_{L^2} - \theta_i)^2 + \lambda\theta_i^2.$$

The last passage holds from Parseval's theorem since $\overline{\varphi}_i$ are orthonormal in $L^2$. Note that, as the $\theta_i$ in the last minimization problem are disentangled, we can find as explicit solution

$$\theta_{\lambda,i} = \frac{\langle f, \overline{\varphi}_i \rangle_{L^2}}{1 + \lambda}, \qquad \Pi_{d,\mu}^\lambda f = \frac{\Pi_{d,\mu} f}{1 + \lambda}.$$

This completes the proof. $\qquad\square$

**Lemma 8.** *Let $\Pi_{d,\mu}^\lambda$ be defined according to Eq. (12). For every feature map $\varphi$ we have*

$$\sup_{f \in L^\infty(\mathcal{X})} \frac{\|\Pi_{d,\mu}^\lambda f - f\|_\infty}{\|\Pi_\infty f - f\|_\infty} \geq \left( \frac{\Lambda_{d,\mu} - 1 - 2\lambda}{1 + \lambda} \right).$$

*Proof.* By definition of Lebesgue constant, for every $\varepsilon > 0$ there is a function $g$ such that

$$\|\Pi_{d,\mu} g\|_\infty = (\Lambda_{d,\mu} - \varepsilon)\|g\|_\infty.$$

Take $f = \Pi_\infty g - g$. We have, by Lemma 7,

$$
\begin{aligned}
\|\Pi_{d,\mu}^\lambda f - f\|_\infty &= \left\| \frac{\Pi_{d,\mu} f}{1 + \lambda} - f \right\|_\infty \\
&= \|(1 + \lambda)^{-1}\Pi_{d,\mu}(P_\infty^d g - g) - \Pi_\infty g + g\|_\infty \\
&= \|(1 + \lambda)^{-1}\Pi_\infty g - (1 + \lambda)^{-1}\Pi_{d,\mu} g - P_\infty^d g + g\|_\infty \\
&= \left\| -(1 + \lambda)^{-1}\Pi_{d,\mu} g - \frac{\lambda}{1 + \lambda} P_\infty^d g + g \right\|_\infty.
\end{aligned}
$$

At this point, note that

$$\|\Pi_\infty g\|_\infty \leq 2\|g\|_\infty,$$

as follows from

$$
\begin{aligned}
\|\Pi_\infty g\|_\infty &\leq \|g - \Pi_\infty g\|_\infty + \|g\|_\infty \\
&\leq \|g - 0\|_\infty + \|g\|_\infty = 2\|g\|_\infty.
\end{aligned}
$$

Using this property, we have

$$
\begin{aligned}
\|\Pi_{d,\mu}^\lambda f - f\|_\infty &\geq \left\| -(1 + \lambda)^{-1}\Pi_{d,\mu} g - \frac{\lambda}{1 + \lambda}\Pi_\infty g + g \right\|_\infty \\
&\geq \| -(1 + \lambda)^{-1}\Pi_{d,\mu} g\|_\infty - \frac{1 + 2\lambda}{1 + \lambda}\|g\|_\infty.
\end{aligned}
$$

At this point, using the definition of $g$,

$$
\begin{aligned}
\| -(1 + \lambda)^{-1}\Pi_{d,\mu} g\|_\infty - \frac{1 + 2\lambda}{1 + \lambda}\|g\|_\infty &\geq \left( \frac{\Lambda_{d,\mu}}{1 + \lambda} - \varepsilon - \frac{1 + 2\lambda}{1 + \lambda} \right) \|\Pi_\infty g - g\|_\infty \\
&= \left( \frac{\Lambda_{d,\mu}}{1 + \lambda} - \varepsilon - \frac{1 + 2\lambda}{1 + \lambda} \right) \|f\|_\infty \\
&\geq \left( \frac{\Lambda_{d,\mu}}{1 + \lambda} - \varepsilon - \frac{1 + 2\lambda}{1 + \lambda} \right) \|\Pi_\infty f - f\|_\infty.
\end{aligned}
$$

The thesis follows letting $\varepsilon \to 0$. $\qquad\square$

**Theorem 5.** *Let $\hat{\theta}_{n,\text{RIDGE}}$ be the $\lambda-$ridge regression estimate. For any feature map $\varphi_d(\cdot) : \mathcal{X} \to \mathbb{R}^d$, there exists a target function $f \in L^\infty(\mathcal{X})$ such that, in the infinite data limit,*

$$\mathcal{E}_\infty(\hat{\theta}_{\infty,\text{RIDGE}}) = \Omega \left( \max \left\{ (\Lambda_{d,\mu} - 2\lambda)\mathcal{E}_\infty(f), \frac{\lambda}{\lambda + 1} \right\} \right).$$

*Proof.* Let $\widehat{f}_n$ be the output of $\lambda-$ridge regression, that is the function $\overline{\varphi}(\cdot)^\top \hat{\theta}_n$, where

$$\hat{\theta}_n := \underset{\theta \in \mathbb{R}^d}{\arg\min} \sum_{t=1}^n (\overline{\varphi}(x_t)^\top \theta - y_t)^2 + \lambda n\|\theta\|_2^2 \qquad x_t \overset{i.i.d.}{\sim} \mu.$$

By the uniform law of large numbers, in the limit, the minimizer $\widehat{f}_n$ converges to $\Pi_{d,\mu}^\lambda f$, the regularized projection operator is defined as follows

$$\Pi_{d,\mu}^\lambda f(\cdot) := \overline{\varphi}(\cdot)^\top \theta_\lambda \qquad \theta_\lambda = \arg\min_\theta \|f(\cdot) - \overline{\varphi}(\cdot)^\top \theta\|_{L^2}^2 + \lambda\|\theta\|_2^2.$$

We start showing the $\frac{\lambda}{\lambda+1}$ lower bound. Taking any function in the span of $\varphi(\cdot)$ with $\|f\|_\infty = 1$ we have, by Lemma 7,

$$\|f - \Pi_{d,\mu}^\lambda f\|_\infty = \|f - (1+\lambda)^{-1}f\|_\infty = \frac{\lambda}{\lambda+1}.$$

To show the other part, use Lemma 8 to define a function $f$ such that

$$\|\Pi_{d,\mu}^\lambda f - f\|_\infty \geq \left(\frac{\Lambda_{d,\mu} - 2 - 2\lambda}{1+\lambda}\right)\|\Pi_\infty f - f\|_\infty.$$

Replacing $\|\Pi_\infty f - f\|_\infty = \mathcal{E}_\infty(f)$ completes the proof. $\qquad\square$

## D.2   Proofs from Section 4.1

**Theorem 7.** *Let Assumption 1 hold. Then, for any $\delta > 0$, with probability $1 - \delta$,*

$$\mathcal{E}_\infty(\hat{\theta}_{n,\boldsymbol{\alpha}}) \leq (1 + \Lambda_{\boldsymbol{\alpha},\mu_n})\mathcal{E}_\infty(f) + \frac{\sigma\widehat{\varphi}_{2,D}\sqrt{2\log(2|\mathcal{X}|/\delta)}}{\sqrt{n}}.$$

*Proof.* In this proof, let $\zeta(\cdot) := f(\cdot) - \Pi_{\boldsymbol{\alpha},\mu_n}^{\text{Ridge}} f(\cdot)$ and $\eta_t = y_t - f(x_t)$ and $\hat{\theta}_n$ be the estimator corresponding to $\Pi_{\boldsymbol{\alpha},\mu_n}^{\text{Ridge}}$ in the parameterization of $\widehat{\varphi}_D(\cdot)$, so that

$$\widehat{\varphi}_d(\cdot)^\top \hat{\theta}_n = \Pi_{\boldsymbol{\alpha},\mu_n}^{\text{Ridge}} \mathbf{f} =: \widehat{f}_n(\cdot).$$

The following decomposition holds:

$$\|f(\cdot) - \widehat{f}_n(\cdot)\|_\infty \leq \|f(\cdot) - \Pi_{\boldsymbol{\alpha},\mu_n}^{\text{Ridge}} f(\cdot)\|_\infty + \|\Pi_{\boldsymbol{\alpha},\mu_n}^{\text{Ridge}} f(\cdot) - \widehat{f}_n(\cdot)\|_\infty$$
$$\leq (1 + \Lambda_{\boldsymbol{\alpha},\mu_n})\mathcal{E}_\infty(f) + \|\Pi_{\boldsymbol{\alpha},\mu_n}^{\text{Ridge}} f(\cdot) - \widehat{f}_n(\cdot)\|_\infty.$$

where we have applied Proposition 6 for $\mu_n$. Let us focus on the second term. As in the proof of the previous theorems, we call $\theta_\star$ the vector corresponding to the orthogonal projection over $\widehat{\varphi}_D(\cdot)$ so that we have, for every $x \in \mathcal{X}$

$$\widehat{f}_n(x) = \widehat{\varphi}_D(x)^\top I_{\boldsymbol{\alpha}} \frac{1}{n}\sum_{t=1}^n y_t \widehat{\varphi}_D(x_t)$$

$$= \widehat{\varphi}_D(x)^\top I_{\boldsymbol{\alpha}} \frac{1}{n}\sum_{t=1}^n (\widehat{\varphi}_D(x_t)^\top \theta_\star + \zeta(x_t) + \eta_t)\widehat{\varphi}_D(x_t)$$

$$= \widehat{\varphi}_D(x)^\top I_{\boldsymbol{\alpha}} \frac{1}{n}\sum_{t=1}^n \widehat{\varphi}_D(x_t)\widehat{\varphi}_D(x_t)^\top \theta_\star$$

$$+ \widehat{\varphi}_D(x)^\top I_{\boldsymbol{\alpha}} \frac{1}{n}\sum_{t=1}^n \zeta(x_t)\widehat{\varphi}_D(x_t)$$

$$+ \widehat{\varphi}_D(x)^\top I_{\boldsymbol{\alpha}} \frac{1}{n}\sum_{t=1}^n \eta_t \widehat{\varphi}_D(x_t).$$

By orthogonality, the first term corresponds to

$$\widehat{\boldsymbol{\varphi}}_D(x)^\top I_{\boldsymbol{\alpha}} \underbrace{\frac{1}{n}\sum_{t=1}^n \widehat{\boldsymbol{\varphi}}_D(x_t)\widehat{\boldsymbol{\varphi}}_D(x_t)^\top}_{I_D} \theta_\star = \widehat{\boldsymbol{\varphi}}_D(x)^\top I_{\boldsymbol{\alpha}}\theta_\star = \Pi^{\text{Ridge}}_{\boldsymbol{\alpha},\mu_n} f(x).$$

The second term is

$$\widehat{\boldsymbol{\varphi}}_D(x)^\top I_{\boldsymbol{\alpha}}\frac{1}{n}\sum_{t=1}^n \zeta(x_t)\widehat{\boldsymbol{\varphi}}_D(x_t) = \widehat{\boldsymbol{\varphi}}_D(x)^\top I_{\boldsymbol{\alpha}} \underbrace{\int_{\mathcal{X}}\zeta(z)\widehat{\boldsymbol{\varphi}}_D(z)\,d\mu_n(z)}_{\mathbf{0}\ \text{vector}} = 0,$$

by definition of orthogonal projection. The third term is

$$\widehat{\boldsymbol{\varphi}}_D(x)^\top I_{\boldsymbol{\alpha}}\frac{1}{n}\sum_{t=1}^n \eta_t\widehat{\boldsymbol{\varphi}}_D(x_t),$$

which can be bounded as the corresponding terms in Theorems 2 and 3: as $\eta_t$ are independent and $\sigma-$subgaussian subgaussian conditionally on $(x_t)_{t=1}^n$, Lemma 5.4 and Theorem 5.3 from Lattimore and Szepesvári [2020] ensure that, with probability at least $1 - 2\delta$

$$\left|\widehat{\boldsymbol{\varphi}}_d(x)^\top I_{\boldsymbol{\alpha}} n^{-1}\sum_{t=1}^n \widehat{\boldsymbol{\varphi}}_d(x_t)\eta_t\right| \leq \sqrt{2\sigma^2 n^{-1}\sum_{t=1}^n (\widehat{\boldsymbol{\varphi}}_d(x)^\top I_{\boldsymbol{\alpha}}\widehat{\boldsymbol{\varphi}}_d(x_t))^2 \log(1/\delta)}$$

$$= \sqrt{2\sigma^2 n^{-1}\|\widehat{\boldsymbol{\varphi}}_d(x)\|_2^2 \log(1/\delta)}$$

$$= \sqrt{2\log(1/\delta)}\sigma n^{-1/2}\|\widehat{\boldsymbol{\varphi}}_d(x)\|_2.$$

Where the only difference w.r.t. the other proofs is the presence of $I_{\boldsymbol{\alpha}}$, which is erased after the first step since, being $\boldsymbol{\alpha} \in \mathcal{A}_d^D$, its norm is $\leq 1$. This proves that the last term is bounded by $\sqrt{2\log(1/\delta)}\sigma n^{-1/2}\widehat{\varphi}_{2,D}$. Making a union bound over $\mathcal{X}$ gives, w.p. $1 - \delta$,

$$\sup_{x\in\mathcal{X}}|\Pi^{\text{Ridge}}_{\boldsymbol{\alpha},\mu_n} f(x) - \widehat{f}_n(x)| \leq \sqrt{2\log(1/\delta)}\sigma n^{-1/2}\widehat{\varphi}_{2,D}.$$

Putting everything together, we have proved that

$$\mathcal{E}_\infty(\hat{\theta}_{n,\boldsymbol{\alpha}}) \leq \|f(\cdot) - \widehat{f}_n(\cdot)\|_\infty \leq (1 + \widehat{\Lambda}_{\boldsymbol{\alpha}})\mathcal{E}_\infty(f) + \frac{\sigma\widehat{\varphi}_{2,D}\sqrt{2\log(2\mathcal{X}/\delta)}}{\sqrt{n}}.$$

$\square$

**Proposition 21.** *Fix $\delta > 0$. With probability $1 - \delta$, the following bounds holds simultaneously for every $\boldsymbol{\alpha} \in \mathcal{A}_d^D$: $|\vartheta_{D,2} - \vartheta_{D,2}^{(n)}| = \widetilde{\mathcal{O}}(\vartheta_{D,2}^2\sqrt{\log(1/\delta)/n})$, and*

$$|\Lambda_{\boldsymbol{\alpha},\mu_n} - \Lambda_{\boldsymbol{\alpha},\mu}| = \widetilde{\mathcal{O}}\left(\frac{\sqrt{d}\vartheta_{D,2}^2\sqrt{\log(1/\delta)}}{\sqrt{n}} + \frac{\sqrt{d}\vartheta_{D,2}^3\log(1/\delta)}{n}\right).$$

We prove this theorem for a generic $d \in \mathbb{N}$. The result follows for $d = D$.

We define $V_n := \frac{1}{n}\sum_{t=1}^n \overline{\boldsymbol{\varphi}}_d(x_t)\overline{\boldsymbol{\varphi}}_d(x_t)^\top$. Let $\widehat{\boldsymbol{\varphi}}_d(\cdot)$ the basis obtained from $\boldsymbol{\varphi}_d$ by Gram-Schmidt orthogonalization w.r.t. $\mu_n$, the empirical distribution of the $\{x_t\}_t$. As in the main paper, we let $R_n = \text{Chol}(V_n)$ and, since the Cholesky factor corresponds to the matrix given by Graham Schmidt orthogonalization (proposition 3.4 in Quarteroni et al. [2010]),

$$\overline{\boldsymbol{\varphi}}_d(x_t) = R_n^\top\widehat{\boldsymbol{\varphi}}_d(x_t) \qquad \widehat{\boldsymbol{\varphi}}_d(x_t) = R_n^{-\top}\overline{\boldsymbol{\varphi}}_d(x_t). \tag{13}$$

so that, under this convenient normalization, we can pass from $\overline{\varphi}_d(x_t)$ to $\widehat{\varphi}_d(x_t)$ trough a matrix that is exactly the Cholesky factor of $V_n$. In this setting, Theorem 2.1. in Drmač et al. [1994], which provides a stability result for the Cholesky decomposition which, combined with our theorem gives

$$1 - \mathcal{O}\left(\vartheta_{d,2}\sqrt{\log(1/\delta)/n}\log(d)\right) \leq \lambda_{\min}(R_n) \leq \lambda_{\max}(R_n) \leq 1 + \mathcal{O}\left(\vartheta_{d,2}\sqrt{\log(1/\delta)/n}\log(d)\right) \tag{14}$$

We can now proceed with the proof.

*Proof.* **Bounding norm difference**

We have to measure

$$\sup_{x\in\mathcal{X}}\|\widehat{\varphi}_d(x) - \overline{\varphi}_d(x)\|_2.$$

As we said, the relation between the two is $\overline{\varphi}_d(x) = R_n^\top\widehat{\varphi}_d(x)$ which we can also wite as $R_n^{-\top}\overline{\varphi}_d(x) = \widehat{\varphi}_d(x)$, so that

$$\sup_{x\in\mathcal{X}}\|\widehat{\varphi}_d(x) - \overline{\varphi}_d(x)\|_2 = \sup_{x\in\mathcal{X}}\|(I_d - R_n^{-\top})\overline{\varphi}_d(x)\|_2.$$

At this point, equation (14) ensures that $\|I_d - R_n^{-\top}\|_{2\to2} = \mathcal{O}\left(\vartheta_{d,2}\sqrt{\log(1/\delta)/n}\log(d)\right)$, so we get

$$\sup_{x\in\mathcal{X}}\|\widehat{\varphi}_d(x) - \overline{\varphi}_d(x)\|_2 \leq \mathcal{O}\left(\vartheta_{d,2}^2\sqrt{\log(1/\delta)/n}\log(d)\right). \tag{15}$$

A simple yet useful consequence of this result is

$$|\vartheta_{d,2} - \vartheta_{d,2}^{(n)}| = \sup_{x\in\mathcal{X}}\|\|\widehat{\varphi}_d(x)\|_2 - \sup_{x\in\mathcal{X}}\|\overline{\varphi}_d(x)\|_2| \tag{16}$$

$$\leq \sup_{x\in\mathcal{X}}|\|\widehat{\varphi}_d(x)\|_2 - \|\overline{\varphi}_d(x)\|_2| \tag{17}$$

$$\leq \sup_{x\in\mathcal{X}}\|\widehat{\varphi}_d(x) - \overline{\varphi}_d(x)\|_2 \tag{18}$$

$$= \mathcal{O}\left(\vartheta_{d,2}^2\sqrt{\log(1/\delta)/n}\log(d)\right) \tag{19}$$

**Lebesgue constants difference**

Let us bound the distance between the estimated and the true Lebesgue constant, for any $\alpha \in \mathcal{A}_d^D$,

$$|\Lambda_{\boldsymbol{\alpha},\mu_n} - \Lambda_{\boldsymbol{\alpha},\mu}| = \left| \sup_{x\in\mathcal{X}} \frac{1}{n} \sum_{t=1}^{n} \left| \sum_{i=1}^{d} \alpha_i \widehat{\varphi}_i(x) \widehat{\varphi}_i(x_t) \right| - \sup_{x\in\mathcal{X}} \int_{\mathcal{X}} \left| \sum_{i=1}^{d} \alpha_i \overline{\varphi}_i(x) \overline{\varphi}_i(z) \right| d\mu(z) \right|$$

$$\leq \sup_{x\in\mathcal{X}} \left| \frac{1}{n} \sum_{t=1}^{n} \left| \sum_{i=1}^{d} \alpha_i \widehat{\varphi}_i(x) \widehat{\varphi}_i(x_t) \right| - \int_{\mathcal{X}} \left| \sum_{i=1}^{d} \alpha_i \overline{\varphi}_i(x) \overline{\varphi}_i(z) \right| d\mu(z) \right|$$

$$\leq \sup_{x\in\mathcal{X}} \left| \frac{1}{n} \sum_{t=1}^{n} \left| \sum_{i=1}^{d} \alpha_i \widehat{\varphi}_i(x) \widehat{\varphi}_i(x_t) \right| - \frac{1}{n} \sum_{t=1}^{n} \left| \sum_{i=1}^{d} \alpha_i \overline{\varphi}_i(x) \overline{\varphi}_i(x_t) \right| \right|$$

$$+ \sup_{x\in\mathcal{X}} \left| \frac{1}{n} \sum_{t=1}^{n} \left| \sum_{i=1}^{d} \alpha_i \overline{\varphi}_i(x) \overline{\varphi}_i(x_t) \right| - \int_{\mathcal{X}} \left| \sum_{i=1}^{d} \alpha_i \overline{\varphi}_i(x) \overline{\varphi}_i(z) \right| d\mu(z) \right|$$

$$= \sup_{x\in\mathcal{X}} \left| \frac{1}{n} \sum_{t=1}^{n} \left| \sum_{i=1}^{d} \alpha_i \widehat{\varphi}_i(x) \widehat{\varphi}_i(x_t) \right| - \left| \sum_{i=1}^{d} \alpha_i \overline{\varphi}_i(x) \overline{\varphi}_i(x_t) \right| \right|$$

$$+ \sup_{x\in\mathcal{X}} \left| \frac{1}{n} \sum_{t=1}^{n} \left| \sum_{i=1}^{d} \alpha_i \overline{\varphi}_i(x) \overline{\varphi}_i(x_t) \right| - \int_{\mathcal{X}} \left| \sum_{i=1}^{d} \alpha_i \overline{\varphi}_i(x) \overline{\varphi}_i(z) \right| d\mu(z) \right| .$$

In the following, we call

$$\text{First term} := \sup_{\boldsymbol{\alpha}\in\mathcal{A}_d^D, x\in\mathcal{X}} \left| \frac{1}{n} \sum_{t=1}^{n} \left| \sum_{i=1}^{d} \alpha_i \widehat{\varphi}_i(x) \widehat{\varphi}_i(x_t) \right| - \left| \sum_{i=1}^{d} \alpha_i \overline{\varphi}_i(x) \overline{\varphi}_i(x_t) \right| \right|$$

and

$$\text{Second term} := \sup_{\boldsymbol{\alpha}\in\mathcal{A}_d^D, x\in\mathcal{X}} \left| \frac{1}{n} \sum_{t=1}^{n} \left| \sum_{i=1}^{d} \alpha_i \overline{\varphi}_i(x) \overline{\varphi}_i(x_t) \right| - \int_{\mathcal{X}} \left| \sum_{i=1}^{d} \alpha_i \overline{\varphi}_i(x) \overline{\varphi}_i(z) \right| d\mu(z) \right| .$$

**Bound the first term**.

Fix $\boldsymbol{\alpha} \in \mathcal{A}_d^D$,

$$\text{First part} = \frac{1}{n} \sum_{t=1}^{n} \left| \left| \sum_{i=1}^{d} \alpha_i \widehat{\varphi}_i(x) \widehat{\varphi}_i(x_t) \right| - \left| \sum_{i=1}^{d} \alpha_i \overline{\varphi}_i(x) \overline{\varphi}_i(x_t) \right| \right|$$

$$\leq \frac{1}{n} \sum_{t=1}^{n} \left| \sum_{i=1}^{d} \alpha_i \widehat{\varphi}_i(x) \widehat{\varphi}_i(x_t) - \sum_{i=1}^{d} \alpha_i \overline{\varphi}_i(x) \overline{\varphi}_i(x_t) \right|$$

$$= \frac{1}{n} \sum_{t=1}^{n} \left| \widehat{\boldsymbol{\varphi}}_d(x)^\top I_{\boldsymbol{\alpha}} \widehat{\boldsymbol{\varphi}}_d(x_t) - \overline{\boldsymbol{\varphi}}_d(x)^\top I_{\boldsymbol{\alpha}} \overline{\boldsymbol{\varphi}}_d(x_t) \right| .$$

Where, $I_{\boldsymbol{\alpha}} = \text{diag}(\boldsymbol{\alpha})$. At this point, we can replace the result of Eq. (13): getting

$$\text{First part} \leq \frac{1}{n}\sum_{t=1}^{n}\left|\widehat{\boldsymbol{\varphi}}_d(x)^\top I_{\boldsymbol{\alpha}}\widehat{\boldsymbol{\varphi}}_d(x_t) - \overline{\boldsymbol{\varphi}}_d(x)^\top I_{\boldsymbol{\alpha}}\overline{\boldsymbol{\varphi}}_d(x_t)\right|$$

$$= \frac{1}{n}\sum_{t=1}^{n}\left|\widehat{\boldsymbol{\varphi}}_d(x)^\top I_{\boldsymbol{\alpha}}\widehat{\boldsymbol{\varphi}}_d(x_t) - \widehat{\boldsymbol{\varphi}}_d(x)^\top R_n I_{\boldsymbol{\alpha}}R_n^\top \widehat{\boldsymbol{\varphi}}_d(x_t)\right|$$

$$\leq \frac{1}{n}\sum_{t=1}^{n}\left|\widehat{\boldsymbol{\varphi}}_d(x)^\top (I_{\boldsymbol{\alpha}} - R_n I_{\boldsymbol{\alpha}}R_n^\top)\widehat{\boldsymbol{\varphi}}_d(x_t)\right|$$

$$\leq \frac{1}{n}\sum_{t=1}^{n}\|\widehat{\boldsymbol{\varphi}}_d(x)\|_2\|I_{\boldsymbol{\alpha}} - R_n I_{\boldsymbol{\alpha}}R_n^\top\|_2\|\widehat{\boldsymbol{\varphi}}_d(x_t)\|_2.$$

This formulation allows us to apply Eq. (14): As $I_{\boldsymbol{\alpha}}$ is diagonal matrix with elements in $[0,1]$, we have

$$\|I_{\boldsymbol{\alpha}} - R_n I_{\boldsymbol{\alpha}}R_n^\top\|_2 = \mathcal{O}\left(\vartheta_{d,2}\sqrt{\log(1/\delta)/n}\log(d)\right).$$

This gives the following

$$\text{First part} \leq \mathcal{O}\left(\frac{1}{n}\sum_{t=1}^{n}\|\widehat{\boldsymbol{\varphi}}_d(x)\|_2\|H - R_n H R_n^\top\|_2\|\widehat{\boldsymbol{\varphi}}_d(x_t)\|_2\right)$$

$$\leq \mathcal{O}\left(\frac{\vartheta_{d,2}\sqrt{\log(1/\delta)}\log(d)}{\sqrt{n}}\frac{1}{n}\sum_{t=1}^{n}\|\widehat{\boldsymbol{\varphi}}_d(x)\|_2\|\widehat{\boldsymbol{\varphi}}_d(x_t)\|_2\right)$$

$$\leq \mathcal{O}\left(\frac{\vartheta_{d,2}\vartheta_{d,2}^{(n)}\sqrt{\log(1/\delta)}\log(d)}{\sqrt{n}}\frac{\sum_{t=1}^{n}\|\widehat{\boldsymbol{\varphi}}_d(x_t)\|_2}{n}\right)$$

$$\leq \mathcal{O}\left(\frac{\vartheta_{d,2}\vartheta_{d,2}^{(n)}\sqrt{\log(1/\delta)}\log(d)}{\sqrt{n}}\frac{\sqrt{n\sum_{t=1}^{n}\|\widehat{\boldsymbol{\varphi}}_d(x_t)\|_2^2}}{n}\right)$$

$$= \mathcal{O}\left(\frac{\vartheta_{d,2}\vartheta_{d,2}^{(n)}\sqrt{\log(1/\delta)}\log(d)}{\sqrt{n}}\frac{\sqrt{n^2 d}}{n}\right)$$

$$= \mathcal{O}\left(\frac{\sqrt{d}\vartheta_{d,2}\vartheta_{d,2}^{(n)}\sqrt{\log(1/\delta)}\log(d)}{\sqrt{n}}\right).$$

Here, the first equality is due to the fact that, being $\widehat{\boldsymbol{\varphi}}_d$ orthonormal w.r.t. $\mu_n$, we have $\sum_{t=1}^{n}\|\widehat{\boldsymbol{\varphi}}_d(x_t)\|_2^2 = nd$. This holds uniformly for every $\boldsymbol{\alpha}$, as we have only used the fact that $\|I_{\boldsymbol{\alpha}}\|_2 \leq 1$.

**Bounding the second term.**

The second term corresponds to

$$\text{Second term} = \sup_{x \in \mathcal{X}}\left|\frac{1}{n}\sum_{t=1}^{n}\left|\sum_{i=1}^{d}\alpha_i\overline{\varphi}_i(x)\overline{\varphi}_i(x_t)\right| - \int_{\mathcal{X}}\left|\sum_{i=1}^{d}\alpha_i\overline{\varphi}_i(x)\overline{\varphi}_i(z)\right|d\mu(z)\right|.$$

First, we fix $x \in \mathcal{X}$ and $\boldsymbol{\alpha} \in \mathcal{A}_d^D$ and use the scalar product to write it as

$$\left|\frac{1}{n}\sum_{t=1}^{n}\left|\overline{\boldsymbol{\varphi}}_d(x)^\top I_{\boldsymbol{\alpha}}\overline{\boldsymbol{\varphi}}_d(x_t)\right| - \int_{\mathcal{X}}\left|\overline{\boldsymbol{\varphi}}_d(x)^\top I_{\boldsymbol{\alpha}}\overline{\boldsymbol{\varphi}}_d(z)\right|d\mu(z)\right|. \tag{20}$$

Note that by definition

$$\mathbb{E}[|\overline{\varphi}_d(x)^\top I_{\boldsymbol{\alpha}} \overline{\varphi}_d(x_t)|] = \int_{\mathcal{X}} |\overline{\varphi}_d(x)^\top I_{\boldsymbol{\alpha}} \overline{\varphi}_d(z)| \, d\mu(z).$$

Moreover,

$$
\begin{aligned}
\mathrm{Var}(|\overline{\varphi}_d(x)^\top I_{\boldsymbol{\alpha}} \overline{\varphi}_d(x_t)|) &\leq \mathbb{E}\left[|\overline{\varphi}_d(x)^\top I_{\boldsymbol{\alpha}} \overline{\varphi}_d(x_t)|^2\right] \\
&= \mathbb{E}\left[\overline{\varphi}_d(x)^\top I_{\boldsymbol{\alpha}} \overline{\varphi}_d(x_t) \overline{\varphi}_d(x_t)^\top I_{\boldsymbol{\alpha}} \overline{\varphi}_d(x)\right] \\
&= \overline{\varphi}_d(x)^\top I_{\boldsymbol{\alpha}} \underbrace{\mathbb{E}\left[\overline{\varphi}_d(x_t) \overline{\varphi}_d(x_t)^\top\right]}_{=I_d} I_{\boldsymbol{\alpha}} \overline{\varphi}_d(x) \\
&= \overline{\varphi}_d(x)^\top I_{\boldsymbol{\alpha}}^2 \overline{\varphi}_d(x) \\
&\leq \vartheta_{d,2}^2,
\end{aligned}
$$

where the last step comes from the fact that $I_{\boldsymbol{\alpha}}^2 \preceq I_d$. For the same reason, we also have $|\overline{\varphi}_d(x)^\top I_{\boldsymbol{\alpha}} \overline{\varphi}_d(x_t)| \leq \vartheta_{d,2}^2$ almost surely. These three results allow us to apply Bernstein's inequality (14) for

- $X_t = |\overline{\varphi}_d(x)^\top I_{\boldsymbol{\alpha}} \overline{\varphi}_d(x_t)| - \mathbb{E}[|\overline{\varphi}_d(x)^\top I_{\boldsymbol{\alpha}} \overline{\varphi}_d(x_t)|]$.
- $\sigma^2 = \sum_{t=1}^n \mathrm{Var}(|\overline{\varphi}_d(x)^\top I_{\boldsymbol{\alpha}} \overline{\varphi}_d(x_t)|) \leq n\vartheta_{d,2}^2$.
- $B = \vartheta_{d,2}^2$.

This gives, with probability at least $1 - \delta$,

$$\left|\sum_{t=1}^n X_t\right| \leq \sqrt{2n\vartheta_{d,2}^2 \log(2/\delta)} + \frac{2\vartheta_{d,2}^2}{3} \log(2/\delta).$$

So, we can bound Eq. (20), which corresponds to $\frac{1}{n}\left|\sum_{t=1}^n X_t\right|$, as follows.

$$\left|\frac{1}{n}\sum_{t=1}^n |\overline{\varphi}_d(x)^\top I_{\boldsymbol{\alpha}} \overline{\varphi}_d(x_t)| - \int_{\mathcal{X}} |\overline{\varphi}_d(x)^\top I_{\boldsymbol{\alpha}} \overline{\varphi}_d(z)| \, d\mu(z)\right| \leq \sqrt{\frac{2\vartheta_{d,2}^2 \log(2/\delta)}{n}} + \frac{2\vartheta_{d,2}^2}{3n} \log(2/\delta).$$

The former holds for any fixed $\boldsymbol{\alpha} \in \mathcal{A}_d^D$. To have a uniform bound, let

$$\mathcal{A}' = \varepsilon - \text{Cover of } \mathcal{A}_d^D \qquad \varepsilon = (n\vartheta_{d,2})^{-1},$$

so that $\log|\mathcal{A}'| \leq d \log(n\vartheta_{d,2})$. Making a union bound gives, $\forall \boldsymbol{\alpha} \in \mathcal{A}'$

$$
\begin{aligned}
&\left|\frac{1}{n}\sum_{t=1}^n |\overline{\varphi}_d(x)^\top I_{\boldsymbol{\alpha}} \overline{\varphi}_d(x_t)| - \int_{\mathcal{X}} |\overline{\varphi}_d(x)^\top I_{\boldsymbol{\alpha}} \overline{\varphi}_d(z)| \, d\mu(z)\right| \\
&\qquad\qquad \leq \sqrt{\frac{2d\vartheta_{d,2}^2 \log(2n\vartheta_{d,2}/\delta)}{n}} + \frac{2d\vartheta_{d,2}^2}{3n} \log(2n\vartheta_{d,2}/\delta).
\end{aligned}
$$

To pass to the general case, note that for every $\boldsymbol{\alpha} \in \mathcal{A}_d^D$ there is $\boldsymbol{\alpha}' \in \mathcal{A}'$ such that $\left|\frac{1}{n}\sum_{t=1}^n |\overline{\varphi}_d(x)^\top I_{\boldsymbol{\alpha}} \overline{\varphi}_d(x_t)| - \int_{\mathcal{X}} |\overline{\varphi}_d(x)^\top I_{\boldsymbol{\alpha}} \overline{\varphi}_d(z)| \, d\mu(z)\right|$ changes no more than $2\vartheta_{d,2}$ between the two, by definition of $\varepsilon-$cover. Therefore, we have, with probability at least $1 - \delta$ over all $\boldsymbol{\alpha} \in \mathcal{A}_d^D$ at the same time

$$\left| \frac{1}{n} \sum_{t=1}^{n} \left| \overline{\boldsymbol{\varphi}}_d(x)^{\top} I_{\boldsymbol{\alpha}} \overline{\boldsymbol{\varphi}}_d(x_t) \right| - \int_{\mathcal{X}} \left| \overline{\boldsymbol{\varphi}}_d(x)^{\top} I_{\boldsymbol{\alpha}} \overline{\boldsymbol{\varphi}}_d(z) \right| d\mu(z) \right|$$

$$\leq \sqrt{\frac{2d\vartheta_{d,2}^2 \log(2n\vartheta_{d,2}/\delta)}{n}} + \frac{2d\vartheta_{d,2}^2}{3n} \log(2n\vartheta_{d,2}/\delta) + 2\vartheta_{d,2}.$$

This means,

$$\text{Second term} \leq \widetilde{\mathcal{O}}\left( \sqrt{\frac{d\vartheta_{d,2}^2 \log(1/\delta)}{n}} + \frac{d\vartheta_{d,2}^2}{n} \log(1/\delta) \right).$$

**Putting the two results together.** By the two bounds that we got for the two terms, it follows with probability at least $1 - \delta$

$$\sup_{\boldsymbol{\alpha} \in \mathcal{A}_d^D} |\Lambda_{\boldsymbol{\alpha},\mu_n} - \Lambda_{\boldsymbol{\alpha},\mu}| \leq \widetilde{\mathcal{O}}\left( \frac{\sqrt{d}\vartheta_{d,2}\vartheta_{d,2}^{(n)} \sqrt{\log(1/\delta)}}{\sqrt{n}} + \sqrt{\frac{d\vartheta_{d,2}^2 \log(1/\delta)}{n}} + \frac{d\vartheta_{d,2}^2}{n} \log(1/\delta) \right).$$

To end the proof, note that, using Eq. (19), the difference between $\vartheta_{d,2}$ and $\vartheta_{d,2}^{(n)}$ is of order $\vartheta_{d,2}^2 \sqrt{\log(1/\delta)/n}$, so that

$$\frac{\sqrt{d}\vartheta_{d,2}\vartheta_{d,2}^{(n)} \sqrt{\log(1/\delta)}}{\sqrt{n}} \leq \frac{\sqrt{d}\vartheta_{d,2}(\vartheta_{d,2} + \vartheta_{d,2}^2 \sqrt{\log(1/\delta)/n}) \sqrt{\log(1/\delta)}}{\sqrt{n}}$$

$$= \frac{\sqrt{d}\vartheta_{d,2}^2 \sqrt{\log(1/\delta)}}{\sqrt{n}} + \frac{\sqrt{d}\vartheta_{d,2}^3 \log(1/\delta)}{n}.$$

Finally, note that, as $\sqrt{d} \leq \vartheta_{d,2}$, the term $\frac{\sqrt{d}\vartheta_{d,2}^3 \log(1/\delta)}{n}$ dominates over $\frac{d\vartheta_{d,2}^2}{n} \log(1/\delta)$ that we had before. $\qquad\square$

### D.3 Proofs about gradient method

**Proposition 22.** *The function $J : \mathcal{A}_d^D \to (0, +\infty)$ defined by $J(\boldsymbol{\alpha}) := \Lambda_{\boldsymbol{\alpha},\mu_n}$ is convex in $\boldsymbol{\alpha}$.*

*Proof.* By definition,

$$J(\boldsymbol{\alpha}) = \|M(\boldsymbol{\alpha})\|_{\infty},$$

where $M(\boldsymbol{\alpha}) = \frac{1}{n} \sum_{i=1}^{d} \alpha_i \widehat{\varphi}_i(x) \widehat{\varphi}_i(x_t)$. Therefore, in particular

$$J(\boldsymbol{\alpha}) = \sup_{x \in \mathcal{X}, \mathbf{f} \in \{-1,1\}^n} \left| \frac{1}{n} \sum_{i=1}^{d} \alpha_i \widehat{\varphi}_i(x) \widehat{\varphi}_i(x_t) \mathbf{f} \right|.$$

This function is convex, being the supremum of a family of linear functions $\frac{1}{n} \sum_{i=1}^{d} \alpha_i \widehat{\varphi}_i(x) \widehat{\varphi}_i(x_t)$ in $\boldsymbol{\alpha}$. $\qquad\square$

**Theorem 10.** *Fix $\epsilon > 0$. After $I = \widetilde{\mathcal{O}}(\epsilon^{-2} \vartheta_{D,2}^{(n)\,2}(D - d))$ iterations, Algorithm 1 outputs $\boldsymbol{\alpha}^{(I)} \in \mathcal{A}_d^D$ such that $J(\boldsymbol{\alpha}^{(I)}) \leq \inf_{\boldsymbol{\alpha} \in \mathcal{A}_d^D} J(\boldsymbol{\alpha}) + \epsilon$.*

*Proof.* The first step of this proof consists in finding an upper bound for any sub-gradient of $\boldsymbol{\alpha}$. As we said,

$$J(\boldsymbol{\alpha}) = \sup_{x \in \mathcal{X}, \mathbf{f} \in \{-1,1\}^n} \left| \frac{1}{n} \sum_{i=1}^{d} \alpha_i \widehat{\varphi}_i(x) \widehat{\varphi}_i(x_t) \mathbf{f} \right| = \sup_{x \in \mathcal{X}, \mathbf{f} \in \{-1,1\}^n} \left| \frac{1}{n} \widehat{\boldsymbol{\varphi}}_D(x)^\top I_{\boldsymbol{\alpha}} \widehat{\Phi}^\top \mathbf{f} \right|,$$

where $I_{\boldsymbol{\alpha}} = \mathrm{diag}(\boldsymbol{\alpha})$ is a $D \times D$ diagonal matrix and $\widehat{\Phi}$ is the $n \times d$ matrix having, as rows, $\widehat{\boldsymbol{\varphi}}_D(x_t)$ for each $t = 1, \ldots n$. At this point note that, by duality

$$J(\boldsymbol{\alpha}) = \sup_{x \in \mathcal{X}, \mathbf{f} \in \{-1,1\}^n} \left| \frac{1}{n} \widehat{\boldsymbol{\varphi}}_D(x)^\top I_{\boldsymbol{\alpha}} \widehat{\Phi}^\top \mathbf{f} \right| = \sup_{x \in \mathcal{X}} \frac{1}{n} \sum_{t=1}^{n} |\{\widehat{\boldsymbol{\varphi}}_D(x)^\top I_{\boldsymbol{\alpha}} \widehat{\Phi}^\top\}_t|,$$

where $\{\}_t$ denotes the $t-$th component of $\widehat{\boldsymbol{\varphi}}_D(x)^\top I_{\boldsymbol{\alpha}} \widehat{\Phi}^\top$, which is a $1 \times n$ row vector. Now, assuming[4] that the supremum is obtained by just one value $x_* \in \mathcal{X}$, we can compute the gradient as

$$\begin{aligned}
\nabla J(\boldsymbol{\alpha}) &= \nabla \frac{1}{n} \sum_{t=1}^{n} |\{\widehat{\boldsymbol{\varphi}}_D(x_*)^\top I_{\boldsymbol{\alpha}} \widehat{\Phi}^\top\}_t| \\
&= \frac{1}{n} \sum_{t=1}^{n} \mathrm{sign}(\{\widehat{\boldsymbol{\varphi}}_D(x_*)^\top I_{\boldsymbol{\alpha}} \widehat{\Phi}^\top\}_t) \nabla \{\widehat{\boldsymbol{\varphi}}_D(x_*)^\top I_{\boldsymbol{\alpha}} \widehat{\Phi}^\top\}_t \\
&= \frac{1}{n} \sum_{t=1}^{n} \mathrm{sign}(\{\widehat{\boldsymbol{\varphi}}_D(x_*)^\top I_{\boldsymbol{\alpha}} \widehat{\Phi}^\top\}_t) \widehat{\boldsymbol{\varphi}}_D(x_*)^\top \odot \{\widehat{\Phi}\}_t^\top.
\end{aligned}$$

In the last line, we have used the Hadamard product $\odot$, that is defined, for two vectors of length $D$ like $\widehat{\boldsymbol{\varphi}}_D(x_*)^\top$ and $\{\widehat{\Phi}\}_t^\top$, as the component-wise product, generating another vector of length $D$.

Now, we are going to bound the two-norm of this gradient:

$$\begin{aligned}
\|\nabla J(\boldsymbol{\alpha})\|_2^2 &= \sum_{i=1}^{D} \left\{ \frac{1}{n} \sum_{t=1}^{n} \mathrm{sign}(\{\widehat{\boldsymbol{\varphi}}_D(x_*)^\top I_{\boldsymbol{\alpha}} \widehat{\Phi}^\top\}_t) \widehat{\boldsymbol{\varphi}}_D(x_*)^\top \odot \{\widehat{\Phi}\}_t^\top \right\}_i^2 \\
&\leq \sum_{i=1}^{D} \frac{1}{n} \sum_{t=1}^{n} \left\{ \mathrm{sign}(\{\widehat{\boldsymbol{\varphi}}_D(x_*)^\top I_{\boldsymbol{\alpha}} \widehat{\Phi}^\top\}_t) \widehat{\boldsymbol{\varphi}}_D(x_*)^\top \odot \{\widehat{\Phi}\}_t^\top \right\}_i^2 \\
&\leq \sum_{i=1}^{D} \frac{1}{n} \sum_{t=1}^{n} \left\{ \widehat{\boldsymbol{\varphi}}_D(x_*)^\top \odot \{\widehat{\Phi}\}_t^\top \right\}_i^2 \\
&\leq \sum_{i=1}^{D} \frac{1}{n} \sum_{t=1}^{n} \widehat{\varphi}_i(x_*)^2 \widehat{\varphi}_i(x_t)^2 \\
&\leq \sum_{i=1}^{D} \widehat{\varphi}_i(x_*)^2 \underbrace{\frac{1}{n} \sum_{t=1}^{n} \widehat{\varphi}_i(x_t)^2}_{=1} = \widehat{\vartheta}_{D,2}^2,
\end{aligned}$$

where the last passage holds since the features $\widehat{\varphi}_i(\cdot)$ are orthonormal w.r.t. $\mu_n(\cdot)$. Under these assumption, namely

1. $J$ is convex

2. Each sub-gradient has norm bounded by $G := \vartheta_{D,2}^{(n)}$

3. The diameter of the optimization space $\mathcal{H}_d^D$ is $R := \sqrt{D-d}$

---

[4]if there are ties, the argument applied to each of them still holds bounding the norm of the sub-gradient

equation (3) on Boyd et al. [2003] guarantees that running the subgradient method for $I$ iterations with step size

$$\gamma_\ell = \frac{R}{G\sqrt{\ell+1}}$$

(corresponding to line 7), achieves suboptimality $\epsilon_I$ bounded by

$$\epsilon_I \leq \frac{R^2 + G^2 \sum_{\ell=1}^I \gamma_\ell^2}{2\sum_{\ell=1}^I \gamma_\ell} \leq \frac{R^2 + R^2(\log(I)+1)}{(R/G)\sqrt{I}} \leq \frac{2RG\log(I)}{\sqrt{I}} = \frac{2\vartheta_{D,2}^{(n)}\sqrt{D-d}\log(I)}{\sqrt{I}}.$$

Therefore, a number of iterations $I = 4\epsilon^{-2}\vartheta_{D,2}^{(n)2}(D-d)\log^3(4\vartheta_{D,2}^{(n)2}(D-d))$ allows to ensure $\epsilon_I \leq \epsilon$. In this way, we have

$$\widehat{\Lambda}_{\boldsymbol{\alpha}^*} - \inf_{\boldsymbol{\alpha}\in\mathcal{A}_d^D}\widehat{\Lambda}_{\boldsymbol{\alpha}} = J(\boldsymbol{\alpha}^{(I)}) - \inf_{\boldsymbol{\alpha}\in\mathcal{A}_d^D}J(\boldsymbol{\alpha}) \leq \epsilon_I \leq \epsilon,$$

which completes the proof. $\qquad\square$

**Theorem 11.** *Let Assumption 1 hold and fix $\delta > 0$. Then, with probability $1-\delta$,*

$$\mathcal{E}_\infty(\hat{\theta}_{n,BWR}) \leq (1 + \Lambda_\mu^{Oracle})\mathcal{E}_\infty(f) + \widetilde{\mathcal{O}}\left(\frac{\vartheta_{D,2}\sqrt{D\log(|\mathcal{X}|/\delta)}}{\sqrt{n}} + \frac{\vartheta_{D,2}^2\log(|\mathcal{X}|/\delta)}{n}\right).$$

*Proof.* By Theorem 7 and the definition of $\hat{\theta}_{n,\text{BWR}}$,

$$\mathcal{E}_\infty(\hat{\theta}_{n,\text{BWR}}) \leq (1 + \Lambda_{\boldsymbol{\alpha}^{(I)},\mu_n})\mathcal{E}_\infty(f) + \frac{\sigma\widehat{\varphi}_{2,D}\sqrt{2\log(2\mathcal{X}/\delta)}}{\sqrt{n}}. \tag{21}$$

By Theorem 10, for fixed $\epsilon$, we have $\Lambda_{\boldsymbol{\alpha}^{(I)},\mu_n} \leq \min_{\boldsymbol{\alpha}\in\mathcal{A}_d^D}\Lambda_{\boldsymbol{\alpha},\mu_n} + \epsilon$. Moreover, note that

$$
\begin{aligned}
\Lambda_\mu^{\text{Oracle}} &= \Lambda_{\boldsymbol{\alpha}_\mu^{\text{Oracle}},\mu} \\
&\geq \Lambda_{\boldsymbol{\alpha}_\mu^{\text{Oracle}},\mu_n} - \widetilde{\mathcal{O}}\left(\frac{\vartheta_{D,2}\sqrt{D\log(|\mathcal{X}|/\delta)}}{\sqrt{n}} + \frac{\vartheta_{D,2}^2\log(|\mathcal{X}|/\delta)}{n}\right) \\
&\geq \min_{\boldsymbol{\alpha}\in\mathcal{A}_d^D}\Lambda_{\boldsymbol{\alpha},\mu_n} - \widetilde{\mathcal{O}}\left(\frac{\vartheta_{D,2}\sqrt{D\log(|\mathcal{X}|/\delta)}}{\sqrt{n}} + \frac{\vartheta_{D,2}^2\log(|\mathcal{X}|/\delta)}{n}\right) \\
&\geq \Lambda_{\boldsymbol{\alpha}^{(I)},\mu_n} - \epsilon - \widetilde{\mathcal{O}}\left(\frac{\vartheta_{D,2}\sqrt{D\log(|\mathcal{X}|/\delta)}}{\sqrt{n}} + \frac{\vartheta_{D,2}^2\log(|\mathcal{X}|/\delta)}{n}\right).
\end{aligned}
$$

Replacing this relation in Eq. (21) we get the result.

$\qquad\square$

## D.4 Gradient method

The algorithm we use for our estimator is called Subgradient Method in the literature, and is presented in Algorithm 1.

# E Proofs of Section 5

**Theorem 12.** *Let $\mu$ be the uniform distribution on $[-1,1]$. There exists a constant $C > 0$ independent of $d$ such that, if we choose $D = 2d$ and $\boldsymbol{\varphi}_D(x) = [1,\ldots x^{2d-1}]$ as the augmented feature map for the target space spanned by $\boldsymbol{\varphi}_d(x) = [1,\ldots,x^{d-1}]^\top$, we have $\Lambda_\mu^{Oracle} \leq C$.*

---
**Algorithm 1** Subgradient Method
---

**Require:** Feature map $\varphi_D$, $d$, Number $I$ of iterations
**Ensure:** Sequence $\alpha^* \in \mathcal{A}_d^D$
 1: Compute $\widehat{\varphi}_D$ from $\varphi_D$ via Gram-Schmidt orthogonalization
 2: Define the following loss:
$$J(\alpha) = \|M(\alpha)\|_\infty$$

 3: Initialize $\alpha^{(0)} \leftarrow [\text{ones}(d), \text{zeros}(D-d)]^\top$
 4: **for** $\ell = 1$ to $I$ **do**
 5:     Compute step size $\gamma_\ell = \frac{\sqrt{D-d}}{\widehat{\varphi}_{2,d}\sqrt{\ell+1}}$
 6:     Compute a subgradient $g_\ell \in \partial J(\alpha^{(\ell-1)})$
 7:     Update: $\alpha^{(\ell)} = \alpha^{(\ell-1)} - \gamma_\ell g_\ell$
 8:     **if** $\alpha^{(\ell)} \notin \mathcal{A}_d^D$ **then**
 9:         Project: $h^{(\ell)} = \Pi_{\mathcal{H}_d^D} \alpha^{(\ell)}$
10:     **end if**
11: **end for**
12: **return** $\alpha^* = \alpha^{(I)}$

---

*Proof.* See Theorem 3.1 by Themistoclakis and Van Barel [2017]                    $\square$

**Proposition 23.** *Fix $\gamma > 0$. Let $\hat{\theta}_n$ be the OLS estimator, and $\hat{\theta}_{n,BWR}$ be our estimator defined in equation* (9). *There exists a function $f : [-1,1] \to \mathbb{R}$ such that, $\mathcal{E}_\infty(f) \to 0$ as $d \to \infty$, and under Assumption 1 with $\mu = \mathcal{U}([-1,1])$, the following hold with probability one:*

$$\lim_{d\to\infty} \lim_{n\to\infty} \|f(\cdot) - \varphi_d(\cdot)^\top \hat{\theta}_{n,BWR}\|_\infty = 0 \qquad \text{while} \qquad \lim_{n\to\infty} \|f(\cdot) - \varphi_d(\cdot)^\top \hat{\theta}_n\|_\infty \gtrsim d^{1-\gamma}.$$

Most of the proof of this proposition is about in building the function, that we are calling $f(\cdot)$.

The construction of the function in this proof is going to be quite involved. The function is going to be a sum over $n$ of terms of the form $\widetilde{f}_n(\cdot)$. The following notation will be used

1. Let $d_n$ dimension of the basis function used at step $n$

2. Let $a_n = d_n^{-\gamma}$, for a parameter $\gamma > 0$ to be defined

3. Let $h_n$ width of the mollifier

4. Let $M_n(\cdot) = M(\cdot/h_n)$, where $M(\cdot)$ is the standard mollifier, that is, a nonnegative function $M(\cdot) \in \mathcal{C}^\infty((-1,1))$ with integral one and compact support.

5. $f_n(\cdot) := \text{sgn}(\overline{\varphi}_{d_n}(\cdot)^\top \overline{\varphi}_{d_n}(x_n))$, where $x_n$ is such that
$$\|\overline{\varphi}_{d_n}(\cdot)^\top \overline{\varphi}_{d_n}(x_n)\|_{L^1} \geq \sup_{x \in (-1,1)} \|\overline{\varphi}_{d_n}(\cdot)^\top \overline{\varphi}_{d_n}(x)\|_{L^1} - 1.$$

6. $\widetilde{f}_n := f_n * M_n$

We are able to prove the following lemmas:

**Lemma 9.** *For every $n$,*
$$\|f_n - \widetilde{f}_n\|_{L^2} = \|f_n - f_n * M_n\|_{L^2} \leq 4\sqrt{h_n} d_n$$

*Proof.* In order to perform this proof, we need one result from mathematical analysis. In fact, call *bounded variation* a function $\mathcal{X} = (-1,1) \to \mathbb{R}$ such that the following norm is bounded

$$\|f\|_{BV} := \sup_{\{x_n\}_n \subset \mathcal{X}} \sum_n |f(x_{n+1}) - f(x_n)|.$$

A well-known characterization of this space Ambrosio et al. [2000] ensures that the former norm is equivalent to

$$\|f_n\|_{BV} \propto \|f\|_{L^1} + \|f'\|_{\mathcal{M}} \qquad \|f'\|_{\mathcal{M}} := \sup_{g \in \mathcal{C}^0(\mathcal{X}), \|g\|_\infty = 1} \int_\mathcal{X} g(x) f'(x) dx.$$

Now, we can proceed to the proof. First, note that by definition $f_n$ is in the $BV((-1, 1))$ class with $\|f_n\|_{BV} = \mathcal{O}(d_n)$. Indeed, $f_n(\cdot)$ takes only values in $\{-1, +1\}$, and can only jump between the two values when $\overline{\varphi}_{d_n}(\cdot)^\top \overline{\varphi}_{d_n}(x_n) = 0$, which happens at most $d_n$ times, as the previous is a polynomial of degree $d_n$. At this point, by the properties of convolution,

$$f_n(y) - f_n * M_n(y) = f_n * (M_n(y) - \delta(y))$$
$$= f'_n * \left( \int_{-1}^y M_n(t) - \delta(t) \, dt \right),$$

Where we have moved the derivative in the first term. At this point, the properties of convolution allow us to say that for any pair of functions $g_1, g_2$, $\|g_1 * g_2\|_{L^2} \leq \|g_1\|_{\mathcal{M}} \|g_2\|_{L^2}$. Therefore, we have

$$\|f_n(\cdot) - f_n * M_n(\cdot)\|_{L^2} \leq \|f'_n(\cdot)\|_{\mathcal{M}} \left\| \int_{-1}^y M_n(t) - \delta(t) \, dt \right\|_{L^2}$$
$$\leq \underbrace{\|f_n(\cdot)\|_{BV}}_{\leq d_n} \left\| \int_{-1}^y M_n(t) - \delta(t) \, dt \right\|_{L^2}.$$

At this point, note that by definition $M_n(t) \geq 0$, its integral is one and its support is contained in $(-h_n, h_n)$. Therefore,

$$\left| \int_{-1}^y M_n(t) - \delta(t) \, dt \right| \leq \begin{cases} 0 & y \geq h_n \\ 2 & -h_n < y < h_n \\ 0 & y \leq -h_n \end{cases},$$

so that its $L^2$ norm is bounded by $4\sqrt{h_n}$. This completes the proof.

$\square$

**Lemma 10.** *For every $m \leq n$, and $s > 0$*
$$\|\widetilde{f}_m - \Pi^\infty_{d_{n+1}, \infty} \widetilde{f}_m\|_\infty \leq \mathcal{O}(d_{n+1}^{-s} h_m^{-s}).$$

*Proof.* First, let us examine the smoothness of $\widetilde{f}_m$. Indeed, we have, for any $s > 0$

$$\|\widetilde{f}_m\|_{\mathcal{C}^s} = \|f_m * M_m\|_{\mathcal{C}^s}$$
$$\leq \|f_m\|_\infty \|M_m\|_{\mathcal{C}^s}$$
$$= \|M_m\|_{\mathcal{C}^s} = \mathcal{O}(h_m^{-s}).$$

Therefore, by Jackson's theorem, we have for any $s$,

$$\|\widetilde{f}_m - \Pi_{d_{n+1}, \infty} \widetilde{f}_m\| \leq \mathcal{O}(d_{n+1}^{-s} \|\widetilde{f}_m\|_{\mathcal{C}^s}) = \mathcal{O}(d_{n+1}^{-s} h_m^{-s}).$$

$\square$

**Theorem 24.** *For any $\gamma < 1/4$ there is $f^*$ such that*

- $\lim_d \|f^* - \Pi_{d,\infty} f^*\|_\infty = 0$

- $\limsup_d \frac{\|f^* - \Pi_{d,\mu} f^*\|_\infty}{d^{1-\gamma}} > 0.$

*Proof.* Let

$$f^*(\cdot) = \sum_{n=1}^{\infty} a_n \widetilde{f}_n(\cdot).$$

**First part**

Fix $\varepsilon > 0$. As $a_n$ goes to zero faster than exponentially and $\|\widetilde{f}_n(\cdot)\|_\infty \leq 1$, we can find $n_0$ such that

$$\left\| f^*(\cdot) - \sum_{n=1}^{n_0} a_n \widetilde{f}_n(\cdot) \right\|_\infty \leq \varepsilon/2.$$

Now, $\sum_{n=1}^{n_0} a_n \widetilde{f}_n(\cdot)$ is a finite sum of $C^\infty([-1,1])$ functions, so it is uniformly continuous, in particular. Therefore, by Stone-Weierstrass theorem, for sufficiently large $d$,

$$\left\| \sum_{n=1}^{n_0} a_n \widetilde{f}_n(\cdot) - \Pi_{d,\infty} \sum_{n=1}^{n_0} a_n \widetilde{f}_n(\cdot) \right\|_\infty \leq \varepsilon/2.$$

Putting the two results together, we have proved that, for sufficiently large $d$,

$$\|f^* - \Pi_{d,\infty} f^*\|_\infty \leq \left\| f^*(\cdot) - \Pi_{d,\infty} \sum_{n=1}^{n_0} a_n \widetilde{f}_n(\cdot) \right\|_\infty$$

$$\leq \varepsilon/2 + \left\| \sum_{n=1}^{n_0} a_n \widetilde{f}_n(\cdot) - \Pi_{d,\infty} \sum_{n=1}^{n_0} a_n \widetilde{f}_n(\cdot) \right\|_\infty$$

$$\leq \varepsilon.$$

**Second part** Let us fix $n = \ell$ and consider

$$\|\Pi_{d_\ell,\mu} f^*\|_\infty = \left\| \Pi_{d_\ell,\mu} \sum_{n=1}^{\infty} a_n \widetilde{f}_n(\cdot) \right\|_\infty$$

$$= \left\| \Pi_{d_\ell,\mu} \sum_{n=1}^{\ell-1} a_n \widetilde{f}_n(\cdot) + \Pi_{d_\ell,\mu} a_\ell \widetilde{f}_\ell(\cdot) + \Pi_{d_\ell,\mu} \sum_{n=\ell+1}^{\infty} a_n \widetilde{f}_n(\cdot) \right\|_\infty$$

$$\geq \underbrace{\left\| \Pi_{d_\ell,\mu} a_\ell \widetilde{f}_\ell(\cdot) \right\|_\infty}_{A} - \underbrace{\left\| \Pi_{d_\ell,\mu} \sum_{n=1}^{\ell-1} a_n \widetilde{f}_n(\cdot) \right\|_\infty}_{B} - \underbrace{\left\| \Pi_{d_\ell,\mu} \sum_{n=\ell+1}^{\infty} a_n \widetilde{f}_n(\cdot) \right\|_\infty}_{C}.$$

We are going to analyze the three terms separately.

(A) We start bounding the first term from below,

$$A = a_\ell \left\| \Pi_{d_\ell,\mu} \widetilde{f}_\ell(\cdot) \right\|_\infty$$

$$\geq a_\ell \|\Pi_{d_\ell,\mu} f_\ell(\cdot)\|_\infty - a_\ell \left\| \Pi_{d_\ell,\mu}(\widetilde{f}_\ell(\cdot) - f_\ell(\cdot)) \right\|_\infty$$

$$\geq \alpha_\ell \Lambda_{d_\ell,\mu} - \alpha_\ell \overline{\varphi}_{2,d_\ell} \|\mathbf{c}_{d_\ell}(\widetilde{f}_\ell(\cdot) - f_\ell(\cdot))\|_2$$

$$= \alpha_\ell \Lambda_{d_\ell,\mu} - \alpha_\ell \overline{\varphi}_{2,d_\ell} \|\widetilde{f}_\ell(\cdot) - f_\ell(\cdot)\|_{L^2}$$

$$\geq \alpha_\ell \Lambda_{d_\ell,\mu} - 4\alpha_\ell \overline{\varphi}_{2,d_\ell} d_\ell \sqrt{h_\ell}.$$

Here, the second inequality comes from Cauchy-Schwartz, the sequent equality from Parseval's theorem and the last comes from Lemma 9. Note that, for the polynomial basis, $\overline{\varphi}_{2,d_\ell} \approx \Lambda_{d_\ell,\mu} \approx d_\ell$, so we get

$$A \geq \Omega\left(\alpha_\ell d_\ell(1 - d_\ell\sqrt{h_\ell})\right)$$

(B) This term is

$$B = \left\|\Pi_{d_\ell,\mu} \sum_{n=1}^{\ell-1} a_n \widetilde{f}_n(\cdot)\right\|_\infty$$

$$\leq \sum_{n=1}^{\ell-1} a_n \left\|\Pi_{d_\ell,\mu} \widetilde{f}_n(\cdot)\right\|_\infty$$

$$\leq \sum_{n=1}^{\ell-1} a_n \left\|\Pi_{d_\ell,\mu}(\widetilde{f}_n(\cdot) - \Pi_{d_\ell,\infty}\widetilde{f}_n(\cdot))\right\|_\infty + a_n \left\|\Pi_{d_\ell,\infty}\widetilde{f}_n(\cdot)\right\|_\infty.$$

The last passage holds as $\Pi_{d_\ell,\mu}\Pi_{d_\ell,\infty}\widetilde{f}_n(\cdot) = \Pi_{d_\ell,\infty}\widetilde{f}_n(\cdot)$. Now, we can apply Lemma 10, as $n < \ell$, which ensures

$$B \leq \sum_{n=1}^{\ell-1} a_n \left\|\Pi_{d_\ell,\mu}(\widetilde{f}_n(\cdot) - \Pi_{d_\ell,\infty}\widetilde{f}_n(\cdot))\right\|_\infty + a_n \left\|\Pi_{d_\ell,\infty}\widetilde{f}_n(\cdot)\right\|_\infty$$

$$\leq \sum_{n=1}^{\ell-1} a_n \left\|\Pi_{d_\ell,\mu}(\widetilde{f}_n(\cdot) - \Pi_{d_\ell,\infty}\widetilde{f}_n(\cdot))\right\|_\infty + a_n \left\|\Pi_{d_\ell,\infty}\widetilde{f}_n(\cdot)\right\|_\infty$$

$$\leq \sum_{n=1}^{\ell-1} a_n \Lambda_\ell \left\|\widetilde{f}_n(\cdot) - \Pi_{d_\ell,\infty}\widetilde{f}_n(\cdot)\right\|_\infty + a_n \left\|\Pi_{d_\ell,\infty}\widetilde{f}_n(\cdot)\right\|_\infty$$

$$\leq \sum_{n=1}^{\ell-1} a_n d_\ell^{-s+1} h_n^{-s} + a_n \left\|\Pi_{d_\ell,\infty}\widetilde{f}_n(\cdot)\right\|_\infty.$$

(C) The last term can be simply bounded due to the fact that $\|\widetilde{f}_n\|_\infty \leq 1$:

$$C \leq d_\ell \sum_{n=\ell+1}^{\infty} a_n.$$

Now, fix any $\gamma < 1/4$ and take

$$s = 2; \quad d_n = \exp(1/\gamma^n); \quad h_n = \exp(-1/(2\gamma^{n+1})); \quad a_n = \exp(-1/\gamma^{n-1}).$$

We get

$$A \geq \Omega\left(\alpha_\ell d_\ell(1 - d_\ell\sqrt{h_\ell})\right)$$

$$\geq \Omega\left(\exp((1-\gamma)/\gamma^\ell)(1 - \exp(1/\gamma^n - 1/(4\gamma^{n+1})))\right)$$

$$\geq \Omega\left(\exp((1-\gamma)/\gamma^\ell)(1 - \exp(1/\gamma^n \underbrace{(1 - 1/(4\gamma))}_{\leq 0}))\right)$$

$$\geq \Omega\left(\exp((1-\gamma)/\gamma^\ell)\right) = \Omega(d_\ell^{1-\gamma}).$$

For term $B$, we have

$$
\begin{aligned}
B &\leq \mathcal{O}\left(\sum_{n=1}^{\ell-1} a_n d_\ell^{-s+1} h_n^{-s} + a_n \left\|\Pi_{d_\ell,\infty} \widetilde{f}_n(\cdot)\right\|_\infty\right) \\
&\leq \mathcal{O}\left(\sum_{n=1}^{\ell-1} a_n d_\ell^{-s+1} h_n^{-s} + a_n\right) \\
&\leq \mathcal{O}\left(\sum_{n=1}^{\ell-1} a_n \exp((-s+1)/\gamma^\ell) \exp(s/(2\gamma^{n+1})) + a_n\right) \\
&\leq \mathcal{O}\left(\sum_{n=1}^{\ell-1} a_n \exp((-s+1)/\gamma^\ell) \exp(s/(2\gamma^\ell)) + a_n\right) \\
&\leq \mathcal{O}\left(\sum_{n=1}^{\ell-1} a_n \underbrace{\exp((-s/2+1)/\gamma^\ell)}_{\leq 1} + a_n\right).
\end{aligned}
$$

Last term:

$$
C \leq \mathcal{O}\left(d_\ell \sum_{n=\ell+1}^\infty a_n\right) \leq \mathcal{O}\left(\exp(1/\gamma^\ell) \sum_{n=\ell+1}^\infty \exp(-1/\gamma^{n-1})\right) = \mathcal{O}\left(\sum_{m=0}^\infty \exp(-1/\gamma^m)\right).
$$

Again, this term satisfies $C = \mathcal{O}(1)$, as the term $\exp(-1/\gamma^m)$ in the last sum decays faster than $2^{-m}$. $\qquad\square$

where the last passage holds as $s = 2$. Therefore we get $B \leq \mathcal{O}(\sum_{n=1}^{\ell-1} a_n) = \mathcal{O}(1)$, since $a_n$ decays faster than $2^{-n}$ which already generates a convergent sequeuence.

All together, these passages prove

$$
\|\Pi_{d_n,\mu} f^*\|_\infty \geq \Omega(d_n^{1-\gamma}).
$$

Therefore, taking this $d_n$ sequence entails $\limsup_{d\to\infty} \frac{\|f^* - \Pi_{d,\mu} f^*\|_\infty}{d^{1-\gamma}} > 0$.

*Proof. (of Proposition 13).* Let $f = f^*$ defined before, for the specific value of $\gamma > 0$. Thanks to part one of Theorem 24[5], assumption $\mathcal{E}_\infty(f) \xrightarrow{d} 0$ is satisfied:

$$
\mathcal{E}_\infty(f) = \|f^* - \Pi_{d,\infty} f^*\|_\infty \xrightarrow{d} 0.
$$

Then, we prove the two theses point by point. Point one: for fixed $d$, Theorem 11 gives

$$
\|f(\cdot) - \varphi_d(\cdot)^\top \hat{\theta}_{n,\mathrm{BWR}}\|_\infty \leq (1+\Lambda_\mu^{\mathrm{Oracle}})\mathcal{E}_\infty(f) + \widetilde{\mathcal{O}}\left(\frac{\vartheta_{D,2}\sqrt{D\log(|\mathcal{X}|/\delta)}}{\sqrt{n}} + \frac{\vartheta_{D,2}{}^2 \log(|\mathcal{X}|/\delta)}{n}\right).
$$

As $\mathcal{X}$ is $[-1,1]$ and the feature map is Lipschitz continuous, we can get rid of the $|\mathcal{X}|$ by a covering argument. As $n \to \infty$, the former gives

$$
\lim_n \|f(\cdot) - \varphi_d(\cdot)^\top \hat{\theta}_{n,\mathrm{BWR}}\|_\infty \leq (1 + \Lambda_\mu^{\mathrm{Oracle}})\mathcal{E}_\infty(f).
$$

---

[5]formally, the result holds for $\gamma > 1/4$ but, for what we are trying to prove, the validity of the statement for $\gamma$ implies its validity for every $\gamma' > \gamma$, therefore we can proceed w.l.o.g.

For $\mu = \mathcal{U}([-1, 1])$, Theorem 12 ensured that $\Lambda_\mu^{\text{Oracle}} < C$, a universal constant independent on $d$. Therefore,

$$\lim_d \lim_n \|f(\cdot) - \varphi_d(\cdot)^\top \hat{\theta}_{n,\text{BWR}}\|_\infty \leq \lim_n (1 + C)\mathcal{E}_\infty(f) = 0.$$

Let us pass to the second thesis:

$$\lim_{n \to \infty} \|f(\cdot) - \varphi_d(\cdot)^\top \hat{\theta}_{n,\text{OLS}}\|_\infty \gtrsim d^{1-\gamma}.$$

This follows from the fact that, for $n \to \infty$, $\varphi_d(\cdot)^\top \hat{\theta}_{n,\text{OLS}} \to \Pi_{d,\mu} f(\cdot)$ and that Theorem 24 ensures $\limsup_d \frac{\|f^* - \Pi_{d,\mu} f^*\|_\infty}{d^{1-\gamma}} > 0$.

$\square$

