# OpenReview forum: "Beyond Least Squares: Uniform Approximation and the Hidden Cost of Misspecification"
_NeurIPS.cc/2025/Conference — NeurIPS 2025 poster_

### Official Review · Reviewer_ZEuE · 2025-06-29

**Clarity:** 4
**Significance:** 3
**Originality:** 3
**Rating:** 5
**Confidence:** 3

**Summary:**

This paper addresses the problem of uniform error in linear regression under model misspecification. The authors focus on the random design setting and aim to control the worst-case (uniform norm) error, rather than the $L_2$ error. The authors show that the amplification of misspecification error in least-squares regression is governed by the Lebesgue constant, a classical notion from approximation theory that quantifies how poorly least-squares can behave in the worst case. They prove that this amplification is intrinsic and cannot be avoided by standard techniques such as ridge regression. To address this, the authors propose a novel estimator—Best Weighted Regularization (BWR)—which augments the feature space with auxiliary functions and optimizes a weighted projection to minimize the uniform error. The proposed method achieves an oracle-type performance guarantee and reduces the worst-case error, particularly in settings with polynomial bases. The theoretical results are complemented by synthetic experiments that clearly illustrate the improvement over standard least-squares methods.

**Questions:**

Generally, I think the paper answers most of my questions when I am reading. Here are some minor questions and comments:

It is better if the paper can give a short introduction  to $\lambda$-ridge regression in the main paper, and the exact definition of $\hat{\theta}_{n,\textup{RIDGE}}$.

The display of Eq.(4) and Eq.(5) is a bit controversial, especially on $\alpha_i$.

Can you complete answering \textbf{Q1} by combining Theorem 9 and Proposition 10?

Can you provide more examples besides the polynomial basis to show that the oracle Lebesgue constant has a better denepndence (or even independence) on the dimension $d$?

 So far the paper discuss the dependence on the dimension $d$. What is the dependence on $\mu$? The paper mainly focus on the uniform distribution. Is this a standard assumption or we can investigate more distributions?

**Ethical Concerns:**

["NO or VERY MINOR ethics concerns only"]

**Final Justification:**

Thank the authors for addressing my questions. I will maintain my score and I believe this is a solid and informative paper to accept

**Limitations:**

yes

**Paper Formatting Concerns:**

In Algorithm 1, the second step. What does the equation (??) refer to?

**Quality:**

3

**Strengths And Weaknesses:**

Overall, this is a strong paper, with solid theoretical contribution as well as the story flow and readability. A small problem is that the main body excess the 9-page limit in the appendix version.

Lebesgue Constant as Amplification Factor: The authors demonstrate that the amplification of the misspecification error under least-squares regression is governed by the Lebesgue constant, a classical concept from approximation theory. They prove this characterization is tight, revealing an intrinsic limitation of ordinary least-squares (OLS), even with infinite data.

Negative Result for Ridge Regression: The paper shows that standard ridge regression fails to mitigate this issue and can still suffer from large uniform errors, despite its regularization effect in other settings.

Theoretical Guarantees: The paper proposes a new method called \emph{Best Weighted Regularization (BWR)} that augments the feature space and optimally attenuates auxiliary components. The estimator provably achieves the optimal Lebesgue constant and thus uniform error..

Case Study -- In the polynomial feature case, the proposed method illustrates reducing the amplification factor from $\Omega(d)$ to $O(1)$, supported by explicit error bounds. The example gives readers a clearer sense of the contribution of the BWR estimator.

---

> ### Author Rebuttal · Authors · 2025-07-30
>
> Thank you for the helpful comments. Your observations are valid, and we will revise the paper accordingly: we will add a brief introduction to ridge regression and clarify the definition of its estimator, as well as improve the presentation of Eqs.(4) and (5).
>
> On the remaining questions:
>
> *Can you complete answering **Q1** by combining Theorem 9 and Proposition 10?* Yes, Theorem 9 and Proposition 10 together provide a complete answer to **Q1**. We will make this explicit in the final version.
>
> *.. more examples besides the polynomial basis ..  better denepndence on $d$?* Yes, for the Fourier basis, the same algorithm reduces the Lebesgue constant from $O(\log(d))$ to $O(1)$. In the multidimensional case (domain is $N$ dimensional), this grows to $O(\sqrt{d})$ for the so-called spherical summation domain where one controls the multi-dimensional frequencies $k$ involved by their $2$-norm. We will be happy to add these. Interestingly, we are not aware of any characterization of the Lebesgue constant of, say, RBF networks, but we expect bad news (as the well-known ill-conditioned property of RBFs is closely related to exploding approximation errors). While large Lebesgue constants are not frequently calculated, we expect that they may appear commonly even if their values are not known.
>
> *Dependence on $\mu$* Note that all the results of the paper, except for Proposition 15 are valid of all choices of $\mu$. It is true that the bounds of the Lebesgue constant in Table 1 only focus on the uniform distribution. The reasons are mostly historical, as the result come from the Fourier analysis or approximation theory literature. We would also note that Proposition 6 allows to generalize all the results for the uniform distribution to any other distribution such that the density ratio is bounded. When the density is unbounded (the distribution has partial support), the uniform approximation error will be uncontrolled.

---

> > ### Comment · Reviewer_ZEuE · 2025-08-06
> > **Thank you**
> >
> > Thank the authors for addressing my questions. I will maintain my score and I believe this is a solid and informative paper.

---

### Official Review · Reviewer_c8kL · 2025-07-01

**Clarity:** 3
**Significance:** 2
**Originality:** 4
**Rating:** 4
**Confidence:** 4

**Summary:**

The paper investigates uniform‐norm generalization error in misspecified linear regression under random design. Its first contribution is to identify the Lebesgue constant of the $L^2(\mu)$-projection onto the feature span as the exact amplification factor by which ordinary least‑squares (OLS) magnifies the misspecification bias.  The authors prove matching upper and lower bounds by showing that no estimator that minimizes empirical squared error can beat this factor in the worst case. The second contribution is a feature‑augmentation and weighted ridge setting. By adding auxiliary basis functions and assigning individual ridge penalties, the learner can change the projection operator itself.  The paper shows an oracle inequality comparing the proposed estimator to the best attenuation vector that minimizes the Lebesgue constant, and a convex sub‑gradient algorithm that provably converges to this optimum using only data. They also demonstrate this in the special case of monomial features on $[-1,1]$, where the method turns the $\Omega(d)$ error amplification by OLS into $O(1)$.

**Questions:**

- Could you share any useful insights from your theory that add to existing perspectives on how the way we design models and feature representations to achieve better generalization? I realize this is somewhat open‑ended, but I would appreciate hearing the authors’ thoughts on it.

This is a minor point but I guess you meant to write $E [exp(\lambda \eta_t)]$ in line 61, not just $\eta_1$, and f is missing in $\varepsilon_{\infty}(\hat{\theta}_{n, \text{OLS}})$ in Theorem 5 statement. I don’t have additional questions aside from the above, please refer to the weaknesses I stated in the previous part.

**Ethical Concerns:**

["NO or VERY MINOR ethics concerns only"]

**Final Justification:**

I still believe that the paper can benefit from more experiments, especially using real data. This is not fully addressed by the authors, and the paper does not well motivate and connect to problems in the RL literature.

**Limitations:**

The submission does not contain a limitations/impact section; they should have discussed at minimum the computational cost overhead originating from the Gram-Schmidt orthonormalization.

**Paper Formatting Concerns:**

I realized that in the supplementary material, the length of the main text in the uploaded PDF exceeds 9 pages. I wanted to bring this to your attention in case it presents any issues.

**Quality:**

3

**Strengths And Weaknesses:**

**Strengths:**

- The paper presents rigorous theory and proves tightly matched upper/lower bounds.

- The amount of the novelty is sufficient, as the paper brings a new perspective on reframing misspecification amplification as a geometric property that can be learned and controlled.
- The paper brings a thorough, well‑motivated discussion of the Lebesgue constant, which helps the reader grasp the intuition behind the results.

**Weaknesses:**

- Only one synthetic experiment; no real data or RL experiment to validate practicality, despite that being the stated motivation, which makes it hard to judge the significance of the paper. Also, there's no discussion about the experiment in Figure 1 or its methodological choices.

- Narrative is a bit hard to follow due to excessive expositions, and there is some overlapping content, such as Theorems 4 and 5, or ridgeless and ridge least squares sections, which could have been written more briefly. I also believe work can be done to improve the grammar (e.g., in the abstract) and presentation in several places.

- The uniform error is rarely encountered in machine‑learning tasks, and the paper needs a stronger conceptual argument for why the theoretical guarantees matter in realistic settings. Although briefly mentioned in the intro, the paper would benefit from further clarifying where uniform error influences/determines performance and how theoretical insights translate to practical scenarios like RL.

---

> ### Author Rebuttal · Authors · 2025-07-30
>
> We thank the reviewer for their comments and the time dedicated to our paper.
>
> ### On the mentioned weaknesses
> **W1a** *To judge significance, given motivation, more empirical work is needed.*. Our ultimate motivation is indeed to make an impact on the design of RL algorithms. We reject the notion that the significance of this work can only be judged through empirical results. Significance can also be judged based on whether a work makes meaningful contribution to the theory of relevant methods. We insist that our work falls into this category, which, according to the call-for-papers is well within the scope of NeurIPS. As such, we ask the reviewer to reconsider their position and look at the merits of the paper on its own terms. For this, we would like to note the following (copied from our response to W3 of reviewer bUTv): "We also would like to emphasize that we think that even with the limited experiments the paper could be of great interest to the RL theory community, which, since the seminal paper of Du et al. (ICLR 2020) was struck by the price one may need to pay for misspecification. Our paper not only significantly refines the "dark picture" painted by the paper of Du et al. (namely, that polynomial sample complexity is only possible when the misspecification error is blown up by a factor of $\sqrt{d}$), in that we connect the blow-up factor to the Lebesgue constant associated with the learning setting, which, as we illustrate with examples can be much smaller than $O(\sqrt{d})$, but we also suggest a way of avoiding the blow-up with a novel technique."
>
> **W1b** *no discussion about the experiment in Figure 1 or its methodological choices* The covariates are uniformly distributed, the standard polynomial basis is used as described in the caption. The target function (as is visible on Figure 1) is a near-periodic piecewise constant function. We aimed for a moderately challenging target function which is piecsewise smooth, as is common in optimal control/RL (think of Mountain-Car, optimal value function) and we set the sample size high enough relative to the frequency of the "slow" changes in the function so that the estimators have a chance to get a good estimate. Significant Gaussian was added to increase difficulty and illustrate robustness of the methods to noise.
> We will add this description with precise detailed to the appendix. Our excuse for omitting this description was that we thought that all these choices are perhaps self-evident from the figure, but we realize it is better to include these details. Thanks for raising this issue, which we hope with this we have satisfactorily addressed.
>
> **W2** *Narrative, presentation, grammar could be improved.* Thanks for the constructive feedback which we will take into account in the revision of our paper.
>
> **W3** *uniform error is rarely encountered in machine‑learning tasks, relevance to RL* The papers of Du et al. (ICLR 2020) and Lattimore et al. (2020) established that controlling the uniform error is necessary in RL. The heart of this argument is as follows: in these tasks, ultimately one needs to choose an action with a high value and one is evaluated by the difference between the value of the best action and that of the action chosen. Ultimately, the algorithms thus need to produce value estimates. Even if the optimal action's value is perfectly known, if *some* suboptimal action's value is overestimated by a large quantity, the resulting value loss can be high. Clearly, it is not sufficient here to have small errors over the action-values on average as this is not sufficient to control the value loss. We have not included this discussion in the paper since it has been presented (formally and in detail) in multiple prior works, but if the reviewer thinks that it would be useful to present this argument, we will try to find a way to add it. See also our response to Reviewer KqBU Q1 for how our bounds specifically help in the RL context.
>
> **Q:** *Could you share any useful insights from your theory that add to existing perspectives on how the way we design models and feature representations to achieve better generalization?*
> The key takeaway for machine learning applications is summarized in three buller points:
> 1) Uniform error, not average error, is the real bottleneck in data-driven optimization, and it is highly sensitive to the choice of features and the sampling distribution.
> 2) When it comes to approximating smooth (incuding piecewise smooth) functions, local and/or periodic features (e.g., wavelets, Fourier) are vastly better than for example the standard polynomial basis.
> 3) The amplification factor can grow with model size: In this case running least-squares methods may not control uniform errors at all. Changing features can help with this.
> 4) Nontrivial regularization methods such as our BWR methods can "tame" even bad feature maps. The design of these methods should be governed by the need to control the uniform approximation error.

---

### Official Review · Reviewer_KqBU · 2025-07-03

**Clarity:** 3
**Significance:** 3
**Originality:** 4
**Rating:** 5
**Confidence:** 4

**Summary:**

Motivated by the error ampification in RL with linear function approximation, the paper studies $L^\infty$ oracle inequalities for least-square linear regression problems. The paper identifies ``Lebesgue constant'' -- the $L^\infty$-operator norm of the $L^2$ orthonormal projection operator -- as a key quantity governing performance of estimators.

The paper shows the following result
1. For the OLS estimator, the paper proves an upper bound with an approximation error term depending on the Lebesgue constant and the $L^\infty$ oracle error, as well as a statistical error term that depends also on the log-cardinality of the data domain.
2. The paper analyzes the Lebesgue constant by showing upper and lower bounds on some specific choices of basis functions.
3. Moving beyond OLS, the paper studies regularized and weighted versions of the least-square estimators, with approximation/statistical error guarantees. In particular, the approximation factor now depends on the $L^\infty$-operator of a weighted version of the orthonormal projection operator. The paper further studies optimization procedures that choose the weight to minimize the approximation factor.

**Questions:**

I think the paper could be significantly strengthened from the following aspects:
1. Concrete applications to RL. As the paper motivates the study of $L^\infty$ oracle inequalities from value learning with function approximation, it would be beneficial to show concrete results to these problems. For example, does the weighted estimator lead to an RL algorithm with better theoretical guarantees?
2. The role of Lebesgue constant. It is helpful to establish lower bounds showing the necessity of Lebesgue constant, at least for OLS estimator. The current Theorem 3 is just an operator norm characterization.
3. The statistical error term depends on the cardinality of data domain, which may be infinite in practice. In principle, it should depend on the metric entropy in the infinite case. Also, the bound may not be tight, as it involves a high-order polynomial dependence on dimension.

**Ethical Concerns:**

["NO or VERY MINOR ethics concerns only"]

**Final Justification:**

Though there are certain limitations (e.g. consequences to RL, lower bounds, etc), the paper already made considerable contribution given the length limit of a conference paper. The author's response also clarify certain technical details. I think the paper should be accepted and therefore I decide to raise my score.

**Limitations:**

The paper has some limitations in terms of the scope and the optimality of the results. There are no potential negative societal impact.

**Paper Formatting Concerns:**

No formatting concerns.

**Quality:**

3

**Strengths And Weaknesses:**

Strength: The paper presents interesting results that are relavant to a well-known limitation in RL with function approximations. Though the oracle error amplification is already observed in literature, it is nice to see a comprehensive study on this. Moreover, the minimization procedure for the approximation factor is novel and interesting. The paper also illustrates a concrete improvement using polynomial basis.

Weakness: The results are relatively limited to specific setups, and some bounds seem weak. I will explain more in questions section below.

---

> ### Author Rebuttal · Authors · 2025-07-30
>
> We thank the reviewer for their comments and the time dedicated to our paper.
>
> **Q1** *Applications to RL*. In a large number of RL papers [e.g., 1-6], the algorithms estimate action-value functions based on least squares. These papers all rely on using least-squares approaches based on rollouts to estimate value functions and controlling uniform approximation errors (as noted in our paper, this is known to be unvoidable by the results proven in [1]). In any of these papers an error term of size $C \epsilon$ appears in the final suboptimality bounds, where $\epsilon$ is the uniform approximation error, $C$ is an "error amplification constant". The constant $C$ is either uncontrolled, or is of order $\sqrt{d}$. By our results, $C$ in these results can be replaced by an appropriate Lebesgue constant without changing anything else. If OLS is replaced by BWR, the constant $C$ can be further reduced (at the price of a modest increase of the estimation error term, which will also result in a modest increase of the corresponding term in the final error bounds). We are happy to include a discussion of this (in addition to the existing discussion) if the reviewer feels this would help. However, we also have to be conscious of that including full details would require introducing the settings of the works, which will not only take up a nontrivial amount of space, but may also distract the reader from the main message of the paper.
>
>
> **Q2** *Role of Lebesgue constant/lower bound for OLS.* As $n\to\infty$, the OLS estimator will return $\Pi_{d,\mu} f$. As such, Theorem 3 provides an asymptotic lower bound in terms of the Lebesgue constant. Combining the ideas in Theorem 3 (i.e., choice of worst-case target function) with finite-sample instance specific lower bound techniques should give rise to a finite-time lower bound. A back-of-the-envelope asymptotic calculation based on the central limit theorem can be used to give a bound that takes the expected form (essentially matching our upper bound). Deriving any of these results will require considerable work, which we leave for future work. We feel, however, that even without providing this lower bound, the reader will agree that it is very likely that even for the finite-sample case the Lebesgue constant should control the error.
> (We find it intriguing that no known lower bounds for OLS that would be strong enough to build on appear to be available in the literature. The closest/strongest results seem to appear in a surprisingly recent paper [7].)
>
>
> **Q3** *Cardinality of data domain and higher order dimension terms*. The reviewer’s intuition is essentially correct, and in the most common settings we can rely on a tool even simpler than metric entropy. The bound extends to infinite index sets through a standard covering argument: as noted on line 105, it grows only logarithmically with the Lipschitz constant of the feature map. Moreover, note that in many optimization problems, which are the motivation for the paper, the optimizer is allowed to work only on a finite and relatively small subset of $\mathcal X$, so that we need a good estimation only in those points.
>
> The higher-order polynomial dependence on the dimension appears in a low-order, or "fast" term, and as such, we made no effort to optimize this term. It may be interesting technical work to try to reduce the magnitude of this term. We also note in passing that the same term does not appear for the fixed-design case. We suspect, some cost needs to be paid for working with a random-design.
>
> [1] Lattimore et al. Learning with good feature representations
> in bandits and in RL with a generative model. ICML, 2020.
>
> [2] Antos et al. Fitted Q-iteration in continuous action-space
> MDPs. In Advances in neural information processing systems, 2008.
>
> [3] Weisz et al. Confident Approximate Policy Iteration for Efficient Local Planning in $q^\pi$-realizable MDPs. NeurIPS, 2022.
>
> [4] Hao et al. Confident Least Square Value Iteration with Local Access to a Simulator. AISTATS, 2022.
>
> [5] Yin et al. Efficient local planning with linear function approximation. ALT, 2022.
>
> [6] Lazic et al. Improved Regret Bound and Experience Replay in Regularized Policy Iteration. ICML, 2021.
>
> [7] Mourtada: Exact minimax risk for linear least squares, and the lower tail of sample covariance matrices. Annals of Statistics, 2022.

---

> > ### Comment · Reviewer_KqBU · 2025-08-07
> >
> > Thank the authors for writing the response. I think the results are novel and already sufficient for a conference paper. The rebuttal also clarified some aspects of the results. So I decide to raise my score.

---

### Official Review · Reviewer_UvJD · 2025-07-06

**Clarity:** 4
**Significance:** 4
**Originality:** 4
**Rating:** 6
**Confidence:** 3

**Summary:**

The paper studies the problem of controlling the worst-case error in misspecified linear regression. It is shown that for least squares methods, the constant in front of the misspecification error is determined by the Lebesgue constant, which itself depends on both the feature map and the distribution from which the covariates are drawn. Bounds on the Lebesgue constant for polynomial, Fourier and wavelet bases (as well as some others) are presented. Based on the insights that: 1) amplification of the misspecification error is determined by the Lebesgue constant; 2) the Lebesgue constant can be reduced by adding features, the paper proposes a new regularised estimator called Best Weighted Regulariser (BWR), which utilises an extended feature map. It is shown that, for a given extended feature map, the maximum error of the BWR estimator asymptotically matches that of the (a priori unknown) oracle regularised estimator. Finally, there is a case study of misspecified linear regression with monomial features, in which BWR achieves significantly lower maximum error than Ordinary Least Squares.

**Questions:**

Is it possible to give general rules of thumb for how many features should be added or which features should be added?

How does the BWR method compare to kernel (ridge) regression (with a kernel function that can be expressed a weighted sum of basis functions)? Could a version of the BWR method be used to learn a kernel function (e.g. by representing the kernel function as a weighted sum of basis functions, and then fitting these weights)?

Since the error bound for BWR depends on $D$ instead of $d$, should one expect BWR to perform worse than OLS when the sample size is small relative to $D$ (and the misspecification is sufficiently small)? If so, do the error bounds in the paper give an indication of how large $n$ and/or the misspecification level has to be before BWR should outperform OLS?

The following is a list of typos:
- In the definition of the Lebesgue constant, there seem to be some extra brackets around $\|P\|_{\infty}$
- I presume that the factor on the RHS of Equation (2) should be $1 + \Lambda$ rather than $\Lambda$
- Line 95: should $\hat{\theta}_n$ be $\hat{\theta}_{n,OLS}$?
- Line 99: “n positive integer” to “positive integer n”
- Line 150: “to under we have to resort of” to “to understand them we have to resort to”
- Line 151: “we enumerate few” to “we enumerate a few”
- Line 191: “this does to corresponds to” to “this does not correspond to”
- Line 233: Gram-Schmidt typo
- Eqn (7): lower case $d$ should be upper case $D$
- Statement of Thm. 5: $\mathcal{X}$ should be $|\mathcal{X}|$
- Statement of Thm. 9 $\mathcal{X}$ should be $|\mathcal{X}|$

**Ethical Concerns:**

["NO or VERY MINOR ethics concerns only"]

**Final Justification:**

The paper studies an interesting problem, it provides an effective solution to the problem and is both insightful and well-written. I feel strongly that this paper should be accepted.

**Limitations:**

Yes.

**Paper Formatting Concerns:**

None.

**Quality:**

4

**Strengths And Weaknesses:**

**Strengths:**
The paper is very well-written and the results of the paper are presented very well.

The problem of misspecified linear regression is interesting and the paper offers several nice insights about this problem and the consequences for methods (such as value function approximation) that use linear regression as a core component. For instance, the fact that the Lebesgue constants of the Fourier basis and localised basis functions are relatively small offers one explanation for the observation that these bases are observed to work well in reinforcement learning problems.

The proposed BWR method appears to be an effective solution to the problems raised in earlier sections of the paper. In particular, it achieves the goal of asymptotically matching the error bound of the oracle regularised estimator and outperforms OLS in terms of maximum error in the case study.

**Weaknesses:**
I don’t think the paper has any major weaknesses (though I did not check the proofs carefully).

It isn’t clear to me that the theoretical results of the paper lead easily to general rules of thumb for how many features one should add or which features should be added for best results.

---

> ### Author Rebuttal · Authors · 2025-07-30
>
> Thanks for the supportive and insightful review and interesting questions. We will address all the questions and fix all the typos listed in the final version of the manuscript provided the paper gets accepted.
>
> **Q1** *Is it possible to give general rules of thumb for how many features should be added or which features should be added?*
> We think that using $D\approx d$, i.e., doubling the number of features should work well in many cases. Indeed, we can confirm that this works when the features come from the Fourier and polynomial basis functions (e.g., Theorem 14). More generally, we expect this to work as long as approximation errors (of continuous functions) decay at a polynomial rate. This fact is stated, even if somehow hidden, in the works that we cite about De La Valleè Poussin methods, and directly translates to our approach (which is stronger since it is automatically able to "guess" the oracle weights).
>
> **Q2** *How does the BWR method compare to kernel (ridge) regression (with a kernel function that can be expressed a weighted sum of basis functions)? Could a version of the BWR method be used to learn a kernel function (e.g. by representing the kernel function as a weighted sum of basis functions, and then fitting these weights)?*
> Indeed, all our metods work with the projection operator, which is indeed the convolution with a kernel. Our approach in deriving the estimator BWR may be seen as a way to find the optimal regularization of the kernel from empirical data, in order to compete with the performance of an oracle, as the reviewer suggests. The exact details of this remain challenging as one needs to address a number of statistical and computational issues that arise. We are nevertheless optimistic that the method can be extended to this case.
>
> **Q3** *Since the error bound for BWR depends on $D$ instead of $d$, should one expect BWR to perform worse than OLS when the sample size is small relative to $D$ (and the misspecification is sufficiently small)? If so, do the error bounds in the paper give an indication of how large and/or the misspecification level has to be before BWR should outperform OLS?*
>
> If we allow ourselves to compare upper bounds, we can conclude the following: The terms in play are the estimation error (EE), and the *amplified* misspecification error (AME).
> BWR increases EE by a factor of
>
> $\phi_{2,D}/\phi_{2,d}$ (dropping overline due to limited markup), while it decreases the AME. In particular, the misspecification error is not increased, while the amplification factor may be drastically reduced. To get a simple answer, ignoring constants, higher order terms and log factors, assuming $\phi_{2,d}$
> $ = \sqrt{d}$,
> we find that EE for OLS is
> $\sqrt{d/n}$,
> and it is $\sqrt{D/n}$ for BWR.
>
> We also find that AME for OLS is $(1+\Lambda_{d}) \epsilon_d$, where $\Lambda_d$ is the Lebesgue constant, $\epsilon_d$ is the approximation error when using $d$ basis functions. We also find that AME for BWR is $\epsilon_d$ (simplified best case; see eg Theorem 14). Solving $\sqrt{D/n}+ \epsilon_d \le \sqrt{d/n}+(1+\Lambda_d) \epsilon_d$, we find that BWR "wins" when
>
> $$n \ge  d (\sqrt{D/d}-1)^2/(\Lambda_d \epsilon_d)^2.$$
>
> Now, if we choose $D=2d$ (cf. Section 5), the condition, ignoring constants, becomes $n\ge d/(\Lambda_d \epsilon_d)^2$. Here, what is important to realize is that as $d\to\infty$, $\Lambda_d \epsilon_d$ may *not* converge to zero, as is the case for the polynomial basis functions for target functions for which $\epsilon_d = \Omega(1/d)$. For cases like this, the condition essentially becomes $n = \Omega(d)$. The worst-case for our method is when $\Lambda_d =1$. In this case, the threshold for $n$ is $d/\epsilon_d^2$.
>
> However, perhaps a more charitable view of the method, which we hope will be appealing to many, is that with $D=2d$, it never loses only minimally compared to OLS (by a factor of $\sqrt{2}$ on the EE), while in cases of poor basis functions, it may gain much. In this view, BWR acts as an insurance strategy where potentially very large errors are drastically reduced while errors are never increased by much compared to OLS.

---

> > ### Comment · Reviewer_UvJD · 2025-08-04
> >
> > Thank you for your responses to my questions. I will raise my score. I believe that this paper should absolutely be accepted.

---

### Official Review · Reviewer_bUTv · 2025-07-11

**Clarity:** 3
**Significance:** 2
**Originality:** 3
**Rating:** 3
**Confidence:** 3

**Summary:**

This paper investigates the problem of uniform error control of ordinary least-squares (OLS) regression in the context of mis-specified linear model — where the true function lies outside the span of the chosen feature set. The task is motivated by applications in value function approximation for bandit algorithms and reinforcement learning (RL), where OLS regression is a standard tool but can suffer from significant error amplification under model mis-specification. The authors provide a sharp characterization of how the amplification of the mis-specification error depends on the interaction between the sampling distribution and the feature subspace, which is governed by the “Lebesgue constant” —  a classical quantity in approximation theory. Moreover, they prove that the dependence on the Lebesgue constant is shown to be tight — no estimators based on empirical squared-loss minimization can substantially improve upon this bound in general. Motivated by these insights, the authors propose a method for reducing the mis-specification error amplification by augmenting the original feature set with auxiliary features and applying a “weighted ridge regression”, optimizing the weights to minimize the empirical Lebesgue constant. The authors offer theoretical guarantees for this novel regularization framework.

**Questions:**

Q1. For augmentation of auxiliary features to the original feature set (here, the original feature set consists of $d$ basis), we choose some integer $D > d$ and then add $D - d$ features to construct the full feature map $\varphi_{D} : \mathcal{X} \to \mathbb{R}^D$. In this framework, selection of an overly large integer $D$ will lead to a demanding computational cost since we need to solve a convex optimization problem in high-dimensional settings (the search space has dimension $D - d$). How can we control the trade-off between $D$ and the choice of $D - d$ suitable auxiliary features for the efficacy of the proposed regularization framework?

Q2. In the equation (2), I would like to confirm that the constant in the right-hand side of the inequality should be $1 + \Gamma_{d, \mu}$ instead of $\Gamma_{d, \mu}$.

**Ethical Concerns:**

["NO or VERY MINOR ethics concerns only"]

**Limitations:**

Yes.

**Paper Formatting Concerns:**

No.

**Quality:**

2

**Strengths And Weaknesses:**

(a) Strengths:

S1. The paper provides tight upper and lower bounds, showing that the dependent on the Lebesgue constant is intrinsic and cannot be substantially improved by any empirical squared-loss minimization method — the connection between classical approximation theory notions and value function approximation in RL and bandit problems is interesting.

S2. The paper elucidates limitation of standard estimators — the authors show that standard regularization techniques including ridge regression, cross-validation, and early-stopping cannot resolve the geometric amplification of error in LS regression, even with infinitely many data.

S3. The proposed framework based on weighted ridge regression with feature augmentation is novel — it can provably reduce the amplification factor to the oracle level.


(b) Weaknesses:

W1. The results are specific to linear function approximation; the paper does not address non-linear models or neural networks, which are increasingly prevalent in modern applications.

W2. The paper does not contain any detailed proofs of their theoretical results in the appendix, even though they seem concrete.

W3. The paper lacks experimental validations to support their theoretical results. Broader empirical validations on real-world datasets would strengthen the practical effectiveness.

---

> ### Author Rebuttal · Authors · 2025-07-30
>
> We thank the reviewer for their comments and the time dedicated to our paper.
>
> **W1**: We acknowledge that the linear case is limiting and that it remains to be seen whether our findings have any relevance to the more general, non-linear case. We still think that this setting is worthy of being studied thoroughly, despite the acknowledged limitations for the following reasons: (i) The linear setting is the simplest case where generalization (function approximation) happens. As such, it makes sense to investigate the linear case first. Accordingly, much attention have been spent to this setting in the theoretical RL/bandits literature. If one does not understand the linear case, there is little hope one will understand the nonlinear case. Some of the insights gained may in fact generalize to the nonlinear setting. This has been the case for a number of related problems (ie exploration can be addressed by optimism and without optimism of the right form, exploration algorithms remain statistically intractable). In our case, we think that some take-home message of this paper may also apply in the general case: In the limit of infinite data, learning can be seen as a map from data-distributions to regression functions. A fundamental insight of this paper is that extrapolation errors are essentially controlled by the discrepancy between this map and the $L^\infty$ projector. Another insight of this paper is that there can be better ways of using extra learning capacity than the naive approach. Combined with tools such as the NTK approach, one may hope to derive effective methods that also help in the nonlinear case, though. We will be happy to add a discussion of this to the paper, though the details will need to be left for future work.
>
> **W2**: We agree that missing proofs would be a serious flaw. All proofs, 47 pages in total, are included in the Supplementary Material zip file, which was uploaded with the submission. Another reviewer's review confirms that the zip file is accessible to them.
>
> **W3**: The goal of this work is primarily theoretical: we identify an intrinsic limitation of all squared-loss based empirical‑risk‑minimisation estimators in the uniform norm and propose a provably optimal remedy. The synthetic experiment in at page 8 already illustrates the phenomenon visually and confirms the theory. A full‑scale RL or bandit benchmark is certainly interesting, linear features (e.g., polynomial or Fourier bases) are standard in those settings, but such an evaluation would require introducing many additional domain‑specific concepts and would distract from the paper’s core contribution. Nevertheless, we plan to conduct this evaluation  and follow-up our paper with an empirical counterpart in the future. We also would like to emphasize that we think that even with the limited experiments the paper could be of great interest to the RL theory community, which, since the seminal paper of Du et al. (ICLR 2020) was struck by the price one may need to pay for misspecification. Our paper not only significantly refines the "dark picture" painted by the paper of Du et al. (namely, that polynomial sample complexity is only possible when the misspecification error is blown up by a factor of $\sqrt{d}$), in that we connect the blow-up factor to the Lebesgue constant associated with the learning setting, which, as we illustrate with examples can be much smaller than $O(\sqrt{d})$, but we also suggest a way of avoiding the blow-up with a novel technique.
>
> **Q1** For many basis function sequences, such as the Fourier or standard polynomial basis, choosing $D=d$ (ie doubling the number of dimensions) is provably sufficient, as we demonstrate it in our paper. The worst-case cost of the subgradient method used to solve the optimization problem in BWR (cf. Theorem 12) can still be relatively high ($O(d^2)$ for Fourier and $O(d^3)$ for polynomials; due to the different value of $\widehat \phi_{D,2}$)). However, we only need a low-accuracy solution to the optimization problem (ie in Theorem 12 we can take $\varepsilon=O(1)$), and as such, we observed that running the subgradient method was not the bottleneck. A more general idea is to use model-selection to tune $D$, following, e.g. [1], where the compute cost was also taken into account. However, more work is needed to see whether ideas like in this paper apply to our setting.
>
> **Q2** Yes, the reviewer is right, we will correct as soon as possible. Thanks for spotting it!
>
> [1] Agarwal, Alekh, et al. "Oracle inequalities for computationally budgeted model selection." Proceedings of the 24th Annual Conference on Learning Theory. JMLR Workshop and Conference Proceedings, 2011.

---

### Decision · Program_Chairs · 2025-09-17

**Decision:**

Accept (poster)

**Comment:**

This paper proves that, under random design and misspecification, the worst-case (uniform-norm) error of least-squares estimators is intrinsically amplified by the Lebesgue constant determined by the feature subspace and covariate distribution, with tight upper and lower bounds showing that no empirical squared-loss minimizer can substantially improve this dependence in general. Main commendations include sharp, conceptually clarifying theory: Lebesgue constant governs amplification; bounds are tight (all reviewers), novel method (BWR) that effectively reduces amplification and competes with an oracle (bUTv, UvJD, c8kL, and ZEuE), illuminating negative result for standard regularization (bUTv and ZEuE), quality of exposition and insight for practice (choice of bases) (UvJD, c8kL, and ZEuE), concrete case study demonstrating improvement (monomials) (UvJD, ZEuE, and c8kL). Main concerns include limited empirical validation; lack of real-data or RL experiments despite RL motivation (bUTv, c8kL, KqBU, and ZEuE), guidance on auxiliary features and model size; practicality of choosing how many/which features (UvJD, bUTv, and ZEuE). Given the paper's focus on theoretical investigation, I believe that these concerns about empirical studies are not substantial enough to diminish the paper's excellent contribution. Therefore, I recommend acceptance.